# Recursive Score Estimation Accelerates Diffusion-Based Monte Carlo

## Abstract

To sample from a general target distribution $p_* \propto e^{-f_*}$ beyond the isoperimetric condition, Huang et al. (2023) proposed to perform sampling through reverse diffusion, giving rise to *Diffusion-based Monte Carlo* (DMC). Specifically, DMC follows the reverse SDE of a diffusion process that transforms the target distribution to the standard Gaussian, utilizing a non-parametric score estimation. However, the original DMC algorithm encountered high gradient complexity[1], resulting in an *exponential dependency* on the error tolerance $\epsilon$ of the obtained samples. In this paper, we demonstrate that the high complexity of the original DMC algorithm originates from its redundant design of score estimation, and proposed a more efficient DMC algorithm, called RS-DMC, based on a novel recursive score estimation method. In particular, we first divide the entire diffusion process into multiple segments and then formulate the score estimation step (at any time step) as a series of interconnected mean estimation and sampling subproblems accordingly, which are correlated in a recursive manner. Importantly, we show that with a proper design of the segment decomposition, all sampling subproblems will only need to tackle a strongly log-concave distribution, which can be very efficient to solve using the standard sampler (e.g., Langevin Monte Carlo) with a provably rapid convergence rate. As a result, we prove that the gradient complexity of RS-DMC only has a *quasi-polynomial dependency* on $\epsilon$, which significantly improves exponential gradient complexity in Huang et al. (2023). Furthermore, under commonly used dissipative conditions, our algorithm is provably much faster than the popular Langevin-based algorithms. Our algorithm design and theoretical framework illuminate a novel direction for addressing sampling problems, which could be of broader applicability in the community.

## 1 Introduction

Sampling problems, i.e., generating samples from a given target distribution $p_* \propto \exp(-f_*)$, have received increasing attention in recent years. For resolving this problem, a popular option is to apply gradient-based Markov chain Monte Carlo (MCMC) methods, such as Unadjusted Langevin Algorithms (ULA) (Neal, 1992; Roberts & Tweedie, 1996), Underdamped Langevin Dynamics (ULD) (Cheng et al., 2018; Ma et al., 2021; Mou et al., 2021), Metropolis-Adjusted Langevin Algorithm (MALA) (Roberts & Stramer, 2002; Xifara et al., 2014), and Hamiltonian Monte Carlo (HMC) (Duane et al., 1987; Neal, 2010). In particular, these algorithms can be seen as the discretization of the continuous Langevin dynamics (LD) and its variants (Ma et al., 2015), which will converge to a unique stationary distribution that follows $p_* \propto \exp(-f_*)$, under regularity conditions on the energy function $f_*(\boldsymbol{x})$ (Roberts & Tweedie, 1996).

However, the convergence rate of the Langevin-based algorithms heavily depends on the target distribution $p_*$: guaranteeing the convergence in polynomial time requiring $p_*$ to have some nice properties, e.g., being strongly log-concave, satisfying log-Sobolev or Poincaré inequality with a large coefficient. However, for more general non-log-concave distributions, the convergence rate may exponentially depend on the problem dimension (Raginsky et al., 2017; Holzmüller & Bach, 2023) (i.e., $\sim \exp(d)$), or even the convergence itself (to $p_*$) cannot be guaranteed (one can only guarantee

---

[1] We denote gradient complexity as the required number of gradient calculations to achieve at most $\epsilon$ sampling error.

to converge to some locally stationary distribution (Balasubramanian et al., 2022)), implying that the Langevin-based algorithms are extremely inefficient for solving such hard sampling problems. To this end, we are interested in addressing the following question:

*Can we develop a new sampling algorithm that enjoys a non-exponential convergence rate for sampling general non-log-concave distributions?*

To address this problem, we are inspired by several recent studies, including Montanari (2023); Huang et al. (2023), that attempt to design samplers based on diffusion models (Sohl-Dickstein et al., 2015; Ho et al., 2020; Vargas et al., 2023), which we refer to as the **diffusion-based Monte Carlo** (DMC). In particular, the algorithm developed in Huang et al. (2023) is based on the reverse process of the Ornstein-Uhlenbeck (OU) process, which starts from the target distribution $p_*$ and converges to a standard Gaussian distribution. The mathematical formula of the OU process and its reverse process are given as follows (Anderson, 1982; Song et al., 2020):

$$d\mathbf{x}_t = -\mathbf{x}_t dt + \sqrt{2}dB_t, \quad \mathbf{x}_0 \sim p_0(\boldsymbol{x}) = p_*, \quad \text{(OU Process)}$$

$$d\mathbf{x}_t^{\leftarrow} = \left[\mathbf{x}_t^{\leftarrow} + 2\nabla\log p_{T-t}(\mathbf{x}_t^{\leftarrow})\right]dt + \sqrt{2}dB_t, \quad \mathbf{x}_0^{\leftarrow} \sim p_T(\boldsymbol{x}) \approx \mathcal{N}(\mathbf{0}, \mathbf{I}), \quad \text{(Reverse Process)}$$

where $B_t$ denotes the Brownian term, $p_t(\boldsymbol{x})$ denotes the underlying distribution of the particle at time $t$ along the OU process, $T$ denotes the end time of the OU process, and $\nabla\log p_t(\boldsymbol{x})$ denotes the score function of the distribution $p_t(\boldsymbol{x})$. In fact, the exponentially slow convergence rate of the Langevin-based algorithms stems from the rather long mixing time of Langevin dynamics to its stationary distribution, while in contrast, the OU process exhibits a much shorter mixing time. Therefore, principally, if the reverse process of the OU process can be perfectly recovered, one can avoid suffering from the issue of slow mixing of Langevin dynamics, and develop more efficient sampling algorithms accordingly.

Then, the key to recovering (Reverse Process) is to obtain a good estimation for the score $\nabla\log p_t(\boldsymbol{x})$ for all $t \in [0, T]$. Huang et al. (2023) proposed a score estimation method called reverse diffusion sampling (RDS) based on an inner-loop ULA. However, it still suffers from the exponential dependency with respect to the target sampling error, which requires $\exp\left(\mathcal{O}(1/\epsilon)\right)$ gradient complexity to achieve the $\epsilon$ sampling error in KL divergence. The reason behind this is that RDS involves many *hard* subproblems that need to sample non-log-concave distributions with bad isoperimetric properties, which incurs huge gradient complexities in the desired Langevin algorithms.

In this work, we argue that the *hard* subproblems in Huang et al. (2023) are redundant or even unnecessary, and propose a more efficient diffusion-based Monte Carlo method, called recursive score DMC (RS-DMC), that only requires **quasi-polynomial gradient complexity** to sampling general non-log-concave distributions. At the core of RS-DMC is a novel non-parametric method for score estimation, which involves a series of interconnected mean estimation and sampling subproblems that are correlated in a recursive manner. In particular, we first divide the entire forward process into several segments starting from $0, S, \ldots, (K-1)S$, and estimate the scores $\{\nabla\log p_{kS}(\boldsymbol{x})\}_{k=0,\ldots,K-1}$ recursively. Given the segments, the score within each segment $\nabla\log p_{kS+\tau}(\boldsymbol{x})$ will be further estimated according to the reference score $\nabla\log p_{kS}(\boldsymbol{x})$, where $\tau \in [0, S]$ can be arbitrarily chosen. Importantly, given proper configuration of the segment length (i.e., $S$), we can show that all sampling subproblems in the developed score estimation method are *much easier*, as long as the target distribution $p_*$ is log-smooth and has bounded second moment. Then, all intermediate target distributions are guaranteed to be strongly log-concave, which can be sampled very efficiently via standard ULA. We then summarize the main contributions of this paper as follows:

- We propose a new Diffusion Monte Carlo algorithm, called RS-DMC, for sampling general non-log-concave distributions. At the core is a novel and efficient recursive score estimation algorithm. In particular, based on a properly designed recursive structure, we show that the hard non-log-concave sampling problem can be divided into a series of benign sampling subproblems that can be solved very efficiently via standard ULA.

- We establish the convergence guarantee of the proposed RS-DMC algorithm under very mild assumptions, which only require the target distribution to be log-smooth and to have a bounded second moment. In contrast, to obtain provable convergence (to the target distribution), the Langevin-based methods typically require additional isoperimetric conditions (e.g., Log-Sobolev inequality, Poincaré inequality, etc). This justifies that our algorithm can be applied to a broader class of distributions with rigorous theoretical convergence guarantees.

- We prove that the gradient complexity of our algorithm is $\exp\left[\mathcal{O}(\log^3(d/\epsilon))\right]$ to achieve $\epsilon$ sampling error in KL divergence, which only has a quasi-polynomial dependency on the target error $\epsilon$ and dimension $d$. In contrast, under even stronger conditions in our work, the gradient complexity in prior works either need exponential dependency in $\epsilon$ (i.e., $\exp\left(\mathcal{O}(1/\epsilon)\right)$) (Huang et al., 2023) or exponential dependency in $d$, (i.e., $\exp\left(\mathcal{O}(d)\right)$) (Raginsky et al., 2017; Xu et al., 2018)[2] (which requires the additional dissipative condition). This demonstrate the efficiency of our algorithm.

## 2 PRELIMINARIES

In this section, we will first introduce the notations and problem settings that are commonly used in the following sections. We will then present some fundamental properties, such as the closed form of the transition kernel and the expectation form of score functions along the OU process. Finally, we will specify the assumptions that the target distribution is required in our algorithms and analysis.

**Notations.** We use lower case bold symbol $\mathbf{x}$ to denote the random vector, we use lower case italicized bold symbol $\boldsymbol{x}$ to denote a fixed vector. We use $\|\cdot\|$ to denote the standard Euclidean distance. We say $a_n = \text{poly}(n)$ if $a_n \leq O(n^c)$ for some constant $c$.

**The segmented OU process.** We define $\mathbb{N}_{a,b} = [a,b] \cap \mathbb{N}_*$ for brevity. Suppose the length of each segment is $S \in \mathbb{R}_+$, and we divide the entire forward process with length $T$ into $K \in \mathbb{N}_+$ segments satisfying $K = T/S$. In this condition, we can reformulate the previous SDE as

$$\begin{aligned} &\mathbf{x}_{k,0} \sim p_{0,0} = p_* \text{ when } k = 0, \text{ else } \mathbf{x}_{k,0} = \mathbf{x}_{k-1,S} \quad k \in \mathbb{N}_{0,K-1} \\ &\mathrm{d}\mathbf{x}_{k,t} = -\mathbf{x}_{k,t}\mathrm{d}t + \sqrt{2}\mathrm{d}B_t \qquad\qquad\qquad k \in \mathbb{N}_{0,K-1}, t \in [0,S], \end{aligned} \quad (1)$$

where $\mathbf{x}_{k,t}$ denotes the random variable of the OU process at time $(kS + t)$ with underlying density $p_{k,t}$. Besides, we define the following conditional density, i.e., $p_{(k,t)|(k',t')}(\boldsymbol{x}|\boldsymbol{x}')$, which presents the probability of obtaining $\mathbf{x}_{k,t} = \boldsymbol{x}$ when $\mathbf{x}_{k',t'} = \boldsymbol{x}'$. The diagram of SDE (1) is presented in Fig 1.

**The reverse segmented OU process.** According to (Reverse Process), the reverse process of the segmented SDE (1) can be presented as

$$\begin{aligned} &\mathbf{x}_{k,0}^{\leftarrow} \sim p_{K-1,S} \text{ when } k = K-1, \text{ else } \mathbf{x}_{k,0}^{\leftarrow} = \mathbf{x}_{k+1,S}^{\leftarrow} \quad k \in \mathbb{N}_{0,K-1} \\ &\mathrm{d}\mathbf{x}_{k,t}^{\leftarrow} = \left[\mathbf{x}_{k,t}^{\leftarrow} + 2\nabla \log p_{k,S-t}(\mathbf{x}_{k,t}^{\leftarrow})\right]\mathrm{d}t + \sqrt{2}\mathrm{d}B_t \qquad k \in \mathbb{N}_{0,K-1}, t \in [0,S] \end{aligned}$$

where particles satisfy $\mathbf{x}_{k,t}^{\leftarrow} = \mathbf{x}_{k,S-t}$ with underlying density $p_{k,t}^{\leftarrow} = p_{k,S-t}$ for any $k \in \mathbb{N}_{0,K-1}$ and $t \in [0,S]$. To approximately solve the SDE with numerical methods, we first split each segment into $R$ intervals $\{[(r-1)\eta, r\eta]\}_{=1,\ldots,R}$, where $\eta$ is the interval length and $R = S/\eta$. Then we can replace the score function $\nabla \log p_{k,S-t}$ as $\mathbf{v}_{k,t}^{\leftarrow}$, and for $t \in [r\eta, (r+1)\eta]$, we freeze the value of this coefficient in the SDE at time $(k, r\eta)$. Then starting from the standard Gaussian distribution, we consider the following new SDE:

$$\begin{aligned} &\mathbf{x}_{k,0}^{\leftarrow} \sim p_\infty = \mathcal{N}(\mathbf{0}, \boldsymbol{I}) \text{ when } k = K-1, \text{ else } \mathbf{x}_{k,0}^{\leftarrow} = \mathbf{x}_{k+1,S}^{\leftarrow} \quad k \in \mathbb{N}_{0,K-1} \\ &\mathrm{d}\mathbf{x}_{k,t}^{\leftarrow} = \left[\mathbf{x}_{k,t}^{\leftarrow} + 2\mathbf{v}_{k,\lfloor t/\eta\rfloor\eta}^{\leftarrow}\left(\mathbf{x}_{k,\lfloor t/\eta\rfloor\eta}^{\leftarrow}\right)\right]\mathrm{d}t + \sqrt{2}\mathrm{d}B_t \qquad k \in \mathbb{N}_{0,K-1}, t \in [0,S] \end{aligned} \quad (2)$$

where $p_\infty$ denotes the stationary distribution of the forward process. Similar to the segmented OU process, we define the following conditional density, i.e., $p_{(k,t)|(k',t')}^{\leftarrow}(\boldsymbol{x}|\boldsymbol{x}')$, which presents the probability of obtaining $\mathbf{x}_{k,t}^{\leftarrow} = \boldsymbol{x}$ when $\mathbf{x}_{k',t'}^{\leftarrow} = \boldsymbol{x}'$. The diagram of SDE (2) is presented in Fig 1.

**Basic properties of the OU process.** In the previous paragraph, we have demonstrated that SDE (1) is an alternative presentation of the OU process. Therefore, the properties in the OU process can be directly introduced for this segmented version. First, the transition kernel in the $k$-th segment satisfies

$$p_{(k,t)|(k,0)}(\boldsymbol{x}|\boldsymbol{x}_0) = \left(2\pi\left(1 - e^{-2t}\right)\right)^{-d/2} \cdot \exp\left[\frac{-\|\boldsymbol{x} - e^{-t}\boldsymbol{x}_0\|^2}{2\left(1 - e^{-2t}\right)}\right], \quad \forall\, 0 < t \leq S.$$

Plugging the transition kernel into Tweedie's formula, the score function can be reformulated as the following lemma whose proof is deferred in Section E.

---

[2]We omit the $d$-dependency in Huang et al. (2023) and $\epsilon$-dependency in Raginsky et al. (2017); Xu et al. (2018) for the ease of presentation.

The forward process

Figure 1: The illustration of SDE (1) and (2), covering the definitions in Section 2. The top of the figure describes the underlying distribution of the segmented OU process, i.e., SDE (1), and the bottom presents the corresponding distribution in the segmented OU process, i.e., SDE (2). For the intermediate part, the upper half describes the gradients of the log densities along the forward SDE (1), while the lower half describes approximated scores used to update particles in the reverse SDE (2).

**Lemma 2.1** (Lemma 1 of Huang et al. (2023)). *For any $k \in \mathbb{N}_{0,K-1}$ and $t \in [0, S]$, the score function can be written as*

$$\nabla \log p_{k,S-t}(\boldsymbol{x}) = \mathbb{E}_{\mathbf{x}_0 \sim q_{k,S-t}(\cdot|\boldsymbol{x})} \left[ -\frac{\boldsymbol{x} - e^{-(S-t)} \boldsymbol{x}_0}{\left(1 - e^{-2(S-t)}\right)} \right]$$

*where the conditional density function $q_{k,S-t}(\cdot|\boldsymbol{x})$ is defined as*

$$q_{k,S-t}(\boldsymbol{x}_0|\boldsymbol{x}) \propto \exp \left( \log p_{k,0}(\boldsymbol{x}_0) - \frac{\left\| \boldsymbol{x} - e^{-(S-t)} \boldsymbol{x}_0 \right\|^2}{2\left(1 - e^{-2(S-t)}\right)} \right).$$

Therefore, to approximate the score $\nabla \log p_{k,S-r\eta}(\boldsymbol{x})$ with an estimator $\mathbf{v}^{\leftarrow}_{k,r\eta}(\boldsymbol{x})$, we can draw samples from $q_{k,S-r\eta}(\cdot|\boldsymbol{x})$ and calculate their empirical mean.

**Assumptions.** To guarantee the convergence in KL divergence, the Langevin-based methods require the target distribution to satisfy certain isoperimetric properties such as Log-Sobolev inequality (LSI) and Poincaré inequality (PI) or even strong log-concavity (Vempala & Wibisono, 2019; Cheng & Bartlett, 2018; Dwivedi et al., 2018) (the formal definitions of these conditions are deferred to Section A). Some other works consider milder assumptions such as modified LSI Erdogdu & Hosseinzadeh (2021) and weak Poincaré inequality Mousavi-Hosseini et al. (2023), but they are only the analytical continuation of LSI and PI, which still exhibit a huge gap with the general non-log-concave distributions. Huang et al. (2023) requires the target distribution $p_*$ to have a heavier tail than that of the Gaussian distribution.

Remarkably, our algorithm does not require any isoperimetric condition or condition on the tail properties of $p_*$ to establish the convergence guarantee. We only require the following mild conditions on the target distribution.

[$\mathbf{A}_1$] For any $k \in \mathbb{N}_{0,K-1}$ and $t \in [0, S]$, the score $\nabla \log p_{k,t}$ is $L$-Lipschitz.

[$\mathbf{A}_2$] The target distribution has a bounded second moment, i.e., $M := \mathbb{E}_{p_*}[\|\cdot\|^2] < \infty$.

Assumption [$\mathbf{A}_1$] corresponds to the $L$-smoothness condition of the log density $f_*$ in traditional ULA analysis. It is often used to ensure that numerical discretization is feasible. We emphasize that Assumption [$\mathbf{A}_1$] can be easily relaxed to only assume the target distribution is smooth rather than the entire OU process, based on the technique in Chen et al. (2023a) (see their Lemmas 12 and 14). We do not include this additional relaxation in this paper to make our analysis clearer. Assumption [$\mathbf{A}_2$] are widely used in the common analysis of traditional gradient-based MCMC.

## 3 Proposed Methods

In this section, we introduce a new approach called Recursive Score Estimation (RSE) and describe the proposed Recursive Score Diffusion-based Monte Carlo (RS-DMC) method. We start by discussing the motivations and intuitions behind the use of recursion. Next, we provide implementation details for the RSE process and emphasize the importance of selecting an appropriate segment length. Finally, we present the RS-DMC method based on the RSE approach.

### 3.1 Difficulties of the vanilla DMC

We consider the reverse segmented OU process, i.e., SDE 2 and begin with the original version of DMC in Huang et al. (2023), which can be seen as a special case of the reverse segmented OU process with a large segment length $S = T$ and a small number of segments $K = 1$. According to the reverse SDE 2, for the $r$-th iteration within one single segment, we need to estimate $\nabla \log p_{0,S-r\eta}$ to update the particles. Specifically, by Lemma 2.1, we have

$$\nabla \log p_{0,S-r\eta}(\boldsymbol{x}) = \mathbb{E}_{\mathbf{x}_0 \sim q_{0,S-r\eta}(\cdot|\boldsymbol{x})} \left[ -\frac{\boldsymbol{x} - e^{-(S-r\eta)}\boldsymbol{x}_0}{\left(1 - e^{-2(S-r\eta)}\right)} \right]$$

for any $\boldsymbol{x} \in \mathbb{R}^d$, where the conditional distribution $q_{0,S-r\eta}(\cdot|\boldsymbol{x})$ is

$$q_{0,S-r\eta}(\boldsymbol{x}_0|\boldsymbol{x}) \propto \exp\left( \log p_{0,0}(\boldsymbol{x}_0) - \frac{\left\| \boldsymbol{x} - e^{-(S-r\eta)}\boldsymbol{x}_0 \right\|^2}{2\left(1 - e^{-2(S-r\eta)}\right)} \right). \tag{3}$$

Since the analytic form $\nabla \log p_{0,0} = -f_*$ exists, we can use the ULA to draw samples from $q_{0,S-r\eta}(\cdot|\boldsymbol{x})$ and calculate the empirical mean to estimate $\nabla \log p_{0,S-r\eta}(\boldsymbol{x})$.

However, sampling from $q_{0,S-r\eta}(\cdot|\boldsymbol{x})$ is not an easy task. When $r$ is very small, sampling $q_{0,S-r\eta}(\cdot|\boldsymbol{x})$ via ULA is almost as difficult as sampling $p_{0,0}(\boldsymbol{x}_0)$ via ULA (see (3)), since the additive quadratic term, whose coefficient is $e^{-2(S-r\eta)}/2(1 - e^{-2(S-r\eta)})$, will be nearly negligible in this case. This is because that $S = T$ is large and then $e^{-2(S-r\eta)}/2(1 - e^{-2(S-r\eta)}) \sim \exp(-2T)$ becomes extremely small when $r\eta = O(T)$. More specifically, as shown in Huang et al. (2023), when $e^{-2(S-r\eta)} \leq 2L/(1 + 2L)$, the LSI parameter of $q_{0,S-r\eta}(\cdot|\boldsymbol{x})$ can be as worse as $\exp\left(-\mathcal{O}(1/\epsilon)\right)$. Then applying ULA for sampling this distribution needs a dramatically high gradient complexity that is exponential in $1/\epsilon$.

### 3.2 Intuition of the recursion

Therefore, the key to avoiding sampling such a hard distribution is to restrict the segment length. By Lemma 2.1, it can be straightforwardly verified that if the segment length satisfies $S \leq \frac{1}{2} \log\left(\frac{2L+1}{2L}\right)$,

$$-\nabla_{\boldsymbol{x}_0}^2 \log q_{k,S-r\eta}(\boldsymbol{x}_0|\boldsymbol{x}) \succeq -\nabla_{\boldsymbol{x}_0}^2 \log p_{k,0}(\boldsymbol{x}_0) + \frac{e^{-2S}}{1 - e^{-2S}} \cdot \boldsymbol{I} \succeq \frac{e^{-2S}}{2(1 - e^{-2S})} \tag{4}$$

where the last inequality follows from Assumption [$\mathbf{A}_1$]. This implies that $q_{k,S-r\eta}(\boldsymbol{x}_0|\boldsymbol{x})$ is strongly log-concave for all $r \leq \lfloor S/\eta \rfloor$, which can be efficiently sampled via the standard ULA. However, ULA requires to calculate the score function $\nabla_{\boldsymbol{x}_0} \log q_{k,S-r\eta}(\boldsymbol{x}_0|\boldsymbol{x})$, which further needs to calculate $\nabla \log p_{k,0}(\boldsymbol{x})$ according to Lemma 2.1. Different from the vanilla DMC where the formula of $\nabla \log p_{0,0}(\boldsymbol{x})$ is known, the score $\nabla \log p_{k,0}(\boldsymbol{x})$ in (4) is an unknown quantity, which also requires to be estimated. In fact, based on our definition, we can rewrite $p_{k,0}(\boldsymbol{x})$ as $p_{k-1,S}(\boldsymbol{x})$ (see Figure 1), then applying Lemma 2.1, we can again decompose the problem of estimating $\nabla \log p_{k-1,S}(\boldsymbol{x})$ into the subproblems of sampling $q_{k-1,S}(\cdot|\boldsymbol{x})$ and the estimation of $\nabla \log p_{k-1,0}(\boldsymbol{x})$, which is naturally organized in a recursive manner. Therefore, by recursively adopting this subproblem decomposition, we summarize the recursive process for approximating $\nabla \log p_{k,S-r\eta}(\boldsymbol{x})$ as follows and illustrate the diagram in Figure 2:

- **Step 1:** We approximate the score $\nabla \log p_{k,S-r\eta}(\boldsymbol{x})$ by a mean estimation with samples generated by running ULA over the intermediate target distribution $q_{k,S-r\eta}(\cdot|\boldsymbol{x})$.
- **Step 2:** When running ULA for $q_{k,S-t}(\cdot|\boldsymbol{x})$, we estimate the score $\nabla \log p_{k,0} = \nabla \log p_{k-1,S}$.
- **Step 3:** We jump to Step 1 to approximate the score $\nabla \log p_{k-1,S}(\boldsymbol{x})$ via drawing samples from $q_{k-1,S}(\cdot|\boldsymbol{x})$, and continue the recursion.

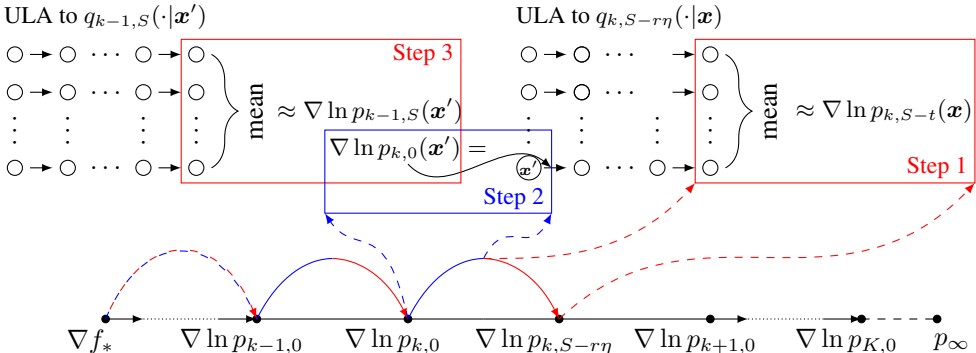

Figure 2: The illustration of recursive score estimation (RSE). The upper half presents RSE from a local view, which shows how to utilize the former score, e.g., $\nabla \log p_{k,0}(\boldsymbol{x}')$ to update particles by ULA in the sampling subproblem formulated by the latter score, e.g., $\nabla \log p_{k,S-t}(\boldsymbol{x})$. The lower half presents RSE from a global view, which is a series of interconnected mean estimation and sampling subproblems accordingly.

## 3.3 RECURSIVE SCORE ESTIMATION AND REVERSE DIFFUSION SAMPLING

**Recursive Score Estimation.** In the previous section, we explained the rough intuition behind introducing recursion. By conducting the recursion, we need to solve a series of sampling and mean estimation subproblems. Then, it is demanding to control the error propagation between these subproblems in order to finally ensure small sampling errors. In particular, this amounts to the adaptive adjustment of the sample numbers for mean estimation and iteration numbers for ULA in solving sampling subproblems. Specifically, if we require score estimation $\mathbf{v}^{\leftarrow}_{k,r\eta} \colon \mathbb{R}^d \to \mathbb{R}^d$ to satisfy

$$\left\| \nabla \log p_{k,S-r\eta}(\boldsymbol{x}) - \mathbf{v}^{\leftarrow}_{k,r\eta}(\boldsymbol{x}) \right\|^2 \le \epsilon, \ \forall \boldsymbol{x} \in \mathbb{R}^d \tag{5}$$

with a high probability, then the sample number in Step 1 and the number of calls of Step 2 (the iteration number of ULA) in Fig 2 will be two functions with respected to the target error $\epsilon$, denoted as $n_{k,r}(\epsilon)$ and $m_{k,r}(\epsilon)$ respectively. Furthermore, when Step 2 is introduced to update ULA, we rely on an approximation of $\nabla \log p_{k,0}$ instead of the exact score. To ensure (5) is met, the error resulting from estimating $\nabla \log p_{k,0}$ should be typically smaller than $\epsilon$. We express this requirement as:

$$\left\| \nabla \log p_{k,0}(\boldsymbol{x}) - \mathbf{v}^{\leftarrow}_{k,0}(\boldsymbol{x}) \right\|^2 \le l_{k,r}(\epsilon), \ \forall \boldsymbol{x} \in \mathbb{R}^d.$$

where $l_{k,r}(\epsilon)$ is a function of $\epsilon$ that satisfies $l_{k,r}(\epsilon) \le \epsilon$. Under this condition, we provide Alg 1, i.e., RSE, to calculate the score function for the $r$-th iteration at the $k$-th segment, i.e., $\nabla \log p_{k,S-r\eta}(\boldsymbol{x})$.

**Quasi-polynomial Complexity.** We consider the ideal case for interpreting the complexity of our score estimation method. In particular, since the benign error propagation, i.e., $l_{k,r}(\epsilon) = \epsilon$, is almost proven in Lemma E.7, we suppose the number of calls to the recursive function, $\mathrm{RSE}(k - 1, 0, \boldsymbol{x}', l_{k,r}(\epsilon))$, is uniformly bounded by $m_{k,r}(\epsilon) \cdot n_{k,r}(\epsilon)$ for all feasible $(k, r)$ pairs when the RSE algorithm is executed with input $(k, r, \boldsymbol{x}, \epsilon)$. Then, recall that we will conduct the recursion in at most $K$ rounds, the total gradient complexity for estimating one score will be

$$[m_{k,r}(\epsilon) \cdot n_{k,r}(\epsilon)]^{\mathcal{O}(K)} = [m_{k,r}(\epsilon) \cdot n_{k,r}(\epsilon)]^{\mathcal{O}(T/S)}.$$

This formula highlights the importance of selecting a sufficiently large segment with length $S$ to reduce the number of recursive function calls and improve gradient complexity. In our analysis, we set $S = \frac{1}{2} \log \left( \frac{2L+1}{2L} \right)$, which is "just" small enough to ensure that all intermediate target distributions in the sampling subproblems are strongly log-concave. In this condition, due to the choice of $T$ is $\mathcal{O}(\log(d/\epsilon))$ in general cases and $m_{k,r}(\cdot)$ and $n_{k,r}(\cdot)$ are typically polynomial w.r.t. the target sampling error $\epsilon$ and dimension $d$ (see Theorem B.1 in Appendix B), we would expect a quasi-polynomial gradient complexity.

**Diffusion-based Monte Carlo with Recursive Score Estimation.** Then based on the RSE algorithm in Alg 1, we can directly apply the DDPM (Ho et al., 2020) based method to perform the sampling, giving rise to the Recursive Score Diffusion-based Monte Carlo (RS-DMC) method. We summarize the proposed RS-DMC algorithm in Alg 2 (the detailed setup of $m_{k,r}(\cdot)$, $n_{k,r}(\cdot)$, $l_{k,r}(\cdot)$ are provided in Theorem B.1 in Appendix B).

---

**Algorithm 1** Recursive Score Estimation (approximate $\nabla \log p_{k,S-r\eta}(\boldsymbol{x})$): $\mathsf{RSE}(k,r,\boldsymbol{x},\epsilon)$

---

1: **Input:** The segment number $k \in \mathbb{N}_{0,K-1}$, the iteration number $r \in \mathbb{N}_{0,R-1}$, variable $\boldsymbol{x}$ requiring the score function, error tolerance $\epsilon$.
2: **if** $k \equiv -1$ **then return** $-\nabla f_*(\boldsymbol{x})$
3: Initial the returned vector $\boldsymbol{v}' \leftarrow \boldsymbol{0}$
4: **for** $i = 1$ to $n_{k,r}(\epsilon)$ **do**
5:     Draw $\boldsymbol{x}'_0$ from an initial distribution $q'_0$
6:     **for** $j = 0$ to $m_{k,r}(\epsilon, \boldsymbol{x}) - 1$ **do**
7:         $\boldsymbol{v}'_j \leftarrow \mathsf{RSE}\left(k-1, 0, \boldsymbol{x}'_j, l_{k,r}(\epsilon)\right)$         ▷ Recursive score estimation $\nabla \log p_{k-1,S}(\boldsymbol{x}'_j)$
8:         **if** $r \not\equiv 0$ **then** $t' \leftarrow S - r\eta$ **else** $t' \leftarrow S$         ▷ The gap of time since the last call
9:         Update the particle

$$\boldsymbol{x}'_{j+1} := \boldsymbol{x}'_j + \tau_r \cdot \underbrace{\left(\boldsymbol{v}'_j + \frac{e^{-t'}\boldsymbol{x} - e^{-2t'}\boldsymbol{x}'_j}{1 - e^{-2t'}}\right)}_{\approx \nabla \log q_{k,S-r\eta}(\boldsymbol{x}'_j | \boldsymbol{x})} + \sqrt{2\tau_r} \cdot \xi$$

10:     Update the score estimation of $\boldsymbol{v}' \approx \nabla \log p_{k,S-r\eta}(\boldsymbol{x})$ with empirical mean as

$$\boldsymbol{v}' := \boldsymbol{v}' + \frac{1}{n_{k,r}(\epsilon)}\left(-\frac{\boldsymbol{x} - e^{-t'}\boldsymbol{x}'_{m_{k,r}(\epsilon)}}{1 - e^{-2t'}}\right)$$

    **return** $\boldsymbol{v}'$.         ▷ As the approximation of $\nabla \log p_{k,S-r\eta}(\boldsymbol{x})$

---

**Algorithm 2** Recursive Score Diffusion-based Monte Carlo (RS-DMC)

---

**Input:** Initial particle $\mathbf{x}^{\leftarrow}_{K,S}$ sampled from $p_\infty$, Terminal time $T$, Step size $\eta$, required convergence accuracy $\epsilon$;
**for** $k = K - 1$ down to $0$ **do**
    Initialize the particle as $\boldsymbol{x}^{\leftarrow}_{k,0} \leftarrow \boldsymbol{x}^{\leftarrow}_{k+1,S}$
    **for** $r = 0$ to $R - 1$ **do**
        Approximate the score, i.e., $\nabla \log p_{k,S-r\eta}(\boldsymbol{x}^{\leftarrow}_{k,r\eta})$ by $\boldsymbol{v}' \leftarrow \mathsf{RSE}(k, r, \boldsymbol{x}^{\leftarrow}_{k,r\eta}, l(\epsilon))$
        $\boldsymbol{x}^{\leftarrow}_{k,(r+1)\eta} \leftarrow e^\eta \boldsymbol{x}^{\leftarrow}_{k,r\eta} + (e^\eta - 1)\boldsymbol{v}' + \xi$ where $\xi$ is sampled from $\mathcal{N}\left(0, \left(e^{2\eta} - 1\right)\boldsymbol{I}_d\right)$
**Return:** $\boldsymbol{x}^{\leftarrow}_{0,S}$.

---

## 4 Analysis of RS-DMC

In this section, we will establish the convergence guarantee for RS-DMC and reveal how the gradient complexity depends on the problem dimension and the target sampling error. We will also compare the gradient complexity of RS-DMC with other sampling methods to justify its strength. Additionally, we will provide a proof roadmap that briefly summarizes the critical theoretical techniques.

### 4.1 Theoretical Results

The following theorem states that RS-DMC can provably converge to the target distribution in KL-divergence with quasi-polynomial gradient complexity.

**Theorem 4.1** (Gradient complexity of RS-DMC, informal). *Under Assumptions [$A_1$]-[$A_2$], let $p^{\leftarrow}_{0,S}$ be the distribution of the samples generated by RS-DMC, then there exists a collection of appropriate hyperparameters $n_{k,r}, m_{k,r}, \tau_r, \eta, l_{k,r}$ and $l$ such that with probability at least $1 - \epsilon$, it holds that* $\mathrm{KL}\left(p_* \| p^{\leftarrow}_{0,S}\right) = \tilde{O}(\epsilon)$. *Besides, the gradient complexity of RS-DMC is*

$$\exp\left[\mathcal{O}\left(\log^3\left((Ld + M)/\epsilon\right) \cdot \max\left\{\log\log Z^2, 1\right\}\right)\right], \tag{6}$$

*where $Z$ denotes the maximum norm of particles which appears in Alg 2.*

We defer the detailed configurations of $n_{k,r}, m_{k,r}, \tau_r, \eta, l_{k,r}, l$ and relative constants in the formal version of this theorem, i.e., Theorem B.1 Appendix B and Table 2 in Appendix A, respectively. Then, we show a comparison between our method and previous work.

**Comparison with ULA.** The gradient complexity of ULA has been well studied for sampling the non-log-concave distribution. However, in order to prove the convergence in KL divergence or TV distance, they typically require additional isoperimetric conditions, such as Log-Soboleve and Poincaré inequality (see Definitions 1 and 2). In particular, when $p_*$ satisfies LSI with parameter $\alpha$, Vempala & Wibisono (2019) proved the $\mathcal{O}\left(d\epsilon^{-1}\alpha^{-2}\right)$ in KL convergence. However, for general non-log-concave distributions, $\alpha$ is not dimension-free. For instance, under the Dissipative condition (Hale, 2010), $\alpha$ can be as worse as $\exp(-\mathcal{O}(d))$ (Raginsky et al., 2017), leading to a $\exp(\mathcal{O}(d))$ gradient complexity results (Xu et al., 2018).

When the isoperimetric condition is absent, Balasubramanian et al. (2022) proved the convergence of ULA based on the Fisher information measure, i.e., $\text{FI}\left(p\|p_*\right) := \mathbb{E}_p[\|\nabla \log(p/p_*)\|^2]$, they showed that ULA can generate the samples that satisfy $\text{FI}\left(p\|p_*\right) \leq \epsilon$ for some small error tolerance $\epsilon$. However, it may be unclear what can be entailed by such a guarantee $\text{FI}\left(p\|p_*\right) \leq \epsilon$. It has demonstrated that, in some cases, even if the Fisher information $\text{FI}\left(p\|p_*\right)$ is very small, the total variation distance/KL divergence remains bounded away from zero. This suggests that the convergence guarantee in Fisher information might be weaker than that in KL divergence (i.e., our convergence guarantee).

**Comparison with RDS.** Then we make a detailed comparison with RDS in (Huang et al., 2023), which is the most similar algorithm compared to ours. Firstly, we would like to strengthen again that our convergence results are obtained on a milder assumption, while Huang et al. (2023) additionally requires the target distribution to have a heavier tail. Besides, as discussed in the introduction section, RDS has a much worse gradient complexity since it performs all score estimation straightforwardly, while RS-DMC is based on a recursive structure. Consequently, RDS involves many hard sampling subproblems that take exponential time to solve, while RS-DMC only involves strongly log-concave subsampling problems that can be efficiently solved within polynomial time. As a result, the gradient complexity of RDS is proved to be $\text{poly}(d) \cdot \text{poly}(1/\epsilon) \cdot \exp\left(\mathcal{O}(1/\epsilon)\right)$, which is significantly worse than the quasi-polynomial gradient complexity of RS-DMC.

## 4.2 PROOF SKETCH

In this section, we aim to highlight the technical innovations by presenting the roadmap of our analysis. Due to space constraints, we have included the technical details in the Appendix.

Firstly, by requiring Novikov's conditions, we can establish an upper bound on the KL divergence gap between the target distribution $p_*$ and the underlying distribution of output particles, i.e., $p_{0,S}^{\leftarrow}$, by Girsanov's Theorem which demonstrates

$$\text{KL}\left(p_*\|p_{0,S}^{\leftarrow}\right) \leq \underbrace{\text{KL}\left(p_{K-1,S}\|p_{K-1,0}^{\leftarrow}\right)}_{\text{Term 1}} + \underbrace{2\sum_{k=0}^{K-1}\sum_{r=0}^{R-1}\int_0^\eta \mathbb{E}_{\mathbf{x}_{k,r\eta}^{\leftarrow}}\left[\left\|\nabla\log p_{k,S-r\eta}(\mathbf{x}_{k,r\eta}^{\leftarrow}) - \mathbf{v}_{k,r\eta}^{\leftarrow}(\mathbf{x}_{k,r\eta}^{\leftarrow})\right\|^2\right]\mathrm{d}t}_{\text{Term 3}}$$

$$+ \underbrace{2\sum_{k=0}^{K-1}\sum_{r=0}^{R-1}\int_0^\eta \mathbb{E}_{(\mathbf{x}_{k,t+r\eta}^{\leftarrow},\mathbf{x}_{k,r\eta}^{\leftarrow})}\left[\left\|\nabla\log p_{k,S-(t+r\eta)}(\mathbf{x}_{k,t+r\eta}^{\leftarrow}) - \nabla\log p_{k,S-r\eta}(\mathbf{x}_{k,r\eta}^{\leftarrow})\right\|^2\right]\mathrm{d}t}_{\text{Term 2}}.$$

Although Novikov's condition may not be met in general, we employ techniques in Chen et al. (2023a) and sidestep this issue by utilizing a differential inequality argument as shown in Lemma F.3.

**Upper bound Term 1.** Intuitively, Term 1 appears since we utilize the standard Gaussian to initialize the reverse OU process (SDE (2)) rather than $p_{K-1,S}$ which can hardly be sampled from directly in practice. Therefore, the first term can be bounded using exponential mixing of the forward (Ornstein-Uhlenbeck) process towards the standard Gaussian in Lemma C.3, i.e.,

$$\text{KL}\left(p_{K-1,S}\|p_{K-1,0}^{\leftarrow}\right) \leq \text{KL}\left(p_*\|p_{K-1,0}^{\leftarrow}\right)\exp(-KS) \leq (Ld + M)\exp(-KS),$$

where $p_{K-1,0}^{\leftarrow} = \mathcal{N}(\mathbf{0}, \mathbf{I})$ as shown SDE (2).

**Upper bound Term 2.** Term 2 corresponds to the discretization error, which has been successfully addressed in previous work Chen et al. (2023b;a). By utilizing the unique structure of the Ornstein-Uhlenbeck process, they managed to limit both the time and space discretization errors, which

decrease as $\eta$ becomes smaller. To ensure the completeness of our proof, we have included it in Lemma D.4, utilizing the segmented notation.

**Upper bound Term 3.** Term 3 represents the accuracy of the score estimation. In diffusion models, due to the parameterization of the target density, this term is trained by a neural network and assumed to be less than $\epsilon$ to ensure the convergence of the reverse process. However, in RS-DMC, the score estimation is obtained using a non-parametric approach, i.e., Alg 1. To this end, we can provide rigorous high probability bound for this term under Alg 1, which is stated in Lemma E.10.

Roughly speaking, for Alg 2 with input each $(k, r, \boldsymbol{x}, \epsilon)$, suppose the score estimation of $\nabla \log p_{k,0}$ is given as $\mathbf{v}_{k-1,0}^{\leftarrow}$ satisfying the following event

$$\bigcap_{\boldsymbol{x}' \in \mathbb{S}_{k,r}(\boldsymbol{x}, \epsilon)} \left\| \nabla \log p_{k,0}(\boldsymbol{x}') - \mathbf{v}_{k-1,0}^{\leftarrow}(\boldsymbol{x}') \right\|^2 \leq l_{k,r}(\epsilon)$$

where $\mathbb{S}_{k,r}(\boldsymbol{x}, \epsilon)$ denotes the set of particles appear in Alg 1 except for the recursion. In this condition, Lemma E.7 provides the upper bound of score estimation error as:

$$\left\| \mathbf{v}_{k,r\eta}^{\leftarrow}(\boldsymbol{x}) - \nabla \log p_{k,S-r\eta}(\boldsymbol{x}) \right\|^2 \leq \frac{2e^{-2(S-r\eta)}}{\left(1 - e^{-2(S-r\eta)}\right)^2} \cdot \underbrace{\left\| -\frac{1}{n_{r,k}(\epsilon)} \sum_{i=1}^{n_{r,k}(\epsilon)} \mathbf{x}_i' + \mathbb{E}_{\mathbf{x}' \sim q_{k,S-r\eta}'(\cdot|\boldsymbol{x})}\left[\mathbf{x}'\right] \right\|^2}_{\text{Term 3.1}}$$

$$+ \frac{2e^{-2(S-r\eta)}}{\left(1 - e^{-2(S-r\eta)}\right)^2} \cdot \underbrace{\left\| -\mathbb{E}_{\mathbf{x}' \sim q_{k,S-r\eta}'(\cdot|\boldsymbol{x})}\left[\mathbf{x}'\right] + \mathbb{E}_{\mathbf{x}' \sim q_{k,S-r\eta}(\cdot|\boldsymbol{x})}\left[\mathbf{x}'\right] \right\|^2}_{\text{Term 3.2}}$$

where $q_{k,S-r\eta}'(\cdot|\boldsymbol{x})$ is the underlying distribution of output particles, i.e., $\mathbf{x}_{m_{k,r}(l_{k,r}(\epsilon))}'$ in Alg 1. Considering that the distribution $q_{k,S-r\eta}$ is strongly log-concave (given in Eq. 4) and we can get a lower bound on the strongly log-concave constant (see Lemma E.2). Therefore, $q_{k,S-r\eta}'$ also satisfies the log-Sobolev inequality due to Lemma F.8, which can imply the variance upper bound (see Lemma F.11). Then, in our proof, we directly make use of the Sobolev inequality to derive the high-probability bound (or concentration results) for estimating the mean of $q_{k,S-r\eta}'(\cdot|\boldsymbol{x})$ in Term 3.1 with Lemma E.7 by selecting sufficiently large $n_{k,r}(\epsilon)$. Besides, Term 3.2 can be upper bounded by $\mathrm{KL}\left(q_{k,S-r\eta}'(\cdot|\boldsymbol{x}) \| q_{k,S-r\eta}(\cdot|\boldsymbol{x})\right)$, which can be well controlled by conducting the ULA with a sufficiently large iteration number $m_{k,r}(\epsilon)$. Therefore, by conducting the following decomposition

$$\mathbb{P}\left[ \left\| \nabla \log p_{k,S-r\eta}(\boldsymbol{x}_{k,r\eta}^{\leftarrow}) - \mathbf{v}_{k,r\eta}^{\leftarrow}(\boldsymbol{x}_{k,r\eta}^{\leftarrow}) \right\|^2 \leq \epsilon \right]$$

$$\geq (1-\delta)\mathbb{P}\left[ \bigcap_{\boldsymbol{x}' \in \mathbb{S}_{k,r}(\boldsymbol{x}, \epsilon)} \left\| \nabla \log p_{k,0}(\boldsymbol{x}') - \mathbf{v}_{k-1,0}^{\leftarrow}(\boldsymbol{x}') \right\|^2 \leq l_{k,r}(\epsilon) \right].$$

We only need to use this proof process recursively with a proper choice of $\delta$ ($\delta$ as a function of $\epsilon$) to get the bound:

$$\mathbb{P}\left[ \left\| \nabla \log p_{k,S-r\eta}(\boldsymbol{x}_{k,r\eta}^{\leftarrow}) - \mathbf{v}_{k,r\eta}^{\leftarrow}(\boldsymbol{x}_{k,r\eta}^{\leftarrow}) \right\|^2 \leq \epsilon \right] \geq 1 - \epsilon,$$

which implies Term 3 $\leq \tilde{O}(\epsilon)$ with a probability at least $1-\epsilon$. Due to the large amount of computation, we defer the details of the recursive proof procedure and the choice of $\delta$ to the Appendix E.3.

## 5 CONCLUSION

In this paper, we propose a novel non-parametric score estimation algorithm, i.e., RSE, presented in Alg 1 and derive its corresponding reverse diffusion sampling algorithm, i.e., RS-DMC, and outlined in Alg 2. By introducing the segment length $S$ to balance the challenges of score estimation and recursive calls, RS-DMC exhibits several advantages over Langevin-based MCMC, e.g., ULA, ULD, and MALA. It can achieve KL convergence beyond isoperimetric target distributions with a quasi-polynomial gradient complexity, i.e.,

$$\exp\left[\mathcal{O}(\log^3(d/\epsilon) \cdot \max\left\{\log \log Z^2, 1\right\})\right].$$

Additionally, the theoretical result also demonstrates the efficiency of RS-DMC in challenging sampling tasks. To the best of our knowledge, this is the first work that eliminates the exponential dependence with only smoothness and the second moment bounded assumptions.

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

CONTENTS

A   NOTATIONS

In this section, we summarize the notations defined in Section 2 in Table 1 for easy reference and cross-checking. Additionally, another important notation is the score estimation, denoted as $\mathbf{v}^{\leftarrow}_{k,r\eta}$, which is used to approximate $\nabla \log p_{k,S-r\eta}$. When $r = 0$, $\mathbf{v}^{\leftarrow}_{k,0}$ is expected to approximate $\nabla \log p_{k,S}$ which is not explicitly defined in SDE 1. However, sine $\mathbf{x}_{k,S} = \mathbf{x}_{k+1,0}$ in Eq 1, the underlying distributions, i.e., $p_{k,S}$ and $p_{k+1,0}$, are equal, and $\tilde{\mathbf{v}}_{k,0}$ can be considered as the score estimation of $\nabla \log p_{k+1,0}$. For $\nabla \log p_{0,0}$, which can be calculated exactly as $\nabla f_*$, we define

$$\mathbf{v}^{\leftarrow}_{-1,0}(\boldsymbol{x}) = \nabla \log p_{0,0}(\boldsymbol{x}) = -\nabla f_*(\boldsymbol{x}) \tag{7}$$

as a complement.

| Symbols | Description |
|---|---|
| $\varphi_{\sigma^2}$ | The density function of the centered Gaussian distribution, i.e., $\mathcal{N}\left(\mathbf{0}, \sigma^2 \boldsymbol{I}\right)$. |
| $p_*, p_{0,0}$ | The target density function (initial distribution of the forward process) |
| $\{\mathbf{x}_{k,t}\}_{k\in\mathbb{N}_{0,K-1}, t\in[0,S]}$ | The forward process, i.e., SDE 1 |
| $p_{k,t}$ | The density function of $\mathbf{x}_{k,t}$, i.e., $\mathbf{x}_{k,t} \sim p_{k,t}$ |
| $p_\infty$ | The density function of the stationary distribution of the forward process |
| $\{\mathbf{x}^{\leftarrow}_{k,t}\}_{k\in\mathbb{N}_{0,K-1}, t\in[0,S]}$ | The practical reverse process following from SDE 2 with initial distribution $p_\infty$ |
| $p^{\leftarrow}_{k,t}$ | The density function of $\mathbf{x}^{\leftarrow}_{k,t}$, i.e., $\mathbf{x}^{\leftarrow}_{k,t} \sim p^{\leftarrow}_{k,t}$ |

Table 1: The list of notations defined in Section 2, where $\mathbb{N}_{a,b}$ is denoted as the set of natural numbers from $a \in \mathbb{N}_*$ to any $b \in \mathbb{N}_+$.

| Constant symbol | Value | Constant symbol | Value |
|---|---|---|---|
| $C_\eta$ | $2^{-14}L^{-2}$ | $C_{m,1}$ | $\log\left(2M \cdot 3^2 \cdot 5L\right) + M \cdot 3L$ |
| $C_n$ | $2^6 \cdot 5^2 \cdot C_\eta^{-1}$ | $C_m$ | $2^9 \cdot 3^2 \cdot 5^3 \cdot C_{m,1}C_\eta^{-1.5}$ |
| $C_{u,1}$ | $\log\left(\frac{5C_nC_m}{10^4}\right) + \log\left(2\max\left\{\log Z, \frac{1}{2}\right\}\right)$ | $C_{u,2}$ | $70/S^2 + 10/S$ |
| $C_{u,3}$ | $2C_{u,1}/S$ | $S$ | $1/2\log((2L+1)/2L)$ |

Table 2: Constant List independent with $\epsilon$ and $d$.

**Isopermetric conditions and assumptions.** According to the classical theory of Markov chains and diffusion processes, some conditions can lead to fast convergence over time without being as strict as log concavity. Isoperimetric inequalities, such as the log-Sobolev inequality (LSI) or the Poincaré inequality (PI), are examples of these conditions defined as follows.

**Definition 1** (Logarithmic Sobolev inequality). *A distribution with density function $p$ satisfies the log-Sobolev inequality with a constant $\mu > 0$ if for all smooth function $g\colon \mathbb{R}^d \to \mathbb{R}$ with $\mathbb{E}_p[g^2] \leq \infty$,*

$$\mathbb{E}_p\left[g^2 \log g^2\right] - \mathbb{E}_p\left[g^2\right]\log\mathbb{E}_p\left[g^2\right] \leq 2\alpha^{-1}\mathbb{E}_p\left[\|\nabla g\|^2\right].$$

By supposing $g = 1 + \epsilon\hat{g}$ with $\epsilon \to 0$, a weaker isoperimetric inequality, i.e., PI can be defined Menz & Schlichting (2014).

**Definition 2** (Poincaré inequality). *A distribution with density function $p$ satisfies the Poincaré inequality with a constant $\mu > 0$ if for all smooth function $\hat{g}\colon \mathbb{R}^d \to \mathbb{R}$,*

$$\mathrm{Var}(\hat{g}) \leq \alpha^{-1}\mathbb{E}_p\left[\|\nabla\hat{g}\|^2\right].$$

We also provide a list of constants used in our following proof in Table 2 to prevent confusion.

## B  PROOF OF THEOREM 4.1

**Theorem B.1.** *The formal version of Theorem 4.1 In Alg 2, suppose we set*

$$S = 1/2 \cdot \log(1 + 1/2L), \quad K = 2\log[(Ld+M)/\epsilon] \cdot S^{-1},$$
$$\eta = C_\eta(M+d)^{-1}\epsilon, \quad R = S/\eta,$$
$$l(\epsilon) = 10\epsilon, \quad l_{k,r}(\epsilon) = \epsilon/960,$$
$$n_{k,r}(\epsilon) = C_n \cdot (d+M)\epsilon^{-2} \cdot \max\{d, -2\log\delta\},$$
$$m_{k,r}(\epsilon, \boldsymbol{x}) = C_m \cdot (d+M)^3\epsilon^{-3} \cdot \max\{\log\|\boldsymbol{x}\|^2, 1\},$$
$$\tau_r = 2^{-5} \cdot 3^{-2} \cdot e^{2(S-r\eta)}\left(1 - e^{-2(S-r\eta)}\right)^2 \cdot d^{-1}\epsilon$$

*where $\delta$ satisfies*

$$\delta = \mathrm{pow}\left(2, -\frac{2}{S}\log\frac{Ld+M}{\epsilon}\right) \cdot \mathrm{pow}\left(\frac{C_\eta S \epsilon^2}{4(d+M)} \cdot \log^{-2}\left(\frac{Ld+M}{\epsilon}\right) \cdot \mathrm{pow}\left(\left(\frac{Ld+M}{\epsilon}\right),\right.\right.$$
$$\left.\left. -C_{u,2}\log\frac{Ld+M}{\epsilon} - C_{u,3}\right), \frac{2}{S}\log\frac{Ld+M}{\epsilon}+1\right),$$

*and the initial underlying distribution $q_0'$ of the Alg 1 with input $(k, r, \boldsymbol{x}, \epsilon)$ satisfies*

$$q_0'(\boldsymbol{x}') \propto \exp\left(-\frac{\left\|\boldsymbol{x} - e^{-(S-r\eta)}\boldsymbol{x}'\right\|^2}{2(1 - e^{-2(S-r\eta)})}\right),$$

*we have*

$$\mathbb{P}\left[\mathrm{KL}\left(\hat{p}_{0,S}\|p_{0,S}^\leftarrow\right) = \tilde{O}(\epsilon)\right] \geq 1 - \epsilon.$$

*In this condition, the gradient complexity will be*

$$\exp\left[\mathcal{O}\left(\log^3\left((Ld+M)/\epsilon\right) \cdot \max\left\{\log\log Z^2, 1\right\}\right)\right]$$

*where $Z$ is the maximal norm of particles appeared in Alg 2.*

*Proof of Theorem B.1.* According to Lemma F.3, suppose $\hat{\mathbf{x}}_{k,t} = \mathbf{x}_{k,S-t}$ whose SDE can be presented as

$$\mathbf{x}_{k,0}^\leftarrow \sim p_{K-1,S} \text{ when } k = K-1, \text{ else } \mathbf{x}_{k,0}^\leftarrow = \mathbf{x}_{k+1,S}^\leftarrow \quad k \in \mathbb{N}_{0,K-1}$$
$$\mathrm{d}\mathbf{x}_{k,t}^\leftarrow = \left[\mathbf{x}_{k,t}^\leftarrow + 2\nabla\log p_{k,S-t}(\mathbf{x}_{k,t}^\leftarrow)\right]\mathrm{d}t + \sqrt{2}\mathrm{d}B_t \quad k \in \mathbb{N}_{0,K-1}, t \in [0,S]$$

due to Chen et al. (2023b). Then, we have $\mathrm{KL}\left(p_*\|p_{0,S}^\leftarrow\right) = \mathrm{KL}\left(\hat{p}_{0,S}\|p_{0,S}^\leftarrow\right)$ which satisfies

$$\mathrm{KL}\left(\hat{p}_{0,S}\|p_{0,S}^\leftarrow\right) \leq \underbrace{\mathrm{KL}\left(\hat{p}_{K-1,0}\|p_{K-1,0}^\leftarrow\right)}_{\text{Term 1}}$$
$$+ \sum_{k=0}^{K-1}\sum_{r=0}^{R-1}\int_0^\eta \mathbb{E}_{(\hat{\mathbf{x}}_{k,t+r\eta}, \hat{\mathbf{x}}_{k,r\eta})}\left[\left\|\nabla\log p_{k,S-(t+r\eta)}(\hat{\mathbf{x}}_{k,t+r\eta}) - \mathbf{v}_{k,r\eta}^\leftarrow(\hat{\mathbf{x}}_{k,r\eta})\right\|^2\right]\mathrm{d}t. \tag{8}$$

**Upper bound Term 1.** Term 1 can be upper-bounded as

$$\text{Term 1} = \mathrm{KL}\left(p_{K-1,S}\|p_{K-1,0}^\leftarrow\right) \leq (Ld+M)\cdot\exp\left(-KS/2\right)$$

with Lemma C.3 when $p_{K-1,0}^\leftarrow$ is chosen as the standard Gaussian. Therefore, we choose

$$S = \frac{1}{2}\log\frac{2L+1}{2L}, \quad K = 2\log\frac{Ld+M}{\epsilon}\cdot\left(\frac{1}{2}\log\frac{2L+1}{2L}\right)^{-1}, \quad \text{and} \quad KS \geq 2\log\frac{Ld+M}{\epsilon},$$

which make the inequality Term 1 $\leq \epsilon$ establish.

For the remaining term of RHS of Eq 8, it can be decomposed as follows:

$$\sum_{k=0}^{K-1}\sum_{r=0}^{R-1}\int_0^\eta \mathbb{E}_{(\hat{\mathbf{x}}_{k,t+r\eta}, \hat{\mathbf{x}}_{k,r\eta})}\left[\left\|\nabla\log p_{k,S-(t+r\eta)}(\hat{\mathbf{x}}_{k,t+r\eta}) - \mathbf{v}_{k,r\eta}^\leftarrow(\hat{\mathbf{x}}_{k,r\eta})\right\|^2\right]\mathrm{d}t$$
$$\leq 2\underbrace{\sum_{k=0}^{K-1}\sum_{r=0}^{R-1}\int_0^\eta \mathbb{E}\left[\left\|\nabla\log p_{k,S-(t+r\eta)}(\hat{\mathbf{x}}_{k,t+r\eta}) - \nabla\log p_{k,S-r\eta}(\hat{\mathbf{x}}_{k,r\eta})\right\|^2\right]\mathrm{d}t}_{\text{Term 2}} \tag{9}$$
$$+ 2\underbrace{\sum_{k=0}^{K-1}\sum_{r=0}^{R-1}\int_0^\eta \mathbb{E}_{(\hat{\mathbf{x}}_{k,t+r\eta}, \hat{\mathbf{x}}_{k,r\eta})}\left[\left\|\nabla\log p_{k,S-r\eta}(\hat{\mathbf{x}}_{k,r\eta}) - \mathbf{v}_{k,r\eta}^\leftarrow(\hat{\mathbf{x}}_{k,r\eta})\right\|^2\right]\mathrm{d}t}_{\text{Term 3}}$$

**Upper bound Term 2.** This term is mainly from the discretization error in the reverse process. Therefore, its analysis is highly related to Chen et al. (2023b;a). To ensure the completeness of our proof, we have included it in our analysis, utilizing the segmented notation presented in Section A. Specifically, we have

$$
\begin{aligned}
\text{Term 2} \leq &4 \sum_{k=0}^{K-1} \sum_{r=0}^{R-1} \int_0^\eta \mathbb{E}\left[\left\|\nabla \log p_{k,S-(t+r\eta)}(\hat{\mathbf{x}}_{k,t+r\eta}) - \nabla \log p_{k,S-(t+r\eta)}(\hat{\mathbf{x}}_{k,r\eta})\right\|^2\right] \\
&+ 4 \sum_{k=0}^{K-1} \sum_{r=0}^{R-1} \int_0^\eta \mathbb{E}\left[\left\|\nabla \log \frac{p_{k,S-(t+r\eta)}(\hat{\mathbf{x}}_{k,r\eta})}{p_{k,S-r\eta}(\hat{\mathbf{x}}_{k,r\eta})}\right\|^2\right] \mathrm{d}t \\
\leq &4 \sum_{k=0}^{K-1} \sum_{r=0}^{R-1} \int_0^\eta \left(\mathbb{E}\left[L^2 \left\|\hat{\mathbf{x}}_{k,t+r\eta} - \hat{\mathbf{x}}_{k,r\eta}\right\|^2\right] + \mathbb{E}\left[\left\|\nabla \log \frac{p_{k,S-r\eta}(\hat{\mathbf{x}}_{k,r\eta})}{p_{k,S-(t+r\eta)}(\hat{\mathbf{x}}_{k,r\eta})}\right\|^2\right]\right) \mathrm{d}t
\end{aligned}
$$

where the last inequality follows from Assumption $[\mathbf{A}_1]$. Combining this result with Lemma D.4, when the stepsize, i.e., $\eta$ of the reverse process is $\eta = C_\eta (M+d)^{-1}\epsilon$, then it has Term 2 $\leq \epsilon$.

**Upper bound Term 3.** Due to the randomness of $\mathbf{v}_{k,r\eta}^{\leftarrow}$, we consider a high probability bound, which is formulated as

$$
\mathbb{P}\left[\bigcap_{\substack{k\in\mathbb{N}_{0,K-1} \\ r\in\mathbb{N}_{0,R-1}}} \left\|\nabla \log p_{k,S-r\eta}(\mathbf{x}_{k,r\eta}^{\leftarrow}) - \mathbf{v}_{k,r\eta}^{\leftarrow}(\mathbf{x}_{k,r\eta}^{\leftarrow})\right\|^2 \leq 10\epsilon\right] \geq 1 - \epsilon, \tag{10}
$$

which means we choose $l(\epsilon) = 10\epsilon$. Lemma E.10 demonstrate that under the following settings, i.e.,

$$
\begin{aligned}
l_{k,r}(\epsilon) &= \epsilon/960, \\
n_{k,r}(\epsilon) &= C_n \cdot (d+M)\epsilon^{-2} \cdot \max\{d, -2\log\delta\}, \\
m_{k,r}(\epsilon, \boldsymbol{x}) &= C_m \cdot (d+M)^3 \epsilon^{-3} \cdot \max\{\log\|\boldsymbol{x}\|^2, 1\},
\end{aligned}
$$

where $\delta$ satisfies

$$
\begin{aligned}
\delta :=&\text{pow}\left(2, -\frac{2}{S}\log\frac{Ld+M}{\epsilon}\right) \cdot \text{pow}\left(\frac{C_\eta S\epsilon^2}{4(d+M)} \cdot \log^{-2}\left(\frac{Ld+M}{\epsilon}\right) \cdot \text{pow}\left(\left(\frac{Ld+M}{\epsilon}\right), \right.\right. \\
&\left.\left. -C_{u,2}\log\frac{Ld+M}{\epsilon} - C_{u,3}\right), \frac{2}{S}\log\frac{Ld+M}{\epsilon} + 1\right),
\end{aligned}
$$

Eq 10 can be achieved with a gradient complexity:

$$
\exp\left[\mathcal{O}\left(\log^3\left((Ld+M)/\epsilon\right) \cdot \max\left\{\log\log Z^2, 1\right\}\right)\right]
$$

where $Z$ is the maximal norm of particles appeared in Alg 2. All constants can be found in Table 2. In this condition, we have

$$
\text{Term 3} \leq 4 \cdot \frac{T}{\eta} \cdot (\eta \cdot 10\epsilon) \leq 40\epsilon\log\frac{Ld+M}{\epsilon} = \tilde{O}(\epsilon).
$$

Combining the upper bound of Term 1, Term 2 and Term 3, we have

$$
\text{KL}\left(\hat{p}_{0,S}\|p_{0,S}^{\leftarrow}\right) = \tilde{O}(\epsilon).
$$

The proof is completed. $\qquad\square$

**Corollary B.2.** *Suppose we set all parameters except for $\delta$ to be the same as that in Theorem B.1, and define*

$$
\begin{aligned}
\delta =&\text{pow}\left(2, -\frac{2}{S}\log\frac{Ld+M}{\epsilon}\right) \cdot \text{pow}\left(\frac{C_\eta S\epsilon\delta'}{4(d+M)} \cdot \log^{-2}\left(\frac{Ld+M}{\epsilon}\right) \cdot \text{pow}\left(\left(\frac{Ld+M}{\epsilon}\right), \right.\right. \\
&\left.\left. -C_{u,2}\log\frac{Ld+M}{\epsilon} - C_{u,3}\right), \frac{2}{S}\log\frac{Ld+M}{\epsilon} + 1\right),
\end{aligned}
$$

*we have*

$$\mathbb{P}\left[\mathrm{KL}\left(\hat{p}_{0,S}\|p_{0,S}^{\leftarrow}\right) = \tilde{O}(\epsilon)\right] \geq 1 - \delta'.$$

*In this condition, the gradient complexity will be*

$$\exp\left[\mathcal{O}\left(\max\left\{\left(\log\frac{Ld+M}{\epsilon}\right)^3, \log\frac{Ld+M}{\epsilon}\cdot\log\frac{1}{\delta'}\right\}\cdot\max\left\{\log\log Z^2, 1\right\}\right)\right]$$

*where $Z$ is the maximal norm of particles appeared in Alg 2.*

*Proof.* In this corollary, we follow the same proof roadmap as that shown in Theorem B.1. Combining Eq 8 and Eq 9, we have

$$\mathrm{KL}\left(\hat{p}_{0,S}\|p_{0,S}^{\leftarrow}\right) \leq \underbrace{\mathrm{KL}\left(\hat{p}_{K-1,0}\|p_{K-1,0}^{\leftarrow}\right)}_{\text{Term 1}}$$

$$\leq \underbrace{2\sum_{k=0}^{K-1}\sum_{r=0}^{R-1}\int_0^\eta \mathbb{E}\left[\left\|\nabla\log p_{k,S-(t+r\eta)}(\hat{\mathbf{x}}_{k,t+r\eta}) - \nabla\log p_{k,S-r\eta}(\hat{\mathbf{x}}_{k,r\eta})\right\|^2\right]\mathrm{d}t}_{\text{Term 2}} \quad (11)$$

$$+ \underbrace{2\sum_{k=0}^{K-1}\sum_{r=0}^{R-1}\int_0^\eta \mathbb{E}_{(\hat{\mathbf{x}}_{k,t+r\eta},\hat{\mathbf{x}}_{k,r\eta})}\left[\left\|\nabla\log p_{k,S-r\eta}(\hat{\mathbf{x}}_{k,r\eta}) - \mathbf{v}_{k,r\eta}^{\leftarrow}(\hat{\mathbf{x}}_{k,r\eta})\right\|^2\right]\mathrm{d}t}_{\text{Term 3}}$$

It should be noted that the techniques for upper-bounding Term 1 and Term 2 are the same as that in Theorem B.1.

**Upper bound Term 3.** Due to the randomness of $\mathbf{v}_{k,r\eta}^{\leftarrow}$, we consider a high probability bound, which is formulated as

$$\mathbb{P}\left[\bigcap_{\substack{k\in\mathbb{N}_{0,K-1}\\r\in\mathbb{N}_{0,R-1}}}\left\|\nabla\log p_{k,S-r\eta}(\mathbf{x}_{k,r\eta}^{\leftarrow}) - \mathbf{v}_{k,r\eta}^{\leftarrow}(\mathbf{x}_{k,r\eta}^{\leftarrow})\right\|^2 \leq 10\epsilon\right] \geq 1 - \delta', \quad (12)$$

which means we choose $l(\epsilon) = 10\epsilon$. Lemma E.11 demonstrate that under the following settings, i.e.,

$$l_{k,r}(\epsilon) = \epsilon/960,$$
$$n_{k,r}(\epsilon) = C_n\cdot(d+M)\epsilon^{-2}\cdot\max\{d, -2\log\delta\},$$
$$m_{k,r}(\epsilon,\boldsymbol{x}) = C_m\cdot(d+M)^3\epsilon^{-3}\cdot\max\{\log\|\boldsymbol{x}\|^2, 1\},$$

where $\delta$ satisfies

$$\delta := \mathrm{pow}\left(2, -\frac{2}{S}\log\frac{Ld+M}{\epsilon}\right)\cdot\mathrm{pow}\left(\frac{C_\eta S\epsilon\delta'}{4(d+M)}\cdot\log^{-2}\left(\frac{Ld+M}{\epsilon}\right)\cdot\mathrm{pow}\left(\left(\frac{Ld+M}{\epsilon}\right),\right.\right.$$
$$\left.\left.-C_{u,2}\log\frac{Ld+M}{\epsilon} - C_{u,3}\right), \frac{2}{S}\log\frac{Ld+M}{\epsilon} + 1\right),$$

Eq 12 can be achieved with a gradient complexity:

$$\exp\left[\mathcal{O}\left(\max\left\{\left(\log\frac{Ld+M}{\epsilon}\right)^3, \log\frac{Ld+M}{\epsilon}\cdot\log\frac{1}{\delta'}\right\}\cdot\max\left\{\log\log Z^2, 1\right\}\right)\right]$$

where $Z$ is the maximal norm of particles appeared in Alg 2. All constants can be found in Table 2. In this condition, we have

$$\mathrm{Term\ 3} \leq 4\cdot\frac{T}{\eta}\cdot(\eta\cdot 10\epsilon) \leq 40\epsilon\log\frac{Ld+M}{\epsilon} = \tilde{O}(\epsilon).$$

Combining the upper bound of Term 1, Term 2 and Term 3, we have

$$\mathrm{KL}\left(\hat{p}_{0,S}\|p_{0,S}^{\leftarrow}\right) = \tilde{O}(\epsilon).$$

The proof is completed. $\qquad\square$

## C  LEMMAS FOR BOUNDING INITIALIZATION ERROR

**Lemma C.1** (Lemma 11 in Vempala & Wibisono (2019)). *Suppose $p \propto \exp(-f)$ and $f \colon \mathbb{R}^d \to \mathbb{R}$ is L-gradient Lipschitz continuous function. Then, we have*

$$\mathbb{E}_{\mathbf{x} \sim p} \left[ \|\nabla f(\mathbf{x})\|^2 \right] \leq Ld$$

**Lemma C.2.** *Under the notation in Section A, suppose $p \propto \exp(-f)$ satisfies Assumption [A₁] and [A₂], then we have*

$$\mathrm{KL} \left( p \| \varphi_1 \right) \leq Ld + M$$

*Proof.* From the analytic form of the standard Gaussian, we have $\nabla^2 \log \varphi_1 = \boldsymbol{I}$. Combining this fact with Lemma F.4, we have

$$\mathrm{KL} \left( p \| \varphi_1 \right) \leq \frac{1}{2} \int p(\boldsymbol{x}) \left\| \nabla \log \frac{p(\boldsymbol{x})}{\varphi_1(\boldsymbol{x})} \right\|^2 \mathrm{d}\boldsymbol{x}$$

$$\leq \int p(\boldsymbol{x}) \|\nabla f(\boldsymbol{x})\|^2 \mathrm{d}\boldsymbol{x} + \int p(\boldsymbol{x}) \|\boldsymbol{x}\|^2 \mathrm{d}\boldsymbol{x} \leq Ld + M.$$

where the last inequality follows from Lemma C.1 and Assumption [A₂]. Hence, the proof is completed. □

**Lemma C.3** (Variant of Theorem 4 in Vempala & Wibisono (2019)). *Under the notation in Section A, suppose $\tilde{p}_{K-1,0}$ is chosen as the standard Gaussian distribution. Then, we have*

$$\mathrm{KL} \left( p_{K-1,S} \| p_\infty \right) \leq (Ld + M) \cdot \exp \left( -KS/2 \right).$$

*Proof.* Suppose another random variable $\mathbf{z}_t := \mathbf{x}_{\lfloor t/S \rfloor, t - \lfloor t/S \rfloor \cdot S}$ where $\mathbf{x}_{k,t}$ is shown in SDE 1, we have

$$\mathrm{d}\mathbf{z}_t = -\mathbf{z}_t \mathrm{d}t + \sqrt{2} \mathrm{d}B_t, \quad \mathbf{z}_0 = \mathbf{x}_{0,0},$$

where the underlying distribution of $\mathbf{x}_{0,0}$ satisfies $p_{0,0} = p_* \propto \exp(-f_*)$. If we denote $\mathbf{z}_t \sim p_t^{(z)}$, then Fokker-Planck equation of the previous SDE will be

$$\partial_t p_t^{(z)}(\boldsymbol{z}) = \nabla \cdot \left( p_t^{(z)}(\boldsymbol{z}) \boldsymbol{z} \right) + \Delta p_t^{(z)}(\boldsymbol{z}) = \nabla \cdot \left( p_t^{(z)}(\boldsymbol{z}) \nabla \log \frac{p_t^{(z)}(\boldsymbol{z})}{\exp \left( -\frac{1}{2} \|\boldsymbol{z}\|^2 \right)} \right).$$

It implies that the stationary distribution is standard Gaussian, i.e., $p_\infty^{(z)} \propto \exp(-1/2 \cdot \|\boldsymbol{z}\|^2)$. Then, we consider the KL convergence of $(\mathbf{z}_t)_{t \geq 0}$, and have

$$\frac{\mathrm{dKL} \left( p_t^{(z)} \| p_\infty^{(z)} \right)}{\mathrm{d}t} = \frac{\mathrm{d}}{\mathrm{d}t} \int p_t^{(z)}(\boldsymbol{z}) \log \frac{p_t^{(z)}(\boldsymbol{z})}{p_\infty^{(z)}(\boldsymbol{z})} \mathrm{d}\boldsymbol{z} = \int \partial_t p_t^{(z)}(\boldsymbol{z}) \log \frac{p_t^{(z)}(\boldsymbol{z})}{p_\infty^{(z)}(\boldsymbol{z})} \mathrm{d}\boldsymbol{z}$$

$$= \int \nabla \cdot \left( p_t^{(z)}(\boldsymbol{z}) \nabla \log \frac{p_t^{(z)}(\boldsymbol{z})}{p_\infty^{(z)}(\boldsymbol{z})} \right) \cdot \log \frac{p_t^{(z)}(\boldsymbol{z})}{p_\infty^{(z)}(\boldsymbol{z})} \mathrm{d}\boldsymbol{z} = - \int p_t^{(z)}(\boldsymbol{z}) \left\| \nabla \log \frac{p_t^{(z)}(\boldsymbol{z})}{p_\infty^{(z)}(\boldsymbol{z})} \right\|^2 \mathrm{d}\boldsymbol{z}. \tag{13}$$

Combining the fact $\nabla^2 (-\log p_\infty^{(z)}) = \boldsymbol{I}$ and Lemma F.4, we have

$$\mathrm{KL} \left( p_t^{(z)} \| p_\infty^{(z)} \right) \leq 2 \int p_t^{(z)}(\boldsymbol{z}) \left\| \nabla \log \frac{p_t^{(z)}(\boldsymbol{z})}{p_\infty^{(z)}(\boldsymbol{z})} \right\|^2 \mathrm{d}\boldsymbol{z}.$$

Plugging this inequality into Eq 13, we have

$$\frac{\mathrm{dKL} \left( p_t^{(z)} \| p_\infty^{(z)} \right)}{\mathrm{d}t} = - \int p_t^{(z)}(\boldsymbol{z}) \left\| \nabla \log \frac{p_t^{(z)}(\boldsymbol{z})}{p_\infty^{(z)}(\boldsymbol{z})} \right\|^2 \mathrm{d}\boldsymbol{z} \leq -\frac{1}{2} \mathrm{KL} \left( p_t^{(z)} \| p_\infty^{(z)} \right).$$

Integrating implies the desired bound, i.e.,

$$\mathrm{KL} \left( p_t^{(z)} \| p_\infty^{(z)} \right) \leq \exp \left( -t/2 \right) \cdot \mathrm{KL} \left( p_0^{(z)} \| p_\infty^{(z)} \right) \leq (Ld + M) \cdot \exp \left( -t/2 \right)$$

where the last inequality follows from Lemma C.2. It implies KL divergence between the underlying distribution of $\mathbf{x}_{K-1,S}$ and $p_\infty$ is

$$\mathrm{KL} \left( p_{K-1,S} \| p_\infty \right) = \mathrm{KL} \left( p_{KS}^{(z)} \| p_\infty^{(z)} \right) \leq (Ld + M) \cdot \exp \left( -KS/2 \right)$$

Hence, the proof is completed. □

# D   LEMMAS FOR BOUNDING DISCRETIZATION ERROR.

**Lemma D.1** (Lemma C.11 in Lee et al. (2022)). *Suppose that $p(\boldsymbol{x}) \propto e^{-f(\boldsymbol{x})}$ is a probability density function on $\mathbb{R}^d$, where $f(\boldsymbol{x})$ is $L$-smooth, and let $\varphi_{\sigma^2}(\boldsymbol{x})$ be the density function of $\mathcal{N}(\mathbf{0}, \sigma^2 \boldsymbol{I}_d)$. Then for $L \leq \frac{1}{2\sigma^2}$, it has*

$$\left\| \nabla \log \frac{p(\boldsymbol{x})}{(p * \varphi_{\sigma^2})(\boldsymbol{x})} \right\| \leq 6L\sigma d^{1/2} + 2L\sigma^2 \left\| \nabla f(\boldsymbol{x}) \right\|.$$

**Lemma D.2** (Lemma 9 in Chen et al. (2023b)). *Under the notation in Section A, suppose that Assumption [A$_1$] and [A$_2$] hold. For any $k \in \mathbb{N}_{0, K-1}$ and $t \in [0, S]$, we have*

1. *Moment bound, i.e.,*

$$\mathbb{E}\left[ \|\mathbf{x}_{k,t}\|^2 \right] \leq d \vee M.$$

2. *Score function bound, i.e.,*

$$\mathbb{E}\left[ \|\nabla \log p_{k,t}(\mathbf{x}_{k,t})\|^2 \right] \leq Ld.$$

**Lemma D.3** (Variant of Lemma 10 in Chen et al. (2023b)). *Under the notation in Section A, Suppose that Assumption [A$_2$] holds. For any $k \in \{0, 1, \dots, K-1\}$ and $0 \leq s \leq t \leq S$, we have*

$$\mathbb{E}\left[ \|\mathbf{x}_{k,t} - \mathbf{x}_{k,s}\|^2 \right] \leq 2\left(M + d\right) \cdot (t-s)^2 + 4d \cdot (t-s)$$

*Proof.* According to the forward process, we have

$$\mathbb{E}\left[ \|\mathbf{x}_{k,t} - \mathbf{x}_{k,s}\|^2 \right] = \mathbb{E}\left[ \left\| \int_s^t -\mathbf{x}_{k,r} \mathrm{d}r + \sqrt{2}\left(B_t - B_s\right) \right\|^2 \right] \leq \mathbb{E}\left[ 2 \left\| \int_s^t \mathbf{x}_{k,r} \mathrm{d}r \right\|^2 + 4 \|B_t - B_s\|^2 \right]$$

$$\leq 2\mathbb{E}\left[ \left( \int_s^t \|\mathbf{x}_{k,r}\| \, \mathrm{d}r \right)^2 \right] + 4d \cdot (t-s) \leq 2 \int_s^t \mathbb{E}\left[ \|\mathbf{x}_{k,r}\|^2 \right] \mathrm{d}r \cdot (t-s) + 4d \cdot (t-s)$$

$$\leq 2\left(M + d\right) \cdot (t-s)^2 + 4d \cdot (t-s),$$

where the third inequality follows from Holder's inequality and the last one follows from Lemma D.2. Hence, the proof is completed. $\square$

**Lemma D.4** (Errors from the discretization). *Under the notation in Section A, if the step size of the outer loops satisfies*

$$\eta \leq C_1(d + M)^{-1}\epsilon,$$

*then, for any $k \in \{0, 1, \dots, K-1\}$, $r \in \{0, 1, \dots, R-1\}$ and $t \in [0, \eta]$, we have*

$$\mathbb{E}\left[ L^2 \|\hat{\mathbf{x}}_{k,t+r\eta} - \hat{\mathbf{x}}_{k,r\eta}\|^2 \right] + \mathbb{E}\left[ \left\| \nabla \log \frac{p_{k,S-r\eta}(\hat{\mathbf{x}}_{k,r\eta})}{p_{k,S-(t+r\eta)}(\hat{\mathbf{x}}_{k,r\eta})} \right\|^2 \right] \leq 4\epsilon.$$

*Proof.* We consider the following formulation with any $t \in [0, \eta]$,

$$\text{Term 2} = \underbrace{\mathbb{E}\left[ \left\| \nabla \log \frac{p_{k,S-r\eta}(\hat{\mathbf{x}}_{k,r\eta})}{p_{k,S-(t+r\eta)}(\hat{\mathbf{x}}_{k,r\eta})} \right\|^2 \right]}_{\text{Term 2.1}} + \mathbb{E}\left[ L^2 \|\hat{\mathbf{x}}_{k,t+r\eta} - \hat{\mathbf{x}}_{k,r\eta}\|^2 \right]. \tag{14}$$

**Upper bound Term 2.1.**   To establish the connection between $p_{k,S-r\eta}$ and $p_{k,S-(t+r\eta)}$, due to the transition kernel of the forward process (OU process), we have

$$p_{k,S-r\eta}(\boldsymbol{x}) = \int p_{k,S-(r\eta+t)}(\boldsymbol{y}) \cdot \mathbb{P}\left[ \boldsymbol{x}, (k, S-r\eta)|\boldsymbol{y}, (S-(r\eta+t)) \right] \mathrm{d}\boldsymbol{y}$$

$$= \int p_{k,S-(r\eta+t)}(\boldsymbol{y}) \cdot \left(2\pi \left(1 - e^{-2t}\right)\right)^{-\frac{d}{2}} \cdot \exp\left[ \frac{-\|\boldsymbol{x} - e^{-t}\boldsymbol{y}\|^2}{2(1 - e^{-2t})} \right] \mathrm{d}\boldsymbol{y} \tag{15}$$

$$= \int e^{td} p_{k,S-(r\eta+t)}(e^t\boldsymbol{z}) \cdot \left(2\pi \left(1 - e^{-2t}\right)\right)^{-\frac{d}{2}} \cdot \exp\left[ \frac{-\|\boldsymbol{x} - \boldsymbol{z}\|^2}{2(1 - e^{-2t})} \right] \mathrm{d}\boldsymbol{z}$$

where the last equation follows from setting $\boldsymbol{z} := e^{-t}\boldsymbol{y}$. We define

$$p'_{k,S-(r\eta+t)}(\boldsymbol{z}) := e^{td}p_{k,S-(r\eta+t)}(e^t\boldsymbol{z})$$

which is also a density function. Therefore, for each element $\hat{\mathbf{x}}_{k,r\eta} = \boldsymbol{x}$, we have

$$\left\| \nabla \log \frac{p_{k,S-(r\eta+t)}(\boldsymbol{x})}{p_{k,S-r\eta}(\boldsymbol{x})} \right\|^2 \leq 2 \left\| \nabla \log \frac{p_{k,S-(r\eta+t)}(\boldsymbol{x})}{p'_{k,S-(r\eta+t)}(\boldsymbol{x})} \right\|^2 + 2 \left\| \nabla \log \frac{p'_{k,S-(r\eta+t)}(\boldsymbol{x})}{p_{k,S-r\eta}(\boldsymbol{x})} \right\|^2$$

$$= 2 \left\| \nabla \log \frac{p_{k,S-(r\eta+t)}(\boldsymbol{x})}{p'_{k,S-(r\eta+t)}(\boldsymbol{x})} \right\|^2 + 2 \left\| \nabla \log \frac{p'_{k,S-(r\eta+t)}(\boldsymbol{x})}{p'_{k,S-(r\eta+t)} * \varphi_{(1-e^{-2t})}(\boldsymbol{x})} \right\|^2$$

where the last inequality follows from Eq 15. For the first term, we have

$$\left\| \nabla \log \frac{p_{k,S-(r\eta+t)}(\boldsymbol{x})}{p'_{k,S-(r\eta+t)}(\boldsymbol{x})} \right\| = \left\| \nabla \log p_{k,S-(r\eta+t)}(\boldsymbol{x}) - e^t \cdot \nabla \log p_{k,S-(r\eta+t)}(e^t\boldsymbol{x}) \right\|$$

$$\leq \left\| \nabla \log p_{k,S-(r\eta+t)}(\boldsymbol{x}) - e^t \nabla \log p_{k,S-(r\eta+t)}(\boldsymbol{x}) \right\| \tag{16}$$

$$+ e^t \cdot \left\| \nabla \log p_{k,S-(r\eta+t)}(\boldsymbol{x}) - \nabla \log p_{k,S-(r\eta+t)}(e^t\boldsymbol{x}) \right\|$$

$$= (e^t - 1) \cdot \left\| \nabla \log p_{k,S-(r\eta+t)}(\boldsymbol{x}) \right\| + e^t \cdot (e^t - 1) L \left\| \boldsymbol{x} \right\|.$$

To upper bound the latter term, we expect to employ Lemma D.1. However, it requires a specific condition which denotes the smoothness of $-\nabla \log p'_{k,S-(r\eta+t)}$ should be upper bounded with the variance of $\varphi_{(1-e^{-2t})}$ as

$$\left\| -\nabla^2 \log p'_{k,S-(r\eta+t)} \right\| \leq \frac{1}{2(1-e^{-2t})},$$

which can be achieved by setting

$$\eta \leq \min \left\{ \frac{1}{4L}, \frac{1}{2} \right\}.$$

Since the smoothness of $-\nabla \log p_{k,S-(r\eta+t)}$, i.e., Assumption $[\mathbf{A}_1]$, implies $-\nabla \log p'_{k,S-(r\eta+t)}$ is $e^{2t}L$-smooth. Besides, there are

$$t \leq \eta \leq \min \left\{ \frac{1}{4L}, \frac{1}{2} \right\} \leq \log \left( 1 + \frac{1}{2L} \right) \quad \text{and} \quad e^{2t}L \leq \frac{1}{2(1-e^{-2t})}.$$

Therefore, we have

$$\left\| \nabla \log p'_{k,S-(r\eta+t)}(\boldsymbol{x}) - \nabla \log \left( p'_{k,S-(r\eta+t)} * \varphi_{(1-e^{-2t})} \right)(\boldsymbol{x}) \right\|$$

$$\leq 6e^{2t}L\sqrt{1-e^{-2t}}d^{1/2} + 2e^{3t}L(1-e^{-2t}) \left\| \nabla \log p_{k,S-(r\eta+t)}(e^t\boldsymbol{x}) \right\|$$

$$\leq 6e^{2t}L\sqrt{1-e^{-2t}}d^{1/2} + 2L \cdot e^t(e^{2t}-1) \left\| \nabla \log p_{k,S-(r\eta+t)}(\boldsymbol{x}) \right\| \tag{17}$$

$$+ 2L \cdot e^t(e^{2t}-1) \left\| \nabla \log p_{k,S-(r\eta+t)}(e^t\boldsymbol{x}) - \nabla \log p_{k,S-(r\eta+t)}(\boldsymbol{x}) \right\|$$

$$\leq 6e^{2t}L\sqrt{1-e^{-2t}}d^{1/2} + 2L \cdot e^t(e^{2t}-1) \left\| \nabla \log p_{k,S-(r\eta+t)}(\boldsymbol{x}) \right\|$$

$$+ 2L^2 \cdot e^t(e^{2t}-1)(e^t-1) \left\| \boldsymbol{x} \right\|,$$

where the first inequality follows from Lemma D.1, the last inequality follows from Assumption $[\mathbf{A}_1]$. Due to the range, i.e., $\eta \leq 1/2$, we have the following inequalities

$$e^{2t} \leq e^{2\eta} \leq 1 + 4\eta \leq 3, \quad 1 - e^{-2t} \leq 2t \leq 2\eta \quad \text{and} \quad e^t \leq e^\eta \leq 1 + \frac{3}{2} \cdot \eta.$$

In this condition, Eq 16 can be reformulated as

$$\left\| \nabla \log \frac{p_{k,S-(r\eta+t)}(\boldsymbol{x})}{p'_{k,S-(r\eta+t)}(\boldsymbol{x})} \right\|^2 \leq 2 \left[ (e^t - 1)^2 \cdot \left\| \nabla \log p_{k,S-(r\eta+t)}(\boldsymbol{x}) \right\|^2 + e^{2t} \cdot (e^t - 1)^2 L^2 \left\| \boldsymbol{x} \right\|^2 \right]$$

$$\leq 5\eta^2 \left\| \nabla \log p_{k,S-(r\eta+t)}(\boldsymbol{x}) \right\|^2 + 14L^2\eta^2 \left\| \boldsymbol{x} \right\|^2,$$

and Eq 17 implies

$$\left\| \nabla \log p'_{k,S-(r\eta+t)}(\boldsymbol{x}) - \nabla \log \left( p'_{k,S-(r\eta+t)} * \phi_{(1-e^{-2t})} \right)(\boldsymbol{x}) \right\|^2$$

$$\leq 3 \cdot \left[ 6^2 e^{4t} L^2 (1-e^{-2t}) d + 4 L^2 e^{2t} (e^{2t}-1)^2 \left\| \nabla \log p_{k,S-(r\eta+t)}(\boldsymbol{x}) \right\|^2 + 4 L^4 e^{2t} (e^{2t}-1)^2 (e^t-1)^2 \left\| \boldsymbol{x} \right\|^2 \right]$$

$$\leq 3 \cdot \left[ 2^3 \cdot 3^4 L^2 \eta d + 2^6 \cdot 3 L^2 \eta^2 \left\| \nabla \log p_{k,S-(r\eta+t)}(\boldsymbol{x}) \right\|^2 + 3^3 \cdot 2^4 L^4 \eta^4 \left\| \boldsymbol{x} \right\|^2 \right]$$

$$\leq 2^3 \cdot 3^5 L^2 \eta d + 2^6 \cdot 3^2 L^2 \eta^2 \left\| \nabla \log p_{k,S-(r\eta+t)}(\boldsymbol{x}) \right\|^2 + 3^4 \cdot L^2 \eta^2 \left\| \boldsymbol{x} \right\|^2,$$

where the last inequality follows from $\eta L \leq 1/4$. Hence, suppose $L \geq 1$ without loss of generality, we have

$$\text{Term 2.1} \leq 2 \cdot \left( \mathbb{E} \left[ \left\| \nabla \log \frac{p_{k,S-(r\eta+t)}(\hat{\mathbf{x}}_{k,r\eta})}{p'_{k,S-(r\eta+t)}(\hat{\mathbf{x}}_{k,r\eta})} \right\|^2 \right] + \mathbb{E} \left[ \left\| \nabla \log \frac{p'_{k,S-(r\eta+t)}(\hat{\mathbf{x}}_{k,r\eta})}{p_{k,S-r\eta}(\hat{\mathbf{x}}_{k,r\eta})} \right\|^2 \right] \right)$$

$$\leq 2^4 \cdot 3^5 L^2 \eta d + 2^8 \cdot 3^2 L^2 \eta^2 \mathbb{E} \left[ \left\| \nabla \log p_{k,S-(r\eta+t)}(\hat{\mathbf{x}}_{k,r\eta}) \right\|^2 \right] + 2^2 \cdot 3^4 L^2 \eta^2 \mathbb{E} \left[ \left\| \hat{\mathbf{x}}_{k,r\eta} \right\|^2 \right]$$

$$\leq 2^{14} L^2 \eta d + 2^{13} L^2 \eta^2 \mathbb{E} \left[ \left\| \nabla \log p_{k,S-(r\eta+t)}(\hat{\mathbf{x}}_{k,r\eta+t}) \right\|^2 \right] + 2^{13} L^4 \eta^2 \mathbb{E} \left[ \left\| \hat{\mathbf{x}}_{k,r\eta+t} - \hat{\mathbf{x}}_{k,r\eta} \right\|^2 \right]$$

$$+ 2^{10} L^2 \eta^2 \mathbb{E} \left[ \left\| \hat{\mathbf{x}}_{k,r\eta} \right\|^2 \right].$$

Therefore, we have

$$\text{Term 2} \leq 2^{14} L^2 \eta d + 2^{10} L^2 \eta^2 \mathbb{E} \left[ \left\| \hat{\mathbf{x}}_{k,r\eta} \right\|^2 \right] + 2^{13} L^2 \eta^2 \mathbb{E} \left[ \left\| \nabla \log p_{k,S-(r\eta+t)}(\hat{\mathbf{x}}_{k,r\eta+t}) \right\|^2 \right]$$

$$+ \left( 2^{13} L^2 \eta^2 + 1 \right) L^2 \mathbb{E} \left[ \left\| \hat{\mathbf{x}}_{k,r\eta+t} - \hat{\mathbf{x}}_{k,r\eta} \right\|^2 \right]$$

$$\leq 2^{14} L^2 \eta d + 2^{10} L^2 \eta^2 (M+d) + 2^{13} L^3 \eta^2 d + 2^{10} L^2 \left( 2(M+d)\eta^2 + 4d\eta \right)$$

where the last inequality follows from Lemma D.2 and Lemma D.3. To diminish the discretization error, we require the step size of backward sampling, i.e., $\eta$ satisfies

$$\begin{cases} 2^{14} L^2 \eta d \leq \epsilon \\ 2^{10} \cdot L^2 \eta^2 (d+M) \leq \epsilon \\ 2^{13} \cdot L^3 \eta^2 d \leq \epsilon \\ 2^{10} \cdot L^2 \left( 2(M+d)\eta^2 + 4d\eta \right) \leq \epsilon \end{cases} \Leftarrow \begin{cases} \eta \leq 2^{-14} L^{-2} d^{-1} \epsilon \\ \eta \leq 2^{-5} \cdot L^{-1} (d+M)^{-0.5} \epsilon^{0.5} \\ \eta \leq 2^{-6.5} \cdot L^{-1.5} d^{-0.5} \epsilon^{0.5} \\ \eta \leq 2^{-6} L^{-0.5} (d+M)^{-0.5} \epsilon^{0.5} \\ \eta \leq 2^{-13} L^{-2} d^{-1} \epsilon. \end{cases}$$

Specifically, if we choose

$$\eta \leq 2^{-14} L^{-2} (d+M)^{-1} \epsilon = C_\eta (d+M)^{-1} \epsilon,$$

we have

$$\mathbb{E} \left[ L^2 \left\| \hat{\mathbf{x}}_{k,t+r\eta} - \hat{\mathbf{x}}_{k,r\eta} \right\|^2 \right] + \mathbb{E} \left[ \left\| \nabla \log \frac{p_{k,S-r\eta}(\hat{\mathbf{x}}_{k,r\eta})}{p_{k,S-(t+r\eta)}(\hat{\mathbf{x}}_{k,r\eta})} \right\|^2 \right] \leq 4\epsilon,$$

and the proof is completed. $\qquad\square$

## E    LEMMAS FOR BOUNDING SCORE ESTIMATION ERROR

**Lemma E.1** (Recursive Form of Score Functions). *Under the notation in Section A, for any $k \in \mathbb{N}_{0,K-1}$ and $t \in [0, S]$, the score function can be written as*

$$\nabla_{\boldsymbol{x}} \log p_{k,S-t}(\boldsymbol{x}) = \mathbb{E}_{\mathbf{x}' \sim q_{k,S-t}(\cdot|\boldsymbol{x})} \left[ -\frac{\boldsymbol{x} - e^{-(S-t)} \mathbf{x}'}{\left( 1 - e^{-2(S-t)} \right)} \right]$$

*where the conditional density function $q_{k,S-t}(\cdot|\boldsymbol{x})$ is defined as*

$$q_{k,S-t}(\boldsymbol{x}'|\boldsymbol{x}) \propto \exp \left( \nabla \log p_{k,0}(\boldsymbol{x}') - \frac{\left\| \boldsymbol{x} - e^{-(S-t)} \boldsymbol{x}' \right\|^2}{2 \left( 1 - e^{-2(S-t)} \right)} \right).$$

*Proof.* When the OU process, i.e., SDE 1, is selected as the forward path, for any $k \in \mathbb{N}_{0,K}$ and $t \in [0, S]$, the transition kernel has a closed form, i.e.,

$$p_{k,t|0}(\boldsymbol{x}|\boldsymbol{x}_0) = \left(2\pi\left(1 - e^{-2t}\right)\right)^{-d/2} \cdot \exp\left[\frac{-\left\|\boldsymbol{x} - e^{-t}\boldsymbol{x}_0\right\|^2}{2\left(1 - e^{-2t}\right)}\right], \quad \forall\, 0 \le t \le S.$$

In this condition, we have

$$p_{k,S-t}(\boldsymbol{x}) = \int_{\mathbb{R}^d} p_{k,0}(\boldsymbol{x}_0) \cdot p_{k,S-t|0}(\boldsymbol{x}|\boldsymbol{x}_0)\mathrm{d}\boldsymbol{x}_0$$

$$= \int_{\mathbb{R}^d} p_{k,0}(\boldsymbol{x}_0) \cdot \left(2\pi\left(1 - e^{-2(S-t)}\right)\right)^{-d/2} \cdot \exp\left[\frac{-\left\|\boldsymbol{x} - e^{-(S-t)}\boldsymbol{x}_0\right\|^2}{2\left(1 - e^{-2(S-t)}\right)}\right]\mathrm{d}\boldsymbol{x}_0$$

Plugging this formulation into the following equation

$$\nabla_{\boldsymbol{x}} \log p_{k,S-t}(\boldsymbol{x}) = \frac{\nabla p_{k,S-t}(\boldsymbol{x})}{p_{k,S-t}(\boldsymbol{x})},$$

we have

$$\nabla_{\boldsymbol{x}} \log p_{k,S-t}(\boldsymbol{x}) = \frac{\nabla \int_{\mathbb{R}^d} p_{k,0}(\boldsymbol{x}_0) \cdot \left(2\pi\left(1 - e^{-2(S-t)}\right)\right)^{-d/2} \cdot \exp\left[\frac{-\left\|\boldsymbol{x} - e^{-(S-t)}\boldsymbol{x}_0\right\|^2}{2\left(1 - e^{-2(S-t)}\right)}\right]\mathrm{d}\boldsymbol{x}_0}{\int_{\mathbb{R}^d} p_{k,0}(\boldsymbol{x}_0) \cdot \left(2\pi\left(1 - e^{-2(S-t)}\right)\right)^{-d/2} \cdot \exp\left[\frac{-\left\|\boldsymbol{x} - e^{-(S-t)}\boldsymbol{x}_0\right\|^2}{2\left(1 - e^{-2(S-t)}\right)}\right]\mathrm{d}\boldsymbol{x}_0}$$

$$= \frac{\int_{\mathbb{R}^d} p_{k,0}(\boldsymbol{x}_0) \cdot \exp\left(\frac{-\left\|\boldsymbol{x} - e^{-(S-t)}\boldsymbol{x}_0\right\|^2}{2\left(1 - e^{-2(S-t)}\right)}\right) \cdot \left(-\frac{\boldsymbol{x} - e^{-(S-t)}\boldsymbol{x}_0}{\left(1 - e^{-2(T-t)}\right)}\right)\mathrm{d}\boldsymbol{x}_0}{\int_{\mathbb{R}^d} p_{k,0}(\boldsymbol{x}_0) \cdot \exp\left(\frac{-\left\|\boldsymbol{x} - e^{-(S-t)}\boldsymbol{x}_0\right\|^2}{2\left(1 - e^{-2(S-t)}\right)}\right)\mathrm{d}\boldsymbol{x}_0}$$

$$= \mathbb{E}_{\mathbf{x}_0 \sim q_{k,S-t}(\cdot|\boldsymbol{x})}\left[-\frac{\boldsymbol{x} - e^{-(S-t)}\mathbf{x}_0}{\left(1 - e^{-2(S-t)}\right)}\right]$$

$$\tag{18}$$

where the density function $q_{T-t}(\cdot|\boldsymbol{x})$ is defined as

$$q_{k,S-t}(\boldsymbol{x}_0|\boldsymbol{x}) = \frac{p_{k,0}(\boldsymbol{x}_0) \cdot \exp\left(\frac{-\left\|\boldsymbol{x} - e^{-(S-t)}\boldsymbol{x}_0\right\|^2}{2\left(1 - e^{-2(S-t)}\right)}\right)}{\int_{\mathbb{R}^d} p_{k,0}(\boldsymbol{x}_0) \cdot \exp\left(\frac{-\left\|\boldsymbol{x} - e^{-(S-t)}\boldsymbol{x}_0\right\|^2}{2\left(1 - e^{-2(S-t)}\right)}\right)\mathrm{d}\boldsymbol{x}_0}$$

$$\propto \exp\left(-f_{k,0}(\boldsymbol{x}_0) - \frac{\left\|\boldsymbol{x} - e^{-(S-t)}\boldsymbol{x}_0\right\|^2}{2\left(1 - e^{-2(S-t)}\right)}\right),$$

where $p_{k,0} \propto \exp(-f_{k,0})$. Hence, the proof is completed. $\square$

**Lemma E.2** (Strong log-concavity and L-smoothness of the auxiliary targets). *Under the notation in Section A, for any $k \in \mathbb{N}_{0,K-1}$, $r \in \mathbb{N}_{0,R-1}$ and $\boldsymbol{x} \in \mathbb{R}^d$, we define the auxiliary target distribution as*

$$q_{k,S-r\eta}(\boldsymbol{x}'|\boldsymbol{x}) \propto \exp\left(\nabla \log p_{k,0}(\boldsymbol{x}') - \frac{\left\|\boldsymbol{x} - e^{-(S-r\eta)}\boldsymbol{x}'\right\|^2}{2\left(1 - e^{-2(S-r\eta)}\right)}\right).$$

*We define*

$$\mu_r := \frac{1}{2} \cdot \frac{e^{-2(S-r\eta)}}{1 - e^{-2(S-r\eta)}} \quad \text{and} \quad L_r := \frac{3}{2} \cdot \frac{e^{-2(S-r\eta)}}{1 - e^{-2(S-r\eta)}}.$$

*Then, we have*

$$\mu_r \boldsymbol{I} \preceq -\nabla^2 \log q_{k,S-r\eta}(\boldsymbol{x}'|\boldsymbol{x}) \preceq L_r \boldsymbol{I}$$

*when the segment length $S$ satisfies $S = \frac{1}{2}\log\left(\frac{2L+1}{2L}\right)$.*

*Proof.* We begin with the formulation of $\nabla^2 \log q_{k,S-t}$, i.e.,

$$-\nabla^2 \log q_{k,S-r\eta}(\boldsymbol{x}'|\boldsymbol{x}) = -\nabla^2 \log p_{k,0}(\boldsymbol{x}') + \frac{e^{-2(S-r\eta)}}{1 - e^{-2(S-r\eta)}} \boldsymbol{I}. \tag{19}$$

By supposing $S = \frac{1}{2} \log \left( \frac{2L+1}{2L} \right)$, we have

$$\frac{e^{-2(S-r\eta)}}{1 - e^{-2(S-r\eta)}} \geq \frac{e^{-2S}}{1 - e^{-2S}} = 2L \geq 2 \left\| \nabla^2 \log p_{k,0} \right\|.$$

Plugging this inequality into Eq 19, we have

$$- \nabla^2 p_{k,0}(\boldsymbol{x}') + \frac{e^{-2(S-r\eta)}}{1 - e^{-2(S-r\eta)}} \cdot \boldsymbol{I} \preceq \left( \left\| \nabla^2 \log p_{k,0}(\boldsymbol{x}') \right\| + \frac{e^{-2(S-r\eta)}}{1 - e^{-2(S-r\eta)}} \right) \cdot \boldsymbol{I}$$

$$\preceq \frac{3}{2} \cdot \frac{e^{-2(S-r\eta)}}{1 - e^{-2(S-r\eta)}} \cdot \boldsymbol{I} = L_r \boldsymbol{I}.$$

Besides, it has

$$- \nabla^2 p_{k,0}(\boldsymbol{x}') + \frac{e^{-2(S-r\eta)}}{1 - e^{-2(S-r\eta)}} \cdot \boldsymbol{I} \succeq \left( - \left\| \nabla^2 \log p_{k,0}(\boldsymbol{x}') \right\| + \frac{e^{-2(S-r\eta)}}{1 - e^{-2(S-r\eta)}} \right) \cdot \boldsymbol{I}$$

$$\succeq \frac{1}{2} \cdot \frac{e^{-2(S-r\eta)}}{1 - e^{-2(S-r\eta)}} \cdot \boldsymbol{I} = \mu_r \boldsymbol{I}.$$

Hence, the proof is completed. $\qquad\square$

### E.1 Score Estimation Error from Empirical Mean

**Lemma E.3.** *With a little abuse of notation, for each $i \in \mathbb{N}_{1,n_{k,r}}$ in Alg 1, we denote the underlying distribution of **output particles** as $\mathbf{x}'_i \sim q'_{k,S-r\eta}$ and suppose it satisfies LSI with the constant $\mu'_r$. Then, for any $\boldsymbol{x} \in \mathbb{R}^d$, we have*

$$\mathbb{P} \left[ \left\| -\frac{1}{n_{k,r}} \sum_{i=1}^{n_{k,r}} \mathbf{x}'_i + \mathbb{E}_{\mathbf{x}' \sim q'_{k,S-r\eta}(\cdot|\boldsymbol{x})} \left[ \mathbf{x}' \right] \right\| \leq 2\epsilon' \right] \geq 1 - \delta$$

*by requiring the sample number $n_{k,r}$ to satisfy*

$$n_{k,r} \geq \frac{\max \left\{ d, -2 \log \delta \right\}}{\mu'_r \epsilon'^2}.$$

*Proof.* For any $\boldsymbol{x} \in \mathbb{R}^d$, we set

$$\boldsymbol{b}' := \mathbb{E}_{q'_{k,S-r\eta}(\cdot|\boldsymbol{x})} \left[ \mathbf{x}' \right] \quad \text{and} \quad \sigma' := \mathbb{E}_{\{\mathbf{x}'_i\}_{i=1}^{n_{k,r}} \sim q'^{(n_{k,r})}_{k,S-r\eta}(\cdot|\boldsymbol{x})} \left[ \left\| \sum_{i=1}^{n_{k,r}} \mathbf{x}'_i - \mathbb{E} \left[ \sum_{i=1}^{n_{k,r}} \mathbf{x}'_i \right] \right\| \right].$$

We begin with the following probability

$$\mathbb{P}_{\{\mathbf{x}'_i\}_{i=1}^{n_{k,r}} \sim q'^{(n_{k,r})}_{k,S-r\eta}(\cdot|\boldsymbol{x})} \left[ \left\| -\frac{1}{n_{k,r}} \sum_{i=1}^{n_{k,r}} \mathbf{x}'_i + \mathbb{E}_{\mathbf{x}' \sim q'_{k,S-r\eta}(\cdot|\boldsymbol{x})} \left[ \mathbf{x}' \right] \right\|^2 \geq \left( \frac{\sigma'}{n_{k,r}} + \epsilon' \right)^2 \right]$$

$$= \mathbb{P}_{\{\mathbf{x}'_i\}_{i=1}^{n_{k,r}} \sim q'^{(n_{k,r})}_{k,S-r\eta}(\cdot|\boldsymbol{x})} \left[ \left\| \sum_{i=1}^{n_{k,r}} \mathbf{x}'_i - n_{k,r} \boldsymbol{b}' \right\| \geq \sigma' + n_{k,r} \epsilon' \right] \tag{20}$$

To lower bound this probability, we expect to utilize Lemma F.9 which requires the following two conditions:

- The distribution of $\sum_{i=1}^{n_{k,r}} \mathbf{x}'_i$ satisfies LSI, and its LSI constant can be obtained.

- The formulation $\left\| \sum_{i=1}^{n_{k,r}} \mathbf{x}'_i - n_{k,r} \boldsymbol{b}' \right\| \geq \sigma' + n_{k,r} \epsilon'$ can be presented as $F \geq \mathbb{E}[F] + \text{bias}$ where $F$ is a 1-Lipschitz function.

For the first condition, by employing Lemma F.5, we have that the LSI constant of

$$\sum_{i=1}^{n_{k,r}} \mathbf{x}'_i \sim \underbrace{q'_{k,S-r\eta}(\cdot|\boldsymbol{x}) * q'_{k,S-r\eta}(\cdot|\boldsymbol{x}) \cdots * q'_{k,S-r\eta}(\cdot|\boldsymbol{x})}_{n_{k,r}}$$

is $\mu'_r/n_{k,r}$. For the second condition, we set the function $F(\boldsymbol{x}) = \|\boldsymbol{x} - n_{k,r}\boldsymbol{b}'\| : \mathbb{R}^d \to \mathbb{R}$ is 1-Lipschitz because

$$\|F\|_{\mathrm{Lip}} = \sup_{\boldsymbol{x} \neq \boldsymbol{y}} \frac{|F(\boldsymbol{x}) - F(\boldsymbol{y})|}{\|\boldsymbol{x} - \boldsymbol{y}\|} = \sup_{\boldsymbol{x} \neq \boldsymbol{y}} \frac{|\|\boldsymbol{x}\| - \|\boldsymbol{y}\||}{\|(\boldsymbol{x} - \boldsymbol{y})\|} = 1.$$

Besides, we have

$$F\left(\sum_{i=1}^{n_{k,r}} \mathbf{x}'_i\right) = \left\|\sum_{i=1}^{n_{k,r}} \mathbf{x}'_i - n_{k,r}\boldsymbol{b}'\right\| \quad \text{and} \quad \mathbb{E}\left[F\left(\sum_{i=1}^{n_{k,r}} \mathbf{x}'_i\right)\right] = \sigma'$$

where the second equation follows from the definition of $\sigma'$. Therefore, with Lemma F.9, we have

$$\mathbb{P}_{\{\mathbf{x}'_i\}_{i=1}^{n_{k,r}} \sim q'^{(n_{k,r})}_{k,S-r\eta}(\cdot|\boldsymbol{x})} \left[\left\|\sum_{i=1}^{n_{k,r}} \mathbf{x}'_i - n_{k,r}\boldsymbol{b}'\right\| \geq \sigma' + n_{k,r}\epsilon'\right] \leq \exp\left(-\frac{\mu'_r \epsilon'^2 n_{k,r}}{2}\right). \quad (21)$$

Then, we consider the range of $\sigma'$ and have

$$\begin{aligned}
\sigma' &= n_{k,r} \cdot \mathbb{E}_{\{\mathbf{x}'_i\}_{i=1}^{n_{k,r}} \sim q'^{(n_{k,r})}_{k,S-r\eta}(\cdot|\boldsymbol{x})} \left\|\frac{1}{n_{k,r}}\sum_{i=1}^{n_{k,r}} \mathbf{x}'_i - \boldsymbol{b}'\right\| \\
&\leq n_{k,r} \cdot \sqrt{\mathrm{var}\left(\frac{1}{n_{k,r}}\sum_{i=1}^{n_{k,r}} \mathbf{x}'_i\right)} = \sqrt{n_{k,r}\mathrm{var}(\mathbf{x}'_i)} \leq \sqrt{\frac{n_{k,r}d}{\mu'_r}},
\end{aligned} \quad (22)$$

the first inequality follows from Holder's inequality and the last follows from Lemma F.11. Combining Eq 21 and Eq 22, it has

$$\mathbb{P}_{\{\mathbf{x}'_i\}_{i=1}^{n_{k,r}} \sim q'^{(n_{k,r})}_{k,S-r\eta}(\cdot|\boldsymbol{x})} \left[\left\|-\frac{1}{n_{k,r}}\sum_{i=1}^{n_{k,r}} \mathbf{x}'_i + \mathbb{E}_{\mathbf{x}' \sim q'_{k,S-r\eta}(\cdot|\boldsymbol{x})}[\mathbf{x}']\right\|^2 \geq \left(\sqrt{\frac{d}{\mu'_r n_{k,r}}} + \epsilon'\right)^2\right] \leq \exp\left(-\frac{\mu'_r \epsilon'^2 n_{k,r}}{2}\right).$$

By requiring

$$\frac{d}{\mu'_r n_{k,r}} \leq \epsilon'^2 \quad \text{and} \quad -\frac{\mu'_r \epsilon'^2 n_{k,r}}{2} \leq \log \delta, \quad (23)$$

we have

$$\begin{aligned}
&\mathbb{P}\left[\left\|-\frac{1}{n_{k,r}}\sum_{i=1}^{n_{k,r}} \mathbf{x}'_i + \mathbb{E}_{\mathbf{x}' \sim q'_{k,S-r\eta}(\cdot|\boldsymbol{x})}[\mathbf{x}']\right\| \leq 2\epsilon'\right] \\
&= 1 - \mathbb{P}\left[\left\|-\frac{1}{n_{k,r}}\sum_{i=1}^{n_{k,r}} \mathbf{x}'_i + \mathbb{E}_{\mathbf{x}' \sim q'_{k,S-r\eta}(\cdot|\boldsymbol{x})}[\mathbf{x}']\right\| \geq 2\epsilon'\right] \geq 1 - \delta.
\end{aligned}$$

Noted that Eq. 23 implies the sample number $n_{k,r}$ should satisfy

$$n_{k,r} \geq \frac{d}{\mu'_r \epsilon'^2} \quad \text{and} \quad n_{k,r} \geq \frac{2\log \delta^{-1}}{\mu'_r \epsilon'^2}.$$

Hence, the proof is completed. $\qquad\square$

### E.2 Score Estimation Error from Mean Gap

**Lemma E.4.** *For any given $(k, r, \boldsymbol{x})$ in Alg 1, suppose the distribution $q_{k,S-r\eta}(\cdot|\boldsymbol{x})$ satisfies*

$$\mu_r \boldsymbol{I} \preceq -\nabla^2 \log q_{k,S-r\eta}(\cdot|\boldsymbol{x}) \preceq L_r \boldsymbol{I},$$

*and $\mathbf{x}'_j \sim q'_j(\cdot|\boldsymbol{x})$ corresponds to Line 9 of Alg 1. If $0 < \tau_r \leq \mu_r/(8L_r^2)$, we have*

$$\mathrm{KL}\left(q'_{j+1}(\cdot|\boldsymbol{x})\|q_{k,S-r\eta}(\cdot|\boldsymbol{x})\right) \leq e^{-\mu_r \tau_r}\mathrm{KL}\left(q'_j(\cdot|\boldsymbol{x})\|q_{k,S-r\eta}(\cdot|\boldsymbol{x})\right) + 28L_r^2 d\tau_r^2$$

*when the score estimation satisfies $\|\nabla \log p_{k,0} - \boldsymbol{v}'\|_\infty \leq L_r\sqrt{2d\tau_r}$.*

*Proof.* Suppose the loop in Line 6 of Alg 1 aims to draw a sample from the target distribution $q_{k,S-r\eta}(\cdot|\boldsymbol{x})$ satisfying

$$q_{k,S-r\eta}(\boldsymbol{x}'|\boldsymbol{x}) \propto \exp(-g_{k,r}(\boldsymbol{x}')) := \exp\left(-f_{k,0}(\boldsymbol{x}') - \frac{\left\|\boldsymbol{x} - e^{-(S-r\eta)}\boldsymbol{x}'\right\|^2}{2(1 - e^{-2(S-r\eta)})}\right).$$

The score function of the target, i.e., $\nabla g_{k,r}(\boldsymbol{x}')$, satisfies

$$\nabla g_{k,r}(\boldsymbol{x}') = \nabla f_{k,0}(\boldsymbol{x}') + \frac{-e^{-(S-r\eta)}\boldsymbol{x} + e^{-2(S-r\eta)}\boldsymbol{x}'}{1 - e^{-2(S-r\eta)}}.$$

At the $j$-th iteration corresponding to Line 9 in Alg 1. The previous score is approximated by

$$\nabla g'(\boldsymbol{x}') = \boldsymbol{v}'(\boldsymbol{x}') + \frac{-e^{-(S-r\eta)}\boldsymbol{x} + e^{-2(S-r\eta)}\boldsymbol{x}'}{1 - e^{-2(S-r\eta)}}.$$

where $\boldsymbol{v}'(\cdot)$ is used to approximate $\nabla \log p_{k,0}(\cdot)$ by calling Alg 1 recursively. Suppose $\mathbf{x}'_j = \boldsymbol{z}_0$, the $j$-th iteration is equivalent to the following SDE

$$\mathrm{d}\mathbf{z}_t = -\nabla g'(\boldsymbol{z}_0)\mathrm{d}t + \sqrt{2}\mathrm{d}B_t,$$

we denote the underlying distribution of $\mathbf{z}_t$ as $q_t$. Similarly, we set $q_{0t}$ as the joint distribution of $(\mathbf{z}_0, \mathbf{z}_t)$, and have

$$q_{0t}(\boldsymbol{z}_0, \boldsymbol{z}_t) = q_0(\boldsymbol{z}_0) \cdot q_{t|0}(\boldsymbol{z}_t|\boldsymbol{z}_0).$$

According to the Fokker-Planck equation, we have

$$\partial_t q_{t|0}(\boldsymbol{z}_t|\boldsymbol{z}_0) = \nabla \cdot \left(q_{t|0}(\boldsymbol{z}_t|\boldsymbol{z}_0) \cdot \nabla g'(\boldsymbol{z}_0)\right) + \Delta q_{t|0}(\boldsymbol{z}_t|\boldsymbol{z}_0)$$

In this condition, we have

$$
\begin{aligned}
\partial_t q_t(\boldsymbol{z}_t) &= \int \frac{\partial q_{t|0}(\boldsymbol{z}_t|\boldsymbol{z}_0)}{\partial t} \cdot q_0(\boldsymbol{z}_0)\mathrm{d}\boldsymbol{z}_0 \\
&= \int \left[\nabla \cdot \left(q_{t|0}(\boldsymbol{z}_t|\boldsymbol{z}_0) \cdot \nabla g'(\boldsymbol{z}_0)\right) + \Delta q_{t|0}(\boldsymbol{z}_t|\boldsymbol{z}_0)\right] \cdot q_0(\boldsymbol{z}_0)\mathrm{d}\boldsymbol{z}_0 \\
&= \nabla \cdot \left(q_t(\boldsymbol{z}_t) \int q_{0|t}(\boldsymbol{z}_0|\boldsymbol{z}_t)\nabla g'(\boldsymbol{z}_0)\mathrm{d}\boldsymbol{z}_0\right) + \Delta q_t(\boldsymbol{z}_t).
\end{aligned}
$$

For abbreviation, we suppose

$$q_*(\cdot) := q_{k,S-r\eta}(\cdot|\boldsymbol{x}) \quad \text{and} \quad g_* := g_{k,r}.$$

With these notations, the dynamic of the KL divergence between $q_t$ and $q_*$ is

$$
\begin{aligned}
\partial_t \mathrm{KL}\left(q_t\|q_*\right) &= \int \partial_t q_t(\boldsymbol{z}_t) \log \frac{q_t(\boldsymbol{z}_t)}{q_*(\boldsymbol{z}_t)}\mathrm{d}\boldsymbol{z}_t \\
&= \int \nabla \cdot \left[q_t(\boldsymbol{z}_t)\left(\int q_{0|t}(\boldsymbol{z}_0|\boldsymbol{z}_t)\nabla g'(\boldsymbol{z}_0)\mathrm{d}\boldsymbol{z}_0 + \nabla \log q_t(\boldsymbol{z}_t)\right)\right] \cdot \log \frac{q_t(\boldsymbol{z}_t)}{q_*(\boldsymbol{z}_t)}\mathrm{d}\boldsymbol{z}_t \\
&= -\int q_t(\boldsymbol{z}_t)\left(\left\|\nabla \log \frac{q_t(\boldsymbol{z}_t)}{q_*(\boldsymbol{z}_t)}\right\|^2 + \left\langle \int q_{0|t}(\boldsymbol{z}_0|\boldsymbol{z}_t)\nabla g'(\boldsymbol{z}_0)\mathrm{d}\boldsymbol{z}_0 + \nabla \log q_*(\boldsymbol{z}_t), \nabla \log \frac{q_t(\boldsymbol{z}_t)}{q_*(\boldsymbol{z}_t)}\right\rangle\right)\mathrm{d}\boldsymbol{z}_t \\
&= -\int q_t(\boldsymbol{z}_t)\left\|\nabla \log \frac{q_t(\boldsymbol{z}_t)}{q_*(\boldsymbol{z}_t)}\right\|^2 \mathrm{d}\boldsymbol{z}_t + \int q_{0t}(\boldsymbol{z}_0, \boldsymbol{z}_t)\left\langle \nabla g'(\boldsymbol{z}_0) - \nabla g_*(\boldsymbol{z}_t), \nabla \log \frac{q_t(\boldsymbol{z}_t)}{q_*(\boldsymbol{z}_t)}\right\rangle \mathrm{d}(\boldsymbol{z}_0, \boldsymbol{z}_t) \\
&\le -\frac{3}{4}\int q_t(\boldsymbol{z}_t)\left\|\nabla \log \frac{q_t(\boldsymbol{z}_t)}{q_*(\boldsymbol{z}_t)}\right\|^2 \mathrm{d}\boldsymbol{z}_t + \int q_{0t}(\boldsymbol{z}_0, \boldsymbol{z}_t)\left\|\nabla g'(\boldsymbol{z}_0) - \nabla g_*(\boldsymbol{z}_t)\right\|^2 \mathrm{d}(\boldsymbol{z}_0, \boldsymbol{z}_t) \\
&\le -\frac{3}{4}\int q_t(\boldsymbol{z}_t)\left\|\nabla \log \frac{q_t(\boldsymbol{z}_t)}{q_*(\boldsymbol{z}_t)}\right\|^2 + 2\int q_{0t}(\boldsymbol{z}_0, \boldsymbol{z}_t)\left\|\nabla g'(\boldsymbol{z}_0) - \nabla g_*(\boldsymbol{z}_0)\right\|^2 \mathrm{d}(\boldsymbol{z}_0, \boldsymbol{z}_t) \\
&\quad + 2\int q_{0t}(\boldsymbol{z}_0, \boldsymbol{z}_t)\left\|\nabla g_*(\boldsymbol{z}_0) - \nabla g_*(\boldsymbol{z}_t)\right\|^2 \mathrm{d}(\boldsymbol{z}_0, \boldsymbol{z}_t).
\end{aligned}
$$

$$(24)$$

**Upper bound the first term in Eq 24.** The target distribution $q_*$ satisfies $\mu_r$-strong convexity, i.e.,

$$\mu_r \boldsymbol{I} \preceq -\nabla^2 \log q_{k,S-r\eta}(\boldsymbol{x}'|\boldsymbol{x}) = -\nabla^2 \log(q_*(\boldsymbol{x}')),$$

It means $q_*$ satisfies LSI with the constant $\mu_r$ due to Lemma F.4. Hence, we have

$$-\frac{3}{4}\int q_t(\boldsymbol{z}_t) \left\|\nabla \log \frac{q_t(\boldsymbol{z}_t)}{q_*(\boldsymbol{z}_t)}\right\|^2 \leq -\frac{3\mu_r}{2}\mathrm{KL}\left(q_t\|q_*\right). \tag{25}$$

**Upper bound the second term in Eq 24.** We assume that there is a uniform upper bound $\epsilon_g$ satisfying

$$\|\nabla g'(\boldsymbol{z}) - \nabla g_*(\boldsymbol{z})\| \leq \epsilon_g \quad \Rightarrow \quad \int q_{0t}(\boldsymbol{z}_0, \boldsymbol{z}_t) \|\nabla g'(\boldsymbol{z}_0) - \nabla g_*(\boldsymbol{z}_0)\|^2 \, \mathrm{d}(\boldsymbol{z}_0, \boldsymbol{z}_t) \leq \epsilon_g^2. \tag{26}$$

**Upper bound the third term in Eq 24.** Due to the monotonicity of $e^{-t}/(1 - e^{-t})$, we have

$$2L \leq \frac{e^{-2(S-r\eta)}}{1 - e^{-2(S-r\eta)}} \leq \frac{e^{-2\eta}}{1 - e^{-2\eta}} \leq \eta^{-1}$$

where we suppose $\eta \leq 1/2$ without loss of the generality to establish the last inequality. Hence, the target distribution $q_*$ satisfies

$$-\nabla^2 \log q_* = -\nabla^2 \log q_{k,S-r\eta}(\cdot|\boldsymbol{x}) = -\nabla^2 \log p_{k,0} + \frac{e^{-2(S-r\eta)}}{1 - e^{-2(S-r\eta)}}$$

$$\preceq \left\|\nabla^2 \log p_{k,0}\right\| \boldsymbol{I} + \frac{e^{-2(S-r\eta)}}{1 - e^{-2(S-r\eta)}}\boldsymbol{I} := L_r \boldsymbol{I} \preceq (L + \eta^{-1})\boldsymbol{I},$$

where the last inequality follows from Assumption [$\mathbf{A}_1$]. This result implies the smoothness of $q_*$, and we have

$$\int q_{0t}(\boldsymbol{z}_0, \boldsymbol{z}_t) \|\nabla g_*(\boldsymbol{z}_0) - \nabla g_*(\boldsymbol{z}_t)\|^2 \, \mathrm{d}(\boldsymbol{z}_0, \boldsymbol{z}_t)$$

$$\leq L_r^2 \int q_{0t}(\boldsymbol{z}_0, \boldsymbol{z}_t) \|\boldsymbol{z}_t - \boldsymbol{z}_0\|^2 \, \mathrm{d}(\boldsymbol{z}_0, \boldsymbol{z}_t) = L_r^2 \cdot \mathbb{E}_{q_{0t}}\left[\left\|-t\nabla g'(\boldsymbol{z}_0) + \sqrt{2t}\xi\right\|^2\right]$$

$$= L_r^2 \cdot \left(2td + t^2\mathbb{E}_{q_0} \|\nabla g'(\boldsymbol{z}_0) - \nabla g_*(\boldsymbol{z}_0) + \nabla g_*(\boldsymbol{z}_0)\|^2\right) \tag{27}$$

$$\leq 2L_r^2 \cdot \left(td + t^2\epsilon_g^2 + t^2\mathbb{E}_{q_0} \|\nabla g_*(\boldsymbol{z}_0)\|^2\right)$$

$$\leq 2L_r^2 dt + 2L_r^2\epsilon_g^2 t^2 + 4L_r^3 dt^2 + \frac{8L_r^4 t^2}{\mu_r}\mathrm{KL}\left(q_0\|q_*\right),$$

where the last inequality follows from Lemma F.12.

Hence, Combining Eq 24, Eq 25, Eq 26, Eq 27 with $t \leq \tau_r \leq 1/(2L_r)$ and $\epsilon_g^2 \leq 2L_r^2 d\tau_r$, we have

$$\partial_t \mathrm{KL}\left(q_t\|q_*\right) \leq -\frac{3\mu_r}{2}\mathrm{KL}\left(q_t\|q_*\right) + 2\epsilon_g^2 + \frac{16L_r^4 t^2}{\mu_r}\mathrm{KL}\left(q_0\|q_*\right) + 4L_r^2 dt + 4L_r^2\epsilon_g^2 t^2 + 8L_r^3 dt^2$$

$$\leq -\frac{3\mu_r}{2}\mathrm{KL}\left(q_t\|q_*\right) + 4L_r^2 d\tau_r + \frac{16L_r^4 \tau_r^2}{\mu_r}\mathrm{KL}\left(q_0\|q_*\right) + 4L_r^2 d\tau_r + 8L_r^4 d\tau_r^3 + 8L_r^3 d\tau_r^2$$

$$\leq -\frac{3\mu_r}{2}\mathrm{KL}\left(q_t\|q_*\right) + \frac{16L_r^4 t^2}{\mu_r}\mathrm{KL}\left(q_0\|q_*\right) + 14L_r^2 d\tau_r.$$

Multiplying both sides by $\exp(\frac{3\mu_r t}{2})$, then the previous inequality can be written as

$$\frac{\mathrm{d}}{\mathrm{d}t}\left(e^{\frac{3\mu_r t}{2}}\mathrm{KL}\left(q_t\|q_*\right)\right) \leq e^{\frac{3\mu_r t}{2}} \cdot \left(\frac{16L_r^4 \tau_r^2}{\mu_r}\mathrm{KL}\left(q_0\|q_*\right) + 14L_r^2 d\tau_r\right).$$

Integrating from $t = 0$ to $t = \tau_r$, we have

$$e^{\frac{3\mu_r \tau_r}{2}}\mathrm{KL}\left(q_t\|q_*\right) - \mathrm{KL}\left(q_0\|q_*\right) \leq \frac{2}{3\mu_r} \cdot \left(e^{\frac{3\mu_r \tau_r}{2}} - 1\right) \cdot \left(\frac{16L_r^4 \tau_r^2}{\mu_r}\mathrm{KL}\left(q_0\|q_*\right) + 14L_r^2 d\tau_r\right)$$

$$\leq 2\tau_r \cdot \left(\frac{16L_r^4 \tau_r^2}{\mu_r}\mathrm{KL}\left(q_0\|q_*\right) + 14L_r^2 d\tau_r\right)$$

where the last inequality establishes due to the fact $e^c \leq 1 + 2c$ when $0 < c \leq \frac{3}{2} \cdot \mu_r \tau_r \leq 1$. It means we have

$$\mathrm{KL}\left(q_t \| q_*\right) \leq e^{-\frac{3\mu_r \tau_r}{2}} \cdot \left(1 + \frac{32 L_r^4 \tau_r^3}{\mu_r}\right) \mathrm{KL}\left(q_0 \| q_*\right) + e^{-\frac{3\mu_r \tau_r}{2}} \cdot 28 L_r^2 d \tau_r^2.$$

By requiring $0 < \tau_r \leq \mu_r/(8L_r^2)$, we have

$$1 + \frac{32 L_r^4 \tau_r^3}{\mu_r} \leq 1 + \frac{\mu_r \tau_r}{2} \leq e^{\frac{\mu_r \tau_r}{2}} \quad \text{and} \quad e^{-\frac{3\mu_r \tau_r}{2}} \leq 1.$$

Hence, there is

$$\mathrm{KL}\left(q_t \| q_*\right) \leq e^{-\mu_r \tau_r} \mathrm{KL}\left(q_0 \| q_*\right) + 28 L_r^2 d \tau_r^2, \tag{28}$$

and the proof is completed. $\qquad \square$

**Lemma E.5.** *In Alg 1, suppose the input is $(k, r, \boldsymbol{x}, \epsilon)$ and $k > 0$, if we choose the initial distribution of the inner loop to be*

$$q_0'(\boldsymbol{x}') \propto \exp\left(-\frac{\left\|\boldsymbol{x} - e^{-(S-r\eta)}\boldsymbol{x}'\right\|^2}{2(1 - e^{-2(S-r\eta)})}\right),$$

*then suppose $q_{k,S-r\eta}(\cdot|\boldsymbol{x})$ satisfies LSI with the constant $\mu_r$ and $L_r$ smoothness. Their KL divergence can be upper-bounded as*

$$\log \mathrm{KL}\left(q_0'(\cdot) \| q_{k,S-r\eta}(\cdot|\boldsymbol{x})\right) \leq \log \|\boldsymbol{x}\|^2 + \log\left[\frac{L_r^2 M}{\mu_r^2} \cdot \frac{d e^S}{1 - e^{-2S}}\right] + \frac{M e^{-S}}{1 - e^{-2S}}.$$

*Proof.* According to Lemma E.1, the density $q_{k,S-r\eta}(\cdot|\boldsymbol{x})$ can be presented as

$$q_{k,S-r\eta}(\boldsymbol{x}'|\boldsymbol{x}) \propto \exp\left(-f_{k,0}(\boldsymbol{x}') - \frac{\left\|\boldsymbol{x} - e^{-(S-r\eta)}\boldsymbol{x}'\right\|^2}{2(1 - e^{-2(S-r\eta)})}\right)$$

where $f_{k,0}(\boldsymbol{x}') = \nabla \log p_{k,0}(\boldsymbol{x}')$. Since it satisfies LSI with the constant, .i.e, $\mu_r$, due to Definition 1, we have

$$\begin{aligned}
\mathrm{KL}\left(q_0'(\cdot) \| q_{k,S-r\eta}(\cdot|\boldsymbol{x})\right) &\leq \frac{1}{2\mu_r} \cdot \int q_0'(\boldsymbol{x}') \left\|\nabla f_{k,0}(\boldsymbol{x}')\right\|^2 \mathrm{d}\boldsymbol{x}' \\
&\leq \mu_r^{-1} \cdot \left(\int q_0'(\boldsymbol{x}') \left\|\nabla f_{k,0}(\boldsymbol{x}') - \nabla f_{k,0}(\boldsymbol{0})\right\|^2 \mathrm{d}\boldsymbol{x}' + \int q_0'(\boldsymbol{x}') \left\|\nabla f_{k,0}(\boldsymbol{0})\right\|^2 \mathrm{d}\boldsymbol{x}'\right).
\end{aligned} \tag{29}$$

For the first term, we have

$$\begin{aligned}
&\int q_0'(\boldsymbol{x}') \left\|\nabla f_{k,0}(\boldsymbol{x}') - \nabla f_{k,0}(\boldsymbol{0})\right\|^2 \mathrm{d}\boldsymbol{x}' \\
&\leq L_r^2 \cdot \int q_0'(\boldsymbol{x}') \|\boldsymbol{x}'\|^2 \mathrm{d}\boldsymbol{x}' = L_r^2 \cdot \mathbb{E}_{q_0'}\left[\|\mathbf{x}'\|^2\right] = L_r^2 \cdot \left[\mathrm{Var}(\mathbf{x}') + \|\mathbb{E}\mathbf{x}'\|^2\right]
\end{aligned}$$

where the first inequality follows from **[A₁]**. The high-dimensional Gaussian distribution, i.e., $q_0'$ satisfies

$$\left\|\mathbb{E}_{q_0'}\left[\mathbf{x}'\right]\right\| = e^{S-r\eta} \|\boldsymbol{x}\| \quad \text{and} \quad \mathrm{Var}(\mathbf{x}') \leq d \cdot \left(e^{2(S-r\eta)} - 1\right),$$

where the last inequality follows from Lemma F.11, hence we have

$$\int q_0'(\boldsymbol{x}') \left\|\nabla f_{k,0}(\boldsymbol{x}') - \nabla f_{k,0}(\boldsymbol{0})\right\|^2 \mathrm{d}\boldsymbol{x}' \leq L_r^2 \cdot e^{2(S-r\eta)}(d + \|\boldsymbol{x}\|^2). \tag{30}$$

Then we consider to bound the second term of Eq 29. According to the definition of $\nabla f_{k,0}$, with the transition kernel of the OU process, we have

$$\begin{aligned}
&-\nabla f_{k,0}(\boldsymbol{x}') = \nabla \log p_{k,0}(\boldsymbol{x}') = \frac{\nabla p_{k,0}(\boldsymbol{x}')}{p_{k,0}(\boldsymbol{x}')} \\
&= \frac{\int_{\mathbb{R}^d} p_*(\boldsymbol{x}_0) \cdot \exp\left(\frac{-\|\boldsymbol{x} - e^{-kS}\boldsymbol{x}_0\|^2}{2(1 - e^{-2kS})}\right) \cdot \left(-\frac{\boldsymbol{x} - e^{-kS}\boldsymbol{x}_0}{(1 - e^{-2(T-t)})}\right) \mathrm{d}\boldsymbol{x}_0}{\int_{\mathbb{R}^d} p_*(\boldsymbol{x}_0) \cdot \exp\left(\frac{-\|\boldsymbol{x} - e^{-kS}\boldsymbol{x}_0\|^2}{2(1 - e^{-2kS})}\right) \mathrm{d}\boldsymbol{x}_0}.
\end{aligned}$$

Therefore, we have

$$
\begin{aligned}
\|\nabla f_{k,0}(\mathbf{0})\|^2 &= \left\| \frac{\int_{\mathbb{R}^d} p_*(\boldsymbol{x}_0) \cdot \exp\left( \frac{-e^{-2kS}\|\boldsymbol{x}_0\|^2}{2(1-e^{-2kS})} \right) \cdot \frac{e^{-kS}\boldsymbol{x}_0}{(1-e^{-2kS})} \,\mathrm{d}\boldsymbol{x}_0}{\int_{\mathbb{R}^d} p_*(\boldsymbol{x}_0) \cdot \exp\left( \frac{-e^{-2kS}\|\boldsymbol{x}_0\|^2}{2(1-e^{-2kS})} \right) \,\mathrm{d}\boldsymbol{x}_0} \right\|^2 \\
&\leq \frac{e^{-kS}}{1-e^{-2kS}} \cdot \int p_*(\boldsymbol{x}_0) \cdot \|\boldsymbol{x}_0\|^2 \,\mathrm{d}\boldsymbol{x}_0 \cdot \frac{\int p_*(\boldsymbol{x}_0) \cdot \exp\left( \frac{-e^{-2kS}\|\boldsymbol{x}_0\|^2}{(1-e^{-2kS})} \right) \,\mathrm{d}\boldsymbol{x}}{\left( \int_{\mathbb{R}^d} p_*(\boldsymbol{x}_0) \cdot \exp\left( \frac{-e^{-2kS}\|\boldsymbol{x}_0\|^2}{2(1-e^{-2kS})} \right) \,\mathrm{d}\boldsymbol{x}_0 \right)^2} \\
&\leq \frac{e^{-kS}}{1-e^{-2kS}} \cdot M \cdot \left( \int_{\mathbb{R}^d} p_*(\boldsymbol{x}_0) \cdot \exp\left( \frac{-e^{-2kS}\|\boldsymbol{x}_0\|^2}{2(1-e^{-2kS})} \right) \,\mathrm{d}\boldsymbol{x}_0 \right)^{-1}
\end{aligned}
\tag{31}
$$

where the first inequality follows from Holders' inequality, the second inequality follows from [**A**$_2$]. With, the following range:

$$
\frac{e^{-2kS}}{1-e^{-2kS}} \leq \frac{e^{-kS}}{1-e^{-2kS}} \leq \frac{e^{-S}}{1-e^{-2S}}
$$

we plug Eq 30 and Eq 31 into Eq 29 and obtain

$$
\begin{aligned}
\log \mathrm{KL}\left( q'_0(\cdot) \| q_{k,S-r\eta}(\cdot|\boldsymbol{x}) \right) \leq &\log \Big[ \mu_r^{-1} \cdot \left( L_r^2 \cdot e^{2(S-r\eta)} (d + \|\boldsymbol{x}\|^2) \right. \\
&+ \frac{e^{-kS}}{1-e^{-2kS}} \cdot M \left( \int_{\mathbb{R}^d} p_*(\boldsymbol{x}_0) \cdot \exp\left( \frac{-e^{-2kS}\|\boldsymbol{x}_0\|^2}{2(1-e^{-2kS})} \right) \,\mathrm{d}\boldsymbol{x}_0 \right)^{-1} \Big) \Big]
\end{aligned}
$$

Without loss of generality, we suppose both RHS of Eq 30 and Eq 31 are larger than 1. Then, we have

$$
\begin{aligned}
&\log \mathrm{KL}\left( q'_0(\cdot) \| q_{k,S-r\eta}(\cdot|\boldsymbol{x}) \right) \\
&\leq \log \left[ \frac{L_r^2}{\mu_r^2} \cdot e^{2(S-r\eta)} M \cdot \frac{e^{-kS}}{1-e^{-2kS}} \cdot (d + \|\boldsymbol{x}\|^2) \right] - \log \left[ \int_{\mathbb{R}^d} p_*(\boldsymbol{x}_0) \cdot \exp\left( \frac{-e^{-2kS}\|\boldsymbol{x}_0\|^2}{2(1-e^{-2kS})} \right) \,\mathrm{d}\boldsymbol{x}_0 \right] \\
&\leq \log \left[ \frac{L_r^2 M}{\mu_r^2} \cdot \frac{e^S}{1-e^{-2S}} \cdot (d + \|\boldsymbol{x}\|^2) \right] + \frac{e^{-2kS}}{2(1-e^{-2kS})} \cdot \int_{\mathbb{R}^d} p_*(\boldsymbol{x}_0) \|\boldsymbol{x}_0\|^2 \,\mathrm{d}\boldsymbol{x}_0 \\
&\leq \log \left[ \frac{L_r^2 M}{\mu_r^2} \cdot \frac{e^S}{1-e^{-2S}} \cdot (d + \|\boldsymbol{x}\|^2) \right] + \frac{Me^{-S}}{1-e^{-2S}} \leq \log \|\boldsymbol{x}\|^2 + \log \left[ \frac{L_r^2 M}{\mu_r^2} \cdot \frac{de^S}{1-e^{-2S}} \right] + \frac{Me^{-S}}{1-e^{-2S}}.
\end{aligned}
$$

Hence, the proof is completed. $\qquad\square$

**Corollary E.6.** *For any given $(k,r)$ in Alg 1 and $\boldsymbol{x} \in \mathbb{R}^d$, suppose the distribution $q_{k,S-r\eta}(\cdot|\boldsymbol{x})$ satisfies*

$$
\mu_r \boldsymbol{I} \preceq -\nabla^2 \log q_{k,S-r\eta}(\cdot|\boldsymbol{x}) \preceq L_r \boldsymbol{I},
$$

*and $\mathbf{x}'_j \sim q'_j(\cdot|\boldsymbol{x})$. If $0 < \tau_r \leq \mu_r/(8L_r^2)$, we have*

$$
\mathrm{KL}\left( q'_j \| q_* \right) \leq \exp\left( -\mu_r \tau_r j \right) \cdot \mathrm{KL}\left( q'_0 \| q_* \right) + \frac{32 L_r^2 d \tau_r}{\mu_r}
$$

*when the score estimation satisfies $\|\nabla \log p_{k,0} - \boldsymbol{v}'\|_\infty \leq L_r \sqrt{2d\tau_r}$.*

*Proof.* Due to the range $0 < \tau_r \leq \mu_r/8L_r^2$, we have $\mu_r \tau_r \leq 1/8$. In this condition, we have

$$
1 - \exp\left( -\mu_r \tau_r \right) \geq \frac{7}{8} \cdot \mu_r \tau_r.
$$

Plugging this into the following inequality obtained by the recursion of Eq. 28, we have

$$
\begin{aligned}
\mathrm{KL}\left( q'_j \| q_* \right) &\leq \exp\left( -\mu_r \tau_r j \right) \cdot \mathrm{KL}\left( q'_0 \| q_* \right) + \frac{28 L_r^2 d \tau_r^2}{(1 - \exp\left( -\mu_r \tau_r \right))} \\
&\leq \exp\left( -\mu_r \tau_r j \right) \cdot \mathrm{KL}\left( q'_0 \| q_* \right) + \frac{32 L_r^2 d \tau_r}{\mu_r}.
\end{aligned}
$$

In this condition, if we require the KL divergence to satisfy $\mathrm{KL}\left(q'_j \| q_*\right) \le \epsilon$, a sufficient condition is that

$$\exp\left(-\mu_r \tau_r j\right) \cdot \mathrm{KL}\left(q'_0 \| q_*\right) \le \frac{\epsilon}{2} \quad \text{and} \quad \frac{32 L_r^2 d\tau}{\mu_r} \le \frac{\epsilon}{2},$$

which is equivalent to

$$\tau_r \le \frac{\mu_r \epsilon}{64 L_r^2 d} \quad \text{and} \quad j \ge \frac{1}{\mu_r \tau_r} \cdot \log \frac{2\mathrm{KL}\left(q'_0 \| q_*\right)}{\epsilon}.$$

According to the upper bound of $\mathrm{KL}\left(q'_0 \| q_*\right)$ shown in Lemma E.5, we require

$$j \ge \frac{1}{\mu_r \tau_r} \cdot \left[\log \frac{\|\boldsymbol{x}\|^2}{\epsilon} + \log\left(\frac{2L_r^2 M}{\mu_r^2} \cdot \frac{de^S}{1 - e^{-2S}}\right) + \frac{Me^{-S}}{1 - e^{-2S}}\right].$$

$\square$

### E.3 CORE LEMMAS

**Lemma E.7.** *In Alg 1, for any $k \in \mathbb{N}_{0,K-1}$, $r \in \mathbb{N}_{0,R-1}$ and $\boldsymbol{x} \in \mathbb{R}^d$, we have*

$$\mathbb{P}\left[\left\|\mathbf{v}^{\leftarrow}_{k,r\eta}(\boldsymbol{x}) - \nabla \log p_{k,S-r\eta}(\boldsymbol{x})\right\|^2 \le 10\epsilon\right] \ge 1 - \delta$$

*by requiring the segment length $S$, the sample number $n_{k,r}$ and the step size of inner loops $\tau_r$ and the iteration number of inner loops $m_{k,r}$ satisfy*

$$S = \frac{1}{2}\log\frac{2L+1}{2L}, \quad n_{k,r} \ge \frac{4}{\epsilon(1 - e^{-2(S-r\eta)})} \cdot \max\left\{d, -2\log\delta\right\},$$

$$\tau_r \le \frac{\mu_r}{64 L_r^2 d} \cdot (1 - e^{-2(S-r\eta)})\epsilon \quad \text{and} \quad m_{k,r} \ge \frac{64 L_r^2 d}{\mu_r^2(1 - e^{-2(S-r\eta)})\epsilon} \cdot \left[\log\frac{d\|\boldsymbol{x}\|^2}{(1 - e^{-2(S-r\eta)})\epsilon} + C_{m,1}\right],$$

*where $C_{m,1} = \log\left(2M \cdot 3^2 \cdot 5L\right) + M \cdot 3L$. In this condition that choosing the $\tau_r$ to its upper bound, we required the score estimation in the inner loop satisfies*

$$\left\|\nabla \log p_{k,0}(\boldsymbol{x}') - \mathbf{v}'_{k,0}(\boldsymbol{x}')\right\| \le \frac{e^{-(S-r\eta)}\epsilon^{0.5}}{8}.$$

*Proof.* With a little abuse of notation, for each loop $i \in \mathbb{N}_{1,n_{k,r}}$ in Line 4 of Alg 1, we denote the underlying distribution of **output particles** as $\mathbf{x}'_i \sim q'_{k,S-r\eta}(\cdot|\boldsymbol{x})$ for any $k \in \mathbb{N}_{0,K-1}$, $r \in \mathbb{N}_{0,R-1}$ and $\boldsymbol{x} \in \mathbb{R}^d$ in this lemma. According to Line 10 in Alg 1, we have

$$
\begin{aligned}
&\left\|\mathbf{v}^{\leftarrow}_{k,r\eta}(\boldsymbol{x}) - \nabla \log p_{k,S-r\eta}(\boldsymbol{x})\right\|^2 \\
&= \left\|\frac{1}{n_{r,k}}\sum_{i=1}^{n_{r,k}}\left(-\frac{\boldsymbol{x} - e^{-(S-r\eta)}\mathbf{x}'_i}{1 - e^{-2(S-r\eta)}}\right) - \mathbb{E}_{\mathbf{x}'\sim q_{k,S-r\eta}(\cdot|\boldsymbol{x})}\left[-\frac{\boldsymbol{x} - e^{-(S-r\eta)}\mathbf{x}'}{1 - e^{-2(S-r\eta)}}\right]\right\|^2 \\
&= \frac{e^{-2(S-r\eta)}}{\left(1 - e^{-2(S-r\eta)}\right)^2} \cdot \left\|-\frac{1}{n_{r,k}}\sum_{i=1}^{n_{r,k}}\mathbf{x}'_i - \mathbb{E}_{\mathbf{x}'\sim q_{k,S-r\eta}(\cdot|\boldsymbol{x})}\left[\mathbf{x}'\right]\right\|^2 \\
&\le \frac{2e^{-2(S-r\eta)}}{\left(1 - e^{-2(S-r\eta)}\right)^2} \cdot \left\|-\frac{1}{n_{r,k}}\sum_{i=1}^{n_{r,k}}\mathbf{x}'_i + \mathbb{E}_{\mathbf{x}'\sim q'_{k,S-r\eta}(\cdot|\boldsymbol{x})}\left[\mathbf{x}'\right]\right\|^2 \\
&\quad + \frac{2e^{-2(S-r\eta)}}{\left(1 - e^{-2(S-r\eta)}\right)^2} \cdot \left\|-\mathbb{E}_{\mathbf{x}'\sim q'_{k,S-r\eta}(\cdot|\boldsymbol{x})}\left[\mathbf{x}'\right] + \mathbb{E}_{\mathbf{x}'\sim q_{k,S-r\eta}(\cdot|\boldsymbol{x})}\left[\mathbf{x}'\right]\right\|^2
\end{aligned}
\tag{32}
$$

In the following, we respectively upper bound the concentration error and the mean gap between $q'_{k,S-r\eta}(\cdot|\boldsymbol{x})$ and $q_{k,S-r\eta}(\cdot|\boldsymbol{x})$ corresponding to the former and the latter term in Eq 32.

**Upper bound the concentration error.** The choice of $S$, i.e., $S = \frac{1}{2}\log\left(\frac{2L+1}{2L}\right)$, Lemma E.2 demonstrate that suppose

$$\mu_r = \frac{1}{2} \cdot \frac{e^{-2(S-r\eta)}}{1 - e^{-2(S-r\eta)}} \quad \text{and} \quad L_r = \frac{3}{2} \cdot \frac{e^{-2(S-r\eta)}}{1 - e^{-2(S-r\eta)}}.$$

Then, we have

$$\mu_r \boldsymbol{I} \preceq -\nabla^2 \log q_{k,S-r\eta}(\boldsymbol{x}'|\boldsymbol{x}) \preceq L_r \boldsymbol{I}.$$

According to Alg 1, we utilize ULA as the inner loop (Line 4 – Line 9) to sample from $q_{k,S-r\eta}(\cdot|\boldsymbol{x})$. By requiring the step size, i.e., $\tau_r$ to satisfy $\tau_r \leq 1/L_r$, with Lemma F.8, we know that the underlying distribution of output particles of the inner loops satisfies, i.e., $q'_{k,S-r\eta}(\cdot|\boldsymbol{x})$ satisfies LSI with a constant $\mu'_r$ satisfying

$$\mu'_r \geq \frac{\mu_r}{2} \geq \frac{e^{-2(S-r\eta)}}{4(1 - e^{-2(S-r\eta)})}.$$

In this condition, we employ Lemma E.3, by requiring

$$n_{k,r} \geq \frac{4}{\epsilon(1 - e^{-2(S-r\eta)})} \cdot \max\{d, -2\log\delta\}$$

$$\geq \frac{1}{\mu'_r} \cdot \left(\frac{e^{-(S-r\eta)}}{(1 - e^{-2(S-r\eta)})\epsilon^{0.5}}\right)^2 \cdot \max\{d, -2\log\delta\}.$$

and obtain

$$\mathbb{P}\left[\frac{2e^{-2(S-r\eta)}}{\left(1 - e^{-2(S-r\eta)}\right)^2} \cdot \left\|-\frac{1}{n_{r,k}}\sum_{i=1}^{n_{r,k}}\mathbf{x}'_i + \mathbb{E}_{\mathbf{x}'\sim q'_{k,S-r\eta}(\cdot|\boldsymbol{x})}\left[\mathbf{x}'\right]\right\|^2 \leq 2\epsilon\right]$$

$$=\mathbb{P}\left[\left\|-\frac{1}{n_{r,k}}\sum_{i=1}^{n_{r,k}}\mathbf{x}'_i + \mathbb{E}_{\mathbf{x}'\sim q'_{k,S-r\eta}(\cdot|\boldsymbol{x})}\left[\mathbf{x}'\right]\right\| \leq \frac{(1 - e^{-2(S-r\eta)})\epsilon^{0.5}}{e^{-(S-r\eta)}}\right] \geq 1 - \delta.$$

**Upper bound the mean gap.** According to Lemma E.2 and Lemma F.4, we know $q_{k,S-r\eta}(\boldsymbol{x}'|\boldsymbol{x})$ satisfies LSI with constant

$$\mu_r \geq \frac{e^{-2(S-r\eta)}}{2(1 - e^{-2(S-r\eta)})}.$$

By introducing the optimal coupling between $q_{k,S-r\eta}(\cdot|\boldsymbol{x})$ and $q'_{k,S-r\eta}(\cdot|\boldsymbol{x})$, we have

$$\left\|-\mathbb{E}_{\mathbf{x}'\sim q'_{k,S-r\eta}(\cdot|\boldsymbol{x})}\left[\mathbf{x}'\right] + \mathbb{E}_{\mathbf{x}'\sim q_{k,S-r\eta}(\cdot|\boldsymbol{x})}\left[\mathbf{x}'\right]\right\|^2$$

$$\leq W_2^2\left(q'_{k,S-r\eta}(\cdot|\boldsymbol{x}), q_{k,S-r\eta}(\cdot|\boldsymbol{x})\right) \leq \frac{2}{\mu_r}\mathrm{KL}\left(q'_{k,S-r\eta}(\cdot|\boldsymbol{x})\|q_{k,S-r\eta}(\cdot|\boldsymbol{x})\right),$$

$$(33)$$

where the last inequality follows from Talagrand inequality Vempala & Wibisono (2019). Hence, the mean gap can be upper-bounded as

$$\frac{2e^{-2(S-r\eta)}}{\left(1 - e^{-2(S-r\eta)}\right)^2} \cdot \left\|-\mathbb{E}_{\mathbf{x}'\sim q'_{k,S-r\eta}(\cdot|\boldsymbol{x})}\left[\mathbf{x}'\right] + \mathbb{E}_{\mathbf{x}'\sim q_{k,S-r\eta}(\cdot|\boldsymbol{x})}\left[\mathbf{x}'\right]\right\|^2$$

$$\leq \frac{2e^{-2(S-r\eta)}}{\left(1 - e^{-2(S-r\eta)}\right)^2} \cdot \frac{2}{\mu_r}\mathrm{KL}\left(q'_{k,S-r\eta}(\cdot|\boldsymbol{x})\|q_{k,S-r\eta}(\cdot|\boldsymbol{x})\right)$$

$$\leq \frac{8}{(1 - e^{-2(S-r\eta)})}\mathrm{KL}\left(q'_{k,S-r\eta}(\cdot|\boldsymbol{x})\|q_{k,S-r\eta}(\cdot|\boldsymbol{x})\right).$$

To provide $\epsilon$-level upper bound, we expect the required accuracy of KL convergence of inner loops to satisfy

$$\mathrm{KL}\left(q'_{k,S-r\eta}(\cdot|\boldsymbol{x})\|q_{k,S-r\eta}(\cdot|\boldsymbol{x})\right) \leq (1 - e^{-2(S-r\eta)})\epsilon.$$

According to Corollary E.6, to achieve such accuracy, we require the step size and the iteration number of inner loops to satisfy

$$\tau_r \leq \frac{\mu_r}{64L_r^2 d} \cdot (1 - e^{-2(S-r\eta)})\epsilon \quad \text{and}$$

$$m_{k,r} \geq \frac{1}{\mu_r} \cdot \frac{64L_r^2 d}{\mu_r(1 - e^{-2(S-r\eta)})\epsilon} \cdot \left[\log\frac{\|\boldsymbol{x}\|^2}{(1 - e^{-2(S-r\eta)})\epsilon} + \log\left(\frac{2L_r^2 M}{\mu_r^2} \cdot \frac{de^S}{1 - e^{-2S}}\right) + \frac{Me^{-S}}{1 - e^{-2S}}\right].$$

To simplify notation, we suppose $L \geq 1$ without loss of generality, and we the following equations:

$$\frac{L_r}{\mu_r} = 3, \quad e^S = \exp\left(\frac{1}{2} \log \frac{2L+1}{2L}\right) = \sqrt{\frac{2L+1}{2L}},$$

$$\left(1 - e^{-2S}\right)^{-1} = (2L+1),$$

which implies

$$\log \frac{d\|\boldsymbol{x}\|^2}{(1 - e^{-2(S-r\eta)})\epsilon} + \log\left(2M \cdot 3^2 \cdot 5L\right) + M \cdot 3L$$

$$\geq \log \frac{d\|\boldsymbol{x}\|^2}{(1 - e^{-2(S-r\eta)})\epsilon} + \log\left(2M \cdot \frac{L_r^2}{\mu_r^2} \cdot \sqrt{\frac{2L+1}{2L}} \cdot (2L+1)\right) + M \cdot (2L+1) \cdot \sqrt{\frac{2L}{2L+1}}$$

$$= \log \frac{d\|\boldsymbol{x}\|^2}{(1 - e^{-2(S-r\eta)})\epsilon} + \log\left(\frac{2L_r^2 M}{\mu_r^2} \cdot \frac{e^S}{1 - e^{-2S}}\right) + \frac{Me^{-S}}{1 - e^{-2S}}.$$

Therefore, we only require $m_{k,r}$ satisfies

$$m_{k,r} \geq \frac{1}{\mu_r} \cdot \frac{64 L_r^2 d}{\mu_r(1 - e^{-2(S-r\eta)})\epsilon} \cdot \left[\log \frac{d\|\boldsymbol{x}\|^2}{(1 - e^{-2(S-r\eta)})\epsilon} + C_{m,1}\right]$$

where $C_{m,1} = \log\left(2M \cdot 3^2 \cdot 5L\right) + M \cdot 3L$. For simplicity, we choose $\tau_r$ as its upper bound and lower bound, respectively. In this condition, we still require

$$\|\nabla \log p_{k,0} - \boldsymbol{v}'\| \leq \frac{e^{-(S-r\eta)}\epsilon^{0.5}}{8} \leq \frac{1}{4} \cdot \sqrt{\frac{\mu_r(1 - e^{-2(S-r\eta)})}{2}} \cdot \epsilon \leq L_r\sqrt{2d\tau_r}$$

where the first inequality follows from the range of $\mu_r$, and the last inequality is satisfied when we choose $\tau_r$ to its upper bound. Hence, the proof is completed. $\square$

**Lemma E.8** (Errors from fine-grained score estimation). *Under the notation in Section A, suppose the step size satisfy $\eta = C_\eta(d+M)^{-1}\epsilon$, we have*

$$\mathbb{P}\left[\left\|\nabla \log p_{k,S-r\eta}(\boldsymbol{x}) - \mathbf{v}_{k,r\eta}^{\leftarrow}(\boldsymbol{x})\right\|^2 \leq 10\epsilon, \forall \boldsymbol{x} \in \mathbb{R}^d\right]$$

$$\geq (1-\delta) \cdot \left(\min_{\boldsymbol{x}' \in \mathbb{S}_{k,r}(\boldsymbol{x},\epsilon)} \mathbb{P}\left[\left\|\nabla \log p_{k,0}(\boldsymbol{x}') - \mathbf{v}_{k-1,0}^{\leftarrow}(\boldsymbol{x}')\right\|^2 \leq \frac{\epsilon}{96}\right]\right)^{n_{k,r}(10\epsilon) \cdot m_{k,r}(10\epsilon,\boldsymbol{x})},$$

*where $\mathbb{S}_{k,r}(\boldsymbol{x}, 10\epsilon)$ denotes the set of particles appear in Alg 1 when the input is $(k, r, \boldsymbol{x}, 10\epsilon)$. For any $(k, r) \in \mathbb{N}_{0,K-1} \times \mathbb{N}_{0,R-1}$ by requiring*

$$n_{k,r}(10\epsilon) = C_n \cdot \frac{(d+M) \cdot \max\{d, -2\log\delta\}}{(10\epsilon)^2} \quad \text{where} \quad C_n = 2^6 \cdot 5^2 \cdot C_\eta^{-1},$$

$$m_{k,r}(10\epsilon, \boldsymbol{x}) = C_m \cdot \frac{(d+M)^3 \cdot \max\{\log \|\boldsymbol{x}\|^2, 1\}}{(10\epsilon)^3} \quad \text{where} \quad C_m = 2^9 \cdot 3^2 \cdot 5^3 \cdot C_{m,1} C_\eta^{-1.5}.$$

*Proof.* According to Line 9 of Alg 1, for any $\boldsymbol{x} \in \mathbb{R}^d$, the score estimation $\mathbf{v}_{k,r\eta}^{\leftarrow}$ is constructed by estimating the mean in RHS of the following expectation using $n_{k,r}$ samples (i.e., calculating the empirical mean):

$$\nabla_{\boldsymbol{x}} \log p_{k,S-r\eta}(\boldsymbol{x}) = \mathbb{E}_{\boldsymbol{x}' \sim q_{k,S-r\eta}(\cdot|\boldsymbol{x})}\left[-\frac{\boldsymbol{x} - e^{-(S-r\eta)}\mathbf{x}'}{(1 - e^{-2(S-r\eta)})}\right] \tag{34}$$

$$\text{where} \quad q_{k,S-r\eta}(\boldsymbol{x}'|\boldsymbol{x}) \propto \exp\left(\log p_{k,0}(\boldsymbol{x}') - \frac{\|\boldsymbol{x} - e^{-(S-r\eta)}\boldsymbol{x}'\|^2}{2\left(1 - e^{-2(S-r\eta)}\right)}\right). \tag{35}$$

Then in order to guarantee an accurate estimation for $\nabla_{\boldsymbol{x}} \log p_{k,S-r\eta}(\boldsymbol{x})$, i.e., denoted by $\mathbf{v}_{k,r\eta}^{\leftarrow}(\boldsymbol{x})$, with Lemma E.7, we require

1. Get a precise estimation for $\nabla \log p_{k,0}(\boldsymbol{x}')$, in order to guarantee that the estimation for $\nabla \log q_{k,S-r\eta}(\boldsymbol{x}'|\boldsymbol{x})$ is accurate. In particular, we require

$$\left\| \nabla \log p_{k,0}(\boldsymbol{x}'_{i,j}) - \mathbf{v}^{\leftarrow}_{k-1,0}(\boldsymbol{x}'_{i,j}) \right\| \leq \frac{e^{-(S-r\eta)}\epsilon^{0.5}}{8}.$$

2. Based on the $\nabla \log q_{k,S-r\eta}(\boldsymbol{x}'|\boldsymbol{x})$, we run ULA with appropriate step size $\tau_r$ and iteration number $m_{k,r}$ satisfying

$$\tau_r \leq \frac{\mu_r}{64 L_r^2 d} \cdot (1 - e^{-2(S-r\eta)})\epsilon \quad \text{and} \quad m_{k,r} \geq \frac{1}{\mu_r} \cdot \frac{64 L_r^2 d}{\mu_r (1 - e^{-2(S-r\eta)})\epsilon} \cdot \log \frac{2C_0}{(1 - e^{-2(S-r\eta)})\epsilon} \tag{36}$$

to generate samples $\boldsymbol{x}'$ whose underlying distribution $q'_{k,S-r\eta}(\cdot|\boldsymbol{x})$ is sufficiently close to $q_{k,S-r\eta}(\boldsymbol{x}'|\boldsymbol{x})$, i.e.,

$$\mathrm{KL}\left( q'_{k,S-r\eta}(\cdot|\boldsymbol{x}) \| q_{k,S-r\eta}(\cdot|\boldsymbol{x}) \right) \leq (1 - e^{-2(S-r\eta)})\epsilon.$$

3. Generate a sufficient number of samples satisfying

$$n_{k,r} \geq \frac{4}{\epsilon(1 - e^{-2(S-r\eta)})} \cdot \max\left\{ d, -2\log\delta \right\}. \tag{37}$$

such that the empirical estimation of the expectation in (34) is accurate, i.e.,

$$\mathbb{P}\left[ \left\| \nabla \log p_{k,S-r\eta}(\boldsymbol{x}) - \tilde{\mathbf{v}}_{k,r\eta}(\boldsymbol{x}) \right\|^2 \leq 10\epsilon \right]$$

$$= \mathbb{P}\left[ \left\| \nabla \log p_{k,S-r\eta}(\boldsymbol{x}) - \frac{1}{n_{k,r}} \sum_{i=1}^{n_{k,r}} \left[ -\frac{\boldsymbol{x} - e^{-(S-r\eta)}\boldsymbol{x}'_{i,m_{k,r}}}{\left(1 - e^{-2(S-r\eta)}\right)} \right] \right\| \leq 10\epsilon \right] \geq 1 - \delta.$$

Due to the fact $r\eta \geq 0$, the first condition can be achieved by requiring

$$\left\| \nabla \log p_{k,0}(\boldsymbol{x}'_{i,j}) - \mathbf{v}^{\leftarrow}_{k-1,0}(\boldsymbol{x}'_{i,j}) \right\| \leq \sqrt{\frac{2}{3}} \cdot \frac{\epsilon^{0.5}}{8} \leq \sqrt{\frac{2L}{2L+1}} \cdot \frac{\epsilon^{0.5}}{8} = \frac{e^{-S}\epsilon^{0.5}}{8} \leq \frac{e^{-(S-r\eta)}\epsilon^{0.5}}{8},$$

where the second inequality is established by supposing $L \geq 1$ without loss of generality, and the last equation follows from the choice of $S$.

To investigate the setting of hyper-parameters, i.e., the number of samples for empirical mean estimation $n_{k,r}$ and the number of iterations for ULA $m_{k,r}$. We first reformulate them as two functions, i.e.,

$$n_{k,r}(10\epsilon) = C_n \cdot \frac{(d+M) \cdot \max\{d, -2\log\delta\}}{(10\epsilon)^2} \quad \text{where} \quad C_n = 2^6 \cdot 5^2 \cdot C_\eta^{-1},$$

$$m_{k,r}(10\epsilon, \boldsymbol{x}) = C_m \cdot \frac{(d+M)^3 \cdot \max\{\log\|\boldsymbol{x}\|^2, 1\}}{(10\epsilon)^3} \quad \text{where} \quad C_m = 2^9 \cdot 3^2 \cdot 5^3 \cdot C_{m,1} C_\eta^{-1.5}.$$

since this presentation helps to explain the connection between them and the input of Alg 1. Different from the results shown in Lemma E.7, $n_{k,r}(\cdot)$ and $m_{k,r}(\cdot, \cdot)$ is independent with $k$ and $r$. However, these choices will still make Eq 36 and Eq 37 establish, because

$$n_{k,r}(10\epsilon) = \frac{16}{\epsilon} \cdot \frac{(d+M)}{C_\eta\epsilon} \cdot \max\left\{d, -2\log\delta\right\} \geq \frac{16}{\epsilon\eta} \cdot \max\left\{d, -2\log\delta\right\}$$

$$\geq \frac{16}{\epsilon(1 - e^{-2\eta})} \cdot \max\left\{d, -2\log\delta\right\} \geq \frac{16}{\epsilon(1 - e^{-2(S-r\eta)})} \cdot \max\left\{d, -2\log\delta\right\}$$

$$m_{k,r}(10\epsilon, \boldsymbol{x}) = 576 \cdot \frac{(d+M)^3}{\epsilon^3} \cdot \frac{C_{m,1}}{C_\eta^{1.5}} \cdot \max\{\log\|\boldsymbol{x}\|^2, 1\} \geq 64 \cdot \frac{L_r^2}{\mu_r^2} \cdot \left(\frac{d}{\epsilon\eta}\right)^{1.5} \cdot C_{m,1} \cdot \max\{\log\|\boldsymbol{x}\|^2, 1\}$$

$$\geq 64 \cdot \frac{L_r^2}{\mu_r^2} \cdot \frac{d}{\epsilon\eta} \log \frac{d}{\epsilon\eta} \cdot C_{m,1} \cdot \max\{\log\|\boldsymbol{x}\|^2, 1\} \geq 64 \cdot \frac{L_r^2}{\mu_r^2} \cdot \frac{d}{\epsilon\eta} \left(\log \frac{d\|\boldsymbol{x}\|^2}{\epsilon\eta} + C_{m,1}\right)$$

$$\geq 64 \cdot \frac{L_r^2}{\mu_r^2} \cdot \frac{d}{\epsilon(1 - e^{-2\eta})} \left(\log \frac{d\|\boldsymbol{x}\|^2}{\epsilon(1 - e^{-2\eta})} + C_{m,1}\right)$$

$$\geq \frac{64 L_r^2 d}{\mu_r^2(1 - e^{-2(S-r\eta)})\epsilon} \cdot \left(\log \frac{d\|\boldsymbol{x}\|^2}{(1 - e^{-2(S-r\eta)})\epsilon} + C_{m,1}\right)$$

with the proper choice of step size, i.e., $\eta = C_\eta(d + M)^{-1}\epsilon$. With these settings, Lemma E.7 demonstrates that

$$\mathbb{P}\left[\left\|\nabla \log p_{k,S-r\eta}(\boldsymbol{x}) - \mathbf{v}_{k,r\eta}^{\leftarrow}(\boldsymbol{x})\right\|^2 \leq 10\epsilon, \forall \boldsymbol{x} \in \mathbb{R}^d\right|$$

$$\bigcap_{\boldsymbol{x}' \in \mathbb{S}_{k,r}(\boldsymbol{x}, 10\epsilon)} \left\|\nabla \log p_{k,0}(\boldsymbol{x}') - \mathbf{v}_{k-1,0}^{\leftarrow}(\boldsymbol{x}')\right\|^2 \leq \frac{\epsilon}{96}\right] \geq 1 - \delta.$$

where $\mathbb{S}_{k,r}(\boldsymbol{x}, 10\epsilon)$ denotes the set of particles appear in Alg 1 when the input is $(k, r, \boldsymbol{x}, 10\epsilon)$ except for the recursion. It satisfies $|\mathbb{S}_{k,r}(\boldsymbol{x}, 10\epsilon)| = n_{k,r}(10\epsilon) \cdot m_{k,r}(10\epsilon, \boldsymbol{x})$. Furthermore, we have

$$\mathbb{P}\left[\left\|\nabla \log p_{k,S-r\eta}(\boldsymbol{x}) - \mathbf{v}_{k,r\eta}^{\leftarrow}(\boldsymbol{x})\right\|^2 \leq 10\epsilon\right]$$

$$\geq \mathbb{P}\left[\left\|\nabla \log p_{k,S-r\eta}(\boldsymbol{x}) - \mathbf{v}_{k,r\eta}^{\leftarrow}(\boldsymbol{x})\right\|^2 \leq 10\epsilon \middle| \bigcap_{\boldsymbol{x}' \in \mathbb{S}_{k,r}(\boldsymbol{x}, 10\epsilon)} \left\|\nabla \log p_{k,0}(\boldsymbol{x}') - \mathbf{v}_{k-1,0}^{\leftarrow}(\boldsymbol{x}')\right\|^2 \leq \frac{\epsilon}{96}\right]$$

$$\cdot \mathbb{P}\left[\bigcap_{\boldsymbol{x}' \in \mathbb{S}_{k,r}(\boldsymbol{x}, 10\epsilon)} \left\|\nabla \log p_{k,0}(\boldsymbol{x}') - \mathbf{v}_{k-1,0}^{\leftarrow}(\boldsymbol{x}')\right\|^2 \leq \frac{\epsilon}{96}\right]$$

$$\geq (1 - \delta) \cdot \mathbb{P}\left[\bigcap_{\boldsymbol{x}' \in \mathbb{S}_{k,r}(\boldsymbol{x}, 10\epsilon)} \left\|\nabla \log p_{k,0}(\boldsymbol{x}') - \mathbf{v}_{k-1,0}^{\leftarrow}(\boldsymbol{x}')\right\|^2 \leq \frac{\epsilon}{96}\right].$$

$$(38)$$

Considering that for each $\boldsymbol{x}'_{i,j}$, the score estimation, i.e., $\mathbf{v}_{k-1,0}^{\leftarrow}(\boldsymbol{x}'_{i,j})$ is independent, hence, we have

$$\mathbb{P}\left[\bigcap_{\boldsymbol{x}' \in \mathbb{S}_{k,r}(\boldsymbol{x}, 10\epsilon)} \left\|\nabla \log p_{k,0}(\boldsymbol{x}') - \mathbf{v}_{k-1,0}^{\leftarrow}(\boldsymbol{x}')\right\|^2 \leq \frac{\epsilon}{96}\right]$$

$$= \prod_{\boldsymbol{x}' \in \mathbb{S}_{k,r}(\boldsymbol{x}, 10\epsilon)} \mathbb{P}\left[\left\|\nabla \log p_{k,0}(\boldsymbol{x}') - \mathbf{v}_{k-1,0}^{\leftarrow}(\boldsymbol{x}')\right\|^2 \leq \frac{\epsilon}{96}\right] \qquad (39)$$

$$\geq \left(\min_{\boldsymbol{x}' \in \mathbb{S}_{k,r}(\boldsymbol{x}, \epsilon)} \mathbb{P}\left[\left\|\nabla \log p_{k,0}(\boldsymbol{x}') - \mathbf{v}_{k-1,0}^{\leftarrow}(\boldsymbol{x}')\right\|^2 \leq \frac{\epsilon}{96}\right]\right)^{|\mathbb{S}_{k,r}(\boldsymbol{x}, \epsilon)|}$$

Therefore, combining Eq 38 and Eq 39, we have

$$\mathbb{P}\left[\left\|\nabla \log p_{k,S-r\eta}(\boldsymbol{x}) - \mathbf{v}_{k,r\eta}^{\leftarrow}(\boldsymbol{x})\right\|^2 \leq 10\epsilon\right]$$

$$\geq (1 - \delta) \cdot \left(\min_{\boldsymbol{x}' \in \mathbb{S}_{k,r}(\boldsymbol{x}, \epsilon)} \mathbb{P}\left[\left\|\nabla \log p_{k,0}(\boldsymbol{x}') - \mathbf{v}_{k-1,0}^{\leftarrow}(\boldsymbol{x}')\right\|^2 \leq \frac{\epsilon}{96}\right]\right)^{n_{k,r}(10\epsilon) \cdot m_{k,r}(10\epsilon, \boldsymbol{x})},$$

and the proof is completed. $\qquad \square$

**Corollary E.9** (Errors from coarse-grained score estimation). *Under the notation in Section A, suppose the step size satisfy $\eta = C_1(d + M)^{-1}\epsilon$, we have*

$$\mathbb{P}\left[\left\|\nabla \log p_{k+1,0}(\boldsymbol{x}) - \mathbf{v}_{k,0}^{\leftarrow}(\boldsymbol{x})\right\|^2 \leq 10\epsilon, \forall \boldsymbol{x} \in \mathbb{R}^d\right]$$

$$\geq (1 - \delta) \cdot \left(\min_{\boldsymbol{x}' \in \mathbb{S}_{k,0}(\boldsymbol{x}, \epsilon)} \mathbb{P}\left[\left\|\nabla \log p_{k,0}(\boldsymbol{x}') - \mathbf{v}_{k-1,0}^{\leftarrow}(\boldsymbol{x}')\right\|^2 \leq \frac{\epsilon}{96}\right]\right)^{n_{k,0}(10\epsilon) \cdot m_{k,0}(10\epsilon, \boldsymbol{x})}, \qquad (40)$$

*where $\mathbb{S}_{k,0}(\boldsymbol{x}, 10\epsilon)$ denotes the set of particles appear in Alg 1 when the input is $(k, 0, \boldsymbol{x}, 10\epsilon)$. For any $k \in \mathbb{N}_{1,K-1}$ by requiring*

$$n_{k,0}(10\epsilon) = C_n \cdot \frac{(d + M) \cdot \max\{d, -2 \log \delta\}}{(10\epsilon)^2} \quad \text{where} \quad C_n = 2^6 \cdot 5^2 \cdot C_\eta^{-1},$$

$$m_{k,0}(10\epsilon, \boldsymbol{x}) = C_m \cdot \frac{(d + M)^3 \cdot \max\{\log \|\boldsymbol{x}\|^2, 1\}}{(10\epsilon)^3} \quad \text{where} \quad C_m = 2^9 \cdot 3^2 \cdot 5^3 \cdot C_{m,1} C_\eta^{-1.5}.$$

*Besides, for any $\boldsymbol{x} \in \mathbb{R}^d$, we have*

$$\mathbb{P}\left[\left\|\nabla \log p_{0,0}(\boldsymbol{x}') - \mathbf{v}_{-1,0}^{\leftarrow}(\boldsymbol{x}')\right\|^2 \leq \frac{\epsilon}{96}, \forall \boldsymbol{x}' \in \mathbb{R}^d\right] = 1$$

*by requiring $\tilde{\mathbf{v}}_{-1,0}(\boldsymbol{x}') = -\nabla f_*(\boldsymbol{x}')$, which corresponds to Line 2 in Alg 1.*

*Proof.* When $k > 0$, plugging $r = 0$ into Lemma E.8, we can obtain the result except inequality Eq 40. Instead, we have

$$\mathbb{P}\left[\left\|\nabla \log p_{k,S}(\boldsymbol{x}) - \mathbf{v}_{k,0}^{\leftarrow}(\boldsymbol{x})\right\|^2 \leq 10\epsilon, \forall \boldsymbol{x} \in \mathbb{R}^d\right]$$

$$\geq (1-\delta) \cdot \left(\min_{\boldsymbol{x}' \in \mathbb{S}_{k,0}(\boldsymbol{x},\epsilon)} \mathbb{P}\left[\left\|\nabla \log p_{k,0}(\boldsymbol{x}') - \mathbf{v}_{k-1,0}^{\leftarrow}(\boldsymbol{x}')\right\|^2 \leq \frac{\epsilon}{96}\right]\right)^{n_{k,0}(10\epsilon) \cdot m_{k,0}(10\epsilon, \boldsymbol{x})} \quad (41)$$

Since the forward process, i.e., SDE 1, satisfies $\mathbf{x}_{k,S} = \mathbf{x}_{k+1,0}$, we have

$$p_{k,S}(\boldsymbol{x}) = p_{k+1,0}(\boldsymbol{x}) = \int p_*(\boldsymbol{y}) \cdot \left(2\pi \left(1 - e^{-2(k+1)S}\right)\right)^{-d/2} \cdot \exp\left[\frac{-\left\|\boldsymbol{x} - e^{-(k+1)S}\boldsymbol{y}\right\|^2}{2\left(1 - e^{-2(k+1)S}\right)}\right] \mathrm{d}\boldsymbol{y},$$

which means $\nabla \log p_{k,S} = \nabla \log p_{k+1,0}$. Therefore, Eq 40 is established.

When $k = 0$, due to the definition of $\tilde{\mathbf{v}}_{-1,0}$ in Eq 7, we know Eq 41 is established. Hence, the proof is completed. □

**Lemma E.10** (Errors from score estimation). *Under the notation in Section A, suppose the step size satisfy $\eta = C_\eta(d + M)^{-1}\epsilon$, we have*

$$\mathbb{P}\left[\bigcap_{\substack{k \in \mathbb{N}_{0,K-1} \\ r \in \mathbb{N}_{0,R-1}}} \left\|\nabla \log p_{k,S-r\eta}(\boldsymbol{x}_{k,r\eta}^{\leftarrow}) - \mathbf{v}_{k,r\eta}^{\leftarrow}(\boldsymbol{x}_{k,r\eta}^{\leftarrow})\right\|^2 \leq 10\epsilon\right] \geq 1 - \epsilon$$

*with Alg 1 by properly choosing the number for mean estimations and ULA iterations. The total gradient complexity will be at most*

$$\exp\left[\mathcal{O}\left(\left(\log \frac{Ld + M}{\epsilon}\right)^3 \cdot \max\left\{\log \log Z^2, 1\right\}\right)\right],$$

*where $Z$ is the maximal norm of particles that appear in Alg 2.*

*Proof.* We begin with lower bounding the following probability with $(i, j) \in \mathbb{N}_{0,K-1} \times \mathbb{N}_{0,R-1}$ and $(i, j) \neq (0, 0)$,

$$\mathbb{P}\left[\left\|\nabla \log p_{k,S-r\eta}(\boldsymbol{x}_{k,r\eta}^{\leftarrow}) - \mathbf{v}_{k,r\eta}^{\leftarrow}(\boldsymbol{x}_{k,r\eta}^{\leftarrow})\right\|^2 \leq 10\epsilon\right].$$

In the following part of this Lemma, we set $\eta = C_\eta(d + M)^{-1}\epsilon$ and denote $\delta$ as a tiny positive constant waiting for determining. With Lemma E.8, we have

$$\mathbb{P}\left[\left\|\nabla \log p_{k,S-r\eta}(\boldsymbol{x}_{k,r\eta}^{\leftarrow}) - \mathbf{v}_{k,r\eta}^{\leftarrow}(\boldsymbol{x}_{k,r\eta}^{\leftarrow})\right\|^2 \leq 10\epsilon\right]$$

$$\geq (1-\delta) \cdot \left(\min_{\boldsymbol{x}' \in \mathbb{S}_{k,r}(\boldsymbol{x}_{k,r\eta}^{\leftarrow},10\epsilon)} \mathbb{P}\left[\left\|\nabla \log p_{k,0}(\boldsymbol{x}') - \mathbf{v}_{k-1,0}^{\leftarrow}(\boldsymbol{x}')\right\|^2 \leq \frac{10\epsilon}{960}\right]\right)^{n_{k,r}(10\epsilon) \cdot m_{k,r}(10\epsilon, \boldsymbol{x}_{k,r\eta}^{\leftarrow})} \quad (42)$$

Then, if $k \geq 1$, for each item of the latter term, supposing $10\epsilon' = \epsilon/96$, Lemma E.9 shows

$$\mathbb{P}\left[\left\|\nabla \log p_{k,0}(\boldsymbol{x}') - \mathbf{v}_{k-1,0}^{\leftarrow}(\boldsymbol{x}')\right\|^2 \leq \frac{\epsilon}{96}\right] = \mathbb{P}\left[\left\|\nabla \log p_{k,0}(\boldsymbol{x}') - \mathbf{v}_{k-1,0}^{\leftarrow}(\boldsymbol{x}')\right\|^2 \leq 10\epsilon'\right]$$

$$\geq (1-\delta) \cdot \left(\min_{\boldsymbol{x}'' \in \mathbb{S}_{k-1,0}(\boldsymbol{x}',10\epsilon')} \mathbb{P}\left[\left\|\nabla \log p_{k-1,0}(\boldsymbol{x}'') - \mathbf{v}_{k-2,0}^{\leftarrow}(\boldsymbol{x}'')\right\|^2 \leq \frac{\epsilon'}{96}\right]\right)^{n_{k,0}(10\epsilon') \cdot m_{k,r}(10\epsilon', \boldsymbol{x}')}$$

$$= (1-\delta) \cdot \left(\min_{\boldsymbol{x}'' \in \mathbb{S}_{k-1,0}(\boldsymbol{x}',\epsilon/96)} \mathbb{P}\left[\left\|\nabla \log p_{k-1,0}(\boldsymbol{x}'') - \mathbf{v}_{k-2,0}^{\leftarrow}(\boldsymbol{x}'')\right\|^2 \leq \frac{\epsilon}{96 \cdot 960}\right]\right)^{n_{k,0}(\epsilon/96) \cdot m_{k,0}(\epsilon/96, \boldsymbol{x}')}$$

Only particles that appear in the iteration will appear in powers of Eq 42. To simplify the notation, we set $Z$ as the upper bound of the norm of particles appear in Alg 2,

$$m_{k,r}(10\epsilon, \boldsymbol{x}) \leq m_{k,r}(10\epsilon) := C_m \cdot \frac{(d + M)^3 \cdot \max\{2 \log Z, 1\}}{(10\epsilon)^3}$$

$$\text{and} \quad u_{k,r}(\epsilon) := n_{k,r}(\epsilon) \cdot m_{k,r}(\epsilon).$$

Plugging this inequality into Eq 42, we have

$$\mathbb{P}\left[\left\|\nabla \log p_{k,S-r\eta}(\boldsymbol{x}_{k,r\eta}^{\leftarrow}) - \mathbf{v}_{k,r\eta}^{\leftarrow}(\boldsymbol{x}_{k,r\eta}^{\leftarrow})\right\|^2 \le 10\epsilon\right]$$

$$\ge (1-\delta)^{1+u_{k,r}(10\epsilon)} \cdot \left(\mathbb{P}\left[\left\|\nabla \log p_{k-1,0}(\boldsymbol{x}'') - \mathbf{v}_{k-2,0}^{\leftarrow}(\boldsymbol{x}'')\right\|^2 \le \frac{10\epsilon}{(960)^2}\right]\right)^{u_{k,r}(10\epsilon)\cdot u_{k,0}\left(\frac{\epsilon}{96}\right)}.$$

Using Lemma E.9 recursively, we will have

$$\mathbb{P}\left[\left\|\nabla \log p_{k,S-r\eta}(\boldsymbol{x}_{k,r\eta}^{\leftarrow}) - \mathbf{v}_{k,r\eta}^{\leftarrow}(\boldsymbol{x}_{k,r\eta}^{\leftarrow})\right\|^2 \le 10\epsilon\right]$$

$$\ge (1-\delta)^{1+u_{k,r}(10\epsilon)+u_{k,r}(10\epsilon)\cdot u_{k,0}\left(\frac{10\epsilon}{960}\right)+\ldots+u_{k,r}(10\epsilon)\cdot\prod_{i=k}^{2}u_{i,0}\left(\frac{10\epsilon}{960^{k-i+1}}\right)}$$

$$\left(\mathbb{P}\left[\left\|\nabla \log p_{0,0}(\boldsymbol{x}') - \tilde{\mathbf{v}}_{-1,0}(\boldsymbol{x}')\right\|^2 \le \frac{10\epsilon}{(960)^{k+1}}, \forall \boldsymbol{x}' \in \mathbb{R}^d\right]\right)^{u_{k,r}(10\epsilon)\cdot\prod_{i=k}^{1}u_{i,0}\left(\frac{10\epsilon}{960^{k-i+1}}\right)} \tag{43}$$

$$= (1-\delta)^{1+u_{k,r}(10\epsilon)+u_{k,r}(10\epsilon)\cdot u_{k,0}\left(\frac{10\epsilon}{960}\right)+\ldots+u_{k,r}(10\epsilon)\cdot\prod_{i=k}^{2}u_{i,0}\left(\frac{10\epsilon}{960^{k-i+1}}\right)}$$

$$\ge 1-\delta\cdot\left(1+u_{k,r}(10\epsilon)+u_{k,r}(10\epsilon)\cdot u_{k,0}\left(\frac{10\epsilon}{960}\right)+\ldots+u_{k,r}(10\epsilon)\cdot\prod_{i=k}^{2}u_{i,0}\left(\frac{10\epsilon}{960^{k-i+1}}\right)\right)$$

where the third inequality follows from the case $k=0$ in Lemma E.9 and the last inequality follows from union bound.

Then, we start to upper bound the coefficient of $\delta$. According to Lemma E.8 and Lemma E.9, it can be noted that the function $u_{k,r}(\cdot)$ is independent with $k$ and $r$. It is actually because we provide a union bound for the sample number $n_{k,r}$ and the iteration number $m_{k,r}$ when $(k,r) \in \mathbb{N}_{0,K-1} \times \mathbb{N}_{0,R-1}$. Therefore, the explicit form of the uniformed $u$ is defined as

$$u(10\epsilon) = \underbrace{C_n C_m \cdot (d+m_2^2)^4 \cdot \max\{d, \log(1/\delta^2)\} \cdot \max\{2\log Z, 1\}}_{\text{independent with } \epsilon} \cdot (10\epsilon)^{-5}$$

Then, we have

$$u\left(\frac{10\epsilon}{960}\right) = u(10\epsilon)\cdot 960^5 \quad \text{and} \quad u\left(\frac{10\epsilon}{960^i}\right) = u(10\epsilon)\cdot 960^{5i}.$$

Combining this result with Eq 43, we obtain

$$1+u_{k,r}(10\epsilon)+u_{k,r}(10\epsilon)\cdot u_{k,0}\left(\frac{10\epsilon}{960}\right)+\ldots+u_{k,r}(10\epsilon)\cdot\prod_{i=k}^{2}u_{i,0}\left(\frac{10\epsilon}{960^{k-i+1}}\right)$$

$$\le (k+1)\cdot u(10\epsilon)\cdot\prod_{i=k}^{2}u\left(\frac{10\epsilon}{960^{5(k-i+1)}}\right) = (k+1)\cdot u(10\epsilon)\cdot\prod_{i=k}^{2}\left(u(10\epsilon)\cdot 960^{k-i+1}\right)$$

$$= (k+1)\cdot 960^{2.5k(k-1)}\cdot u(10\epsilon)^k \le K\cdot 960^{2.5(K-1)(K-2)}\cdot u(10\epsilon)^{K-1}.$$

Considering that $K = 2/S \cdot \log[(Ld+M)/\epsilon]$, to bound RHS of the previous inequality, we have

$$\log\left(960^{2.5(K-1)(K-2)}\cdot u(10\epsilon)^{K-1}\right) = 2.5(K-1)(K-2)\log(960) + (K-1)\log(u(10\epsilon))$$

$$\le 2.5\cdot\log(960)\cdot\left(\frac{2}{S}\log\frac{Ld+M}{\epsilon}\right)^2 + \frac{2}{S}\log\frac{Ld+M}{\epsilon}\cdot\left(\log C_n C_m + 4\log(d+M) + \log d + \log\left(2\log\frac{1}{\delta}\right)\right.$$

$$\left.+ \log\left(2\max\left\{\log Z, \frac{1}{2}\right\}\right) + \log(10^{-5}) + 5\log\frac{1}{\epsilon}\right).$$

To make the result more clear, we set

$$C_{u,1} := \log(C_n C_m) + \log 2 + \log\left(2\max\left\{\log Z, \frac{1}{2}\right\}\right) - 5\log 10$$

which is independent with $d$, $\epsilon$ and $\delta$. Then, it has

$$\log\left(960^{2.5(K-1)(K-2)}\cdot u(10\epsilon)^{K-1}\right)$$

$$\le \frac{70}{S^2}\left(\log\frac{Ld+M}{\epsilon}\right)^2 + \frac{2}{S}\log\frac{Ld+M}{\epsilon}\cdot\left[C_{u,1} + 5\log(d+M) + \log\log\frac{1}{\delta} + 5\log\frac{1}{\epsilon}\right].$$

which means
$$960^{2.5(K-1)(K-2)} \cdot u(10\epsilon)^{K-1}$$
$$\leq \exp\left[\frac{70}{S^2}\left(\log\frac{Ld+M}{\epsilon}\right)^2 + \frac{2}{S}\log\frac{Ld+M}{\epsilon} \cdot \left(C_{u,1} + 5\log(d+M) + \log\log\frac{1}{\delta} + 5\log\frac{1}{\epsilon}\right)\right]$$
$$\leq \mathrm{pow}\left(\frac{Ld+M}{\epsilon}, \left(\left(\frac{70}{S^2} + \frac{10}{S}\right)\log\frac{Ld+M}{\epsilon} + \frac{2}{S}\log\log\frac{1}{\delta} + \frac{2C_{u,1}}{S}\right)\right)$$
(44)

where the last inequality suppose $L \geq 1$ as the previous settings. To simplify notation, we set
$$C_{u,2} := \frac{70}{S^2} + \frac{10}{S} \quad \text{and} \quad C_{u,3} := \frac{2C_{u,1}}{S}.$$

Plugging this result into Eq 43, we have
$$\mathbb{P}\left[\left\|\nabla\log p_{k,S-r\eta}(\boldsymbol{x}_{k,r\eta}^{\leftarrow}) - \mathbf{v}_{k,r\eta}^{\leftarrow}(\boldsymbol{x}_{k,r\eta}^{\leftarrow})\right\|^2 \leq 10\epsilon\right]$$
$$\geq 1 - \delta \cdot K \cdot \mathrm{pow}\left(\frac{Ld+M}{\epsilon}, C_{u,2}\log\frac{Ld+M}{\epsilon} + \frac{2}{S}\log\log\frac{1}{\delta} + C_{u,3}\right).$$
(45)

With these conditions, we can lower bound score estimation errors along Alg 2. That is
$$\mathbb{P}\left[\bigcap_{\substack{k\in\mathbb{N}_{0,K-1}\\r\in\mathbb{N}_{0,R-1}}}\left\|\nabla\log p_{k,S-r\eta}(\boldsymbol{x}_{k,r\eta}^{\leftarrow}) - \mathbf{v}_{k,r\eta}^{\leftarrow}(\boldsymbol{x}_{k,r\eta}^{\leftarrow})\right\|^2 \leq 10\epsilon\right]$$
$$= \prod_{\substack{k\in\mathbb{N}_{0,K-1}\\r\in\mathbb{N}_{0,R-1}}}\mathbb{P}\left[\left\|\nabla\log p_{k,S-r\eta}(\boldsymbol{x}_{k,r\eta}^{\leftarrow}) - \mathbf{v}_{k,r\eta}^{\leftarrow}(\boldsymbol{x}_{k,r\eta}^{\leftarrow})\right\|^2 \leq 10\epsilon\right]$$

where the first inequality establishes because the random variables, $\mathbf{v}_{k,r\eta}^{\leftarrow}$, are independent for each $(k,r)$ pair. By introducing Eq 45, we have
$$\prod_{\substack{k\in\mathbb{N}_{0,K-1}\\r\in\mathbb{N}_{0,R-1}}}\mathbb{P}\left[\left\|\nabla\log p_{k,S-r\eta}(\boldsymbol{x}_{k,r\eta}^{\leftarrow}) - \mathbf{v}_{k,r\eta}^{\leftarrow}(\boldsymbol{x}_{k,r\eta}^{\leftarrow})\right\|^2 \leq 10\epsilon\right]$$
$$\geq \left(1 - \delta \cdot K \cdot \mathrm{pow}\left(\frac{Ld+M}{\epsilon}, C_{u,2}\log\frac{Ld+M}{\epsilon} + \frac{2}{S}\log\log\frac{1}{\delta} + C_{u,3}\right)\right)^{KR}$$
$$\geq 1 - \delta \cdot K^2 R \cdot \mathrm{pow}\left(\frac{Ld+M}{\epsilon}, C_{u,2}\log\frac{Ld+M}{\epsilon} + \frac{2}{S}\log\log\frac{1}{\delta} + C_{u,3}\right)$$
$$= 1 - \delta \cdot \frac{4(d+M)}{SC_\eta\epsilon}\left(\log\frac{Ld+M}{\epsilon}\right)^2 \cdot \mathrm{pow}\left(\frac{Ld+M}{\epsilon}, C_{u,2}\log\frac{Ld+M}{\epsilon} + \frac{2}{S}\log\log\frac{1}{\delta} + C_{u,3}\right)$$
(46)

where the first inequality follows from Eq 45 and the second inequality follows from the union bound, and the last inequality follows from the combination of the choice of the step size, i.e., $\eta = C_1(d+M)^{-1}\epsilon$ and the definition of $K$ and $R$, i.e.,
$$K = \frac{T}{S} = \frac{2}{S}\log\frac{C_0}{\epsilon}, \quad R = \frac{S}{\eta} = \frac{S(d+M)}{C_\eta\epsilon}.$$

It means when $\delta$ is small enough, we can control the recursive error with a high probability, i.e.,
$$\prod_{\substack{k\in\mathbb{N}_{0,K-1}\\r\in\mathbb{N}_{0,R-1}}}\mathbb{P}\left[\left\|\nabla\log p_{k,S-r\eta}(\boldsymbol{x}_{k,r\eta}^{\leftarrow}) - \mathbf{v}_{k,r\eta}^{\leftarrow}(\boldsymbol{x}_{k,r\eta}^{\leftarrow})\right\|^2 \leq 10\epsilon\right] \geq 1 - \epsilon.$$
(47)

Compared with Eq 46, Eq 47 can be achieved by requiring
$$\underbrace{\frac{4(d+M)}{SC_\eta\epsilon}\left(\log\frac{Ld+M}{\epsilon}\right)^2 \cdot \mathrm{pow}\left(\frac{Ld+M}{\epsilon}, C_{u,2}\log\frac{Ld+M}{\epsilon} + C_{u,3}\right)}_{\text{defined as } C_B} \cdot \delta\,\mathrm{pow}\left(\frac{Ld+M}{\epsilon}, \frac{2}{S}\log\log\frac{1}{\delta}\right) \leq \epsilon,$$

which can be obtained by requiring

$$C_B \delta (-\log \delta)^{\frac{2}{S} \log \frac{Ld+M}{\epsilon}} \leq \epsilon \quad \Leftrightarrow \quad (-\log \delta)^{\frac{2}{S} \log \frac{Ld+M}{\epsilon}} \leq \frac{\epsilon}{C_B \delta}$$

$$\Leftrightarrow \quad \frac{2}{S} \log \frac{Ld+M}{\epsilon} \cdot \log \log \frac{1}{\delta} \leq \log \frac{\epsilon}{C_B \delta} \tag{48}$$

We suppose $\delta = \epsilon/C_B \cdot a^{-2/S \cdot \log((Ld+M)/\epsilon)}$ and the last inequality of Eq 48 becomes

$$\text{LHS} = \frac{2}{S} \log \frac{Ld+M}{\epsilon} \cdot \log \left[ \log \frac{C_B}{\epsilon} + \frac{2}{S} \log \frac{Ld+M}{\epsilon} \cdot \log a \right] \leq \frac{2}{S} \log \frac{Ld+M}{\epsilon} \cdot \log a = \text{RHS},$$

which is hold if we require

$$a \geq \max \left\{ \frac{2C_B}{\epsilon}, \left( \frac{Ld+M}{\epsilon} \right)^{2/S}, 1 \right\}.$$

Because in this condition, we have

$$\log \frac{C_B}{\epsilon} + \frac{2}{S} \log \frac{Ld+M}{\epsilon} \cdot \log a \leq \log \frac{a}{2} + (\log a)^2 \leq \frac{2a}{5} + \frac{3a}{5} = a \quad \text{when} \quad a \geq 1,$$

where the first inequality follows from the monotonicity of function $\log(\cdot)$. Therefore, we have

$$\log \left[ \log \frac{C_B}{\epsilon} + \frac{2}{S} \log \frac{Ld+M}{\epsilon} \cdot \log a \right] \leq \log a$$

and Eq 48 establishes. Without loss of generality, we suppose $3C_B/\epsilon$ dominates the lower bound of $a$. Hence, the choice of $\delta$ can be determined.

After determining the choice of $\delta$, the only problem left is the gradient complexity of Alg 2. The number of gradients calculated in Alg 2 is equal to the number of calls for $\tilde{\mathbf{v}}_{-1,0}$. According to Eq 43, we can easily note that the number of calls of $\tilde{\mathbf{v}}_{-1,0}$ is

$$u_{k,r}(10\epsilon) \cdot \prod_{i=k}^{1} u_{i,0} \left( \frac{10\epsilon}{960^{k-i+1}} \right) = u(10\epsilon) \prod_{i=k}^{1} u \left( \frac{10\epsilon}{960^{k-i+1}} \right)$$

for each $(k, r)$ pair. We can upper bound RHS of the previous equation as

$$u(10\epsilon) \prod_{i=k}^{1} u \left( \frac{10\epsilon}{960^{k-i+1}} \right) = u(10\epsilon) \cdot \prod_{i=k}^{2} \left( u(10\epsilon) \cdot 960^{k-i+1} \right)$$

$$= 960^{2.5k(k-1)} \cdot u(10\epsilon)^k \leq 960^{2.5(K-1)(K-2)} \cdot u(10\epsilon)^{K-1}.$$

Combining this result with the total number of $(k, r)$ pair, i.e., $T/\eta$, the total gradient complexity can be relaxed as

$$\frac{T}{\eta} \cdot 960^{2.5k(k-1)} \cdot u(10\epsilon)^k \leq K^2 R \cdot 960^{2.5(K-1)(K-2)} \cdot u(10\epsilon)^{K-1}$$

$$\leq \frac{4(d+M)}{SC_\eta \epsilon} \left( \log \frac{Ld+M}{\epsilon} \right)^2 \cdot \text{pow} \left( \frac{Ld+M}{\epsilon}, C_{u,2} \log \frac{Ld+M}{\epsilon} + \frac{2}{S} \log \log \frac{1}{\delta} + C_{u,3} \right)$$

$$= C_B \cdot (-\log \delta)^{\frac{2}{S} \log \frac{Ld+M}{\epsilon}} \leq \frac{\epsilon}{\delta} = C_B \cdot a^{\frac{2}{S} \log \frac{Ld+M}{\epsilon}} \tag{49}$$

where the first inequality follows from the fact $T/\eta = KR$, the second inequality follows from the combination of the choice of the step size, i.e., $\eta = C_1(d+M)^{-1}\epsilon$ and the definition of $K$ and $R$, i.e.,

$$K = \frac{T}{S} = \frac{2}{S} \log \frac{C_0}{\epsilon}, \quad R = \frac{S}{\eta} = \frac{S(d+M)}{C_1 \epsilon}$$

and the last inequality follows from 48. Choosing $a$ as its lower bound, i.e., $2C_B/\epsilon$, RHS of Eq 49 satisfies

$$
\begin{aligned}
C_B \cdot a^{\frac{2}{S} \log \frac{Ld+M}{\epsilon}} = C_B \cdot \left(\frac{2C_B}{\epsilon}\right)^{\frac{2}{S} \log \frac{Ld+M}{\epsilon}} &\leq \left(\frac{2C_B}{\epsilon}\right)^{\frac{4}{S} \log \frac{Ld+M}{\epsilon}} \\
&\leq \mathrm{pow}\left(\frac{8(d+M)}{SC_\eta \epsilon^2} \cdot \left(\log \frac{Ld+M}{\epsilon}\right)^2, \frac{4}{S} \log \frac{Ld+M}{\epsilon}\right) \\
&\quad \cdot \mathrm{pow}\left(\frac{Ld+M}{\epsilon}, \frac{4C_{u,2}}{S}\left(\log \frac{Ld+M}{\epsilon}\right)^2 + \frac{4C_{u,3}}{S}\left(\log \frac{Ld+M}{\epsilon}\right)\right) \\
&= \exp\left[\mathcal{O}\left(\left(\log \frac{Ld+M}{\epsilon}\right)^3\right)\right].
\end{aligned}
\tag{50}
$$

If we consider the effect of the norm of particles since we have

$$
C_{u,3} = \frac{2C_{u,1}}{S} = \mathcal{O}\left(\left(\max\left\{\log\log Z^2, 1\right\}\right)\right),
$$

Combining this result with Eq 50, the proof is completed. $\qquad \square$

**Lemma E.11.** *Under the notation in Section A, suppose the step size satisfy $\eta = C_\eta(d+M)^{-1}\epsilon$, we have*

$$
\mathbb{P}\left[\bigcap_{\substack{k \in \mathbb{N}_{0,K-1} \\ r \in \mathbb{N}_{0,R-1}}} \left\|\nabla \log p_{k,S-r\eta}(\overleftarrow{\boldsymbol{x}}_{k,r\eta}) - \mathbf{v}_{k,r\eta}^{\leftarrow}(\overleftarrow{\boldsymbol{x}}_{k,r\eta})\right\|^2 \leq 10\epsilon\right] \geq 1 - \delta'
$$

*with Alg 1 by properly choosing the number for mean estimations and ULA iterations. The total gradient complexity will be at most*

$$
\exp\left(\mathcal{O}\left(\max\left\{\left(\log \frac{Ld+M}{\epsilon}\right)^3, \log \frac{Ld+M}{\epsilon} \cdot \log \frac{1}{\delta'}\right\} \cdot \max\left\{\log\log Z^2, 1\right\}\right)\right),
$$

*where $Z$ is the maximal norm of particles appeared in Alg 2.*

*Proof.* In this lemma, we follow the same proof roadmap as that shown in Lemma E.10. According to Eq 46, we have

$$
\prod_{\substack{k \in \mathbb{N}_{0,K-1} \\ r \in \mathbb{N}_{0,R-1}}} \mathbb{P}\left[\left\|\nabla \log p_{k,S-r\eta}(\overleftarrow{\boldsymbol{x}}_{k,r\eta}) - \mathbf{v}_{k,r\eta}^{\leftarrow}(\overleftarrow{\boldsymbol{x}}_{k,r\eta})\right\|^2 \leq 10\epsilon\right]
$$

$$
\geq 1 - \delta \cdot \frac{4(d+M)}{SC_\eta \epsilon}\left(\log \frac{Ld+M}{\epsilon}\right)^2 \cdot \mathrm{pow}\left(\frac{Ld+M}{\epsilon}, C_{u,2}\log \frac{Ld+M}{\epsilon} + \frac{2}{S}\log\log \frac{1}{\delta} + C_{u,3}\right)
$$

where the parameter $\delta$ satisfies Lemma E.8 under certain conditions. It means we can control the recursive error with a high probability, i.e.,

$$
\prod_{\substack{k \in \mathbb{N}_{0,K-1} \\ r \in \mathbb{N}_{0,R-1}}} \mathbb{P}\left[\left\|\nabla \log p_{k,S-r\eta}(\overleftarrow{\boldsymbol{x}}_{k,r\eta}) - \mathbf{v}_{k,r\eta}^{\leftarrow}(\overleftarrow{\boldsymbol{x}}_{k,r\eta})\right\|^2 \leq 10\epsilon\right] \geq 1 - \delta'.
\tag{51}
$$

when $\delta$ satisfies

$$
\underbrace{\frac{4(d+M)}{SC_\eta \epsilon}\left(\log \frac{Ld+M}{\epsilon}\right)^2 \cdot \mathrm{pow}\left(\frac{Ld+M}{\epsilon}, C_{u,2}\log \frac{Ld+M}{\epsilon} + C_{u,3}\right)}_{\text{defined as } C_B} \cdot \delta \mathrm{pow}\left(\frac{Ld+M}{\epsilon}, \frac{2}{S}\log\log \frac{1}{\delta}\right) \leq \delta'.
$$

We can reformulate the above inequality as follows.

$$
\begin{aligned}
C_B \delta(-\log \delta)^{\frac{2}{S} \log \frac{Ld+M}{\epsilon}} \leq \delta' &\quad\Leftrightarrow\quad (-\log \delta)^{\frac{2}{S} \log \frac{Ld+M}{\epsilon}} \leq \frac{\delta'}{C_B \delta} \\
&\Leftrightarrow\quad \frac{2}{S}\log \frac{Ld+M}{\epsilon} \cdot \log\log \frac{1}{\delta} \leq \log \frac{\delta'}{C_B \delta}.
\end{aligned}
\tag{52}
$$

By requiring $\delta = \delta'/C_B \cdot a^{-2/S \cdot \log((Ld+M)/\epsilon)}$, the last inequality of the above can be written as

$$\text{LHS} = \frac{2}{S} \log \frac{Ld+M}{\epsilon} \cdot \log \left[ \log \frac{C_B}{\delta'} + \frac{2}{S} \log \frac{Ld+M}{\epsilon} \cdot \log a \right] \le \frac{2}{S} \log \frac{Ld+M}{\epsilon} \cdot \log a = \text{RHS},$$

when the choice of $a$ satisfies

$$a \ge \max \left\{ \frac{2C_B}{\delta'}, \left( \frac{Ld+M}{\epsilon} \right)^{2/S}, 1 \right\}. \tag{53}$$

Since we have

$$\log \frac{C_B}{\delta'} + \frac{2}{S} \log \frac{Ld+M}{\epsilon} \cdot \log a \le \log \frac{a}{2} + (\log a)^2 \le \frac{2a}{5} + \frac{3a}{5} = a \quad \text{when} \quad a \ge 1,$$

where the first inequality follows from the monotonicity of function $\log(\cdot)$. Then, it has

$$\log \left[ \log \frac{C_B}{\delta'} + \frac{2}{S} \log \frac{Ld+M}{\epsilon} \cdot \log a \right] \le \log a$$

and Eq 52 establishes.

To achieve the accurate score estimation with a high probability shown in Eq 51, the total gradient complexity will be

$$\frac{T}{\eta} \cdot 960^{2.5k(k-1)} \cdot u(10\epsilon)^k \le C_B \cdot a^{\frac{2}{S} \log \frac{Ld+M}{\epsilon}}$$

shown in Eq 49. Plugging the choice of $a$ (Eq 53) into the above inequality, we have

$$C_B \cdot a^{\frac{2}{S} \log \frac{Ld+M}{\epsilon}} \le C_B \cdot \max \left\{ \text{pow}\left( \frac{2C_B}{\delta'}, \frac{2}{S} \log \frac{Ld+M}{\epsilon} \right), \text{pow}\left( \frac{Ld+M}{\epsilon}, \frac{4}{S^2} \log \frac{Ld+M}{\epsilon} \right) \right\}$$

$$\le \max \left\{ \underbrace{\text{pow}\left( \frac{2C_B}{\delta'}, \frac{4}{S} \log \frac{Ld+M}{\epsilon} \right)}_{\text{Term Comp.1}}, \underbrace{C_B \cdot \text{pow}\left( \frac{Ld+M}{\epsilon}, \frac{4}{S^2} \log \frac{Ld+M}{\epsilon} \right)}_{\text{Term Comp.2}} \right\}$$

It can be easily noted that Term Comp.2 will be dominated by Term Comp.1. Then, we provide the upper bound of Comp.1 as

$$\log(\text{Comp.1}) = \frac{4}{S} \log \frac{Ld+M}{\epsilon} \cdot \left( \log 2C_B + \log(1/\delta') \right)$$

$$= \frac{4}{S} \log \frac{Ld+M}{\epsilon} \cdot \left( \log \frac{8}{SC_\eta} + \log \frac{d+M}{\epsilon} + 2 \log \log \frac{Ld+M}{\epsilon} \right.$$

$$\left. + \log \frac{Ld+M}{\epsilon} \cdot \left( C_{u,2} \log \frac{Ld+M}{\epsilon} + C_{u,3} \right) + \log(1/\delta') \right)$$

$$= \mathcal{O}\left( \max \left\{ \left( \log \frac{Ld+M}{\epsilon} \right)^3, \log \frac{Ld+M}{\epsilon} \cdot \log \frac{1}{\delta'} \right\} \right),$$

which means

$$C_B \cdot a^{\frac{2}{S} \log \frac{Ld+M}{\epsilon}} \le \exp \left( \mathcal{O}\left( \max \left\{ \left( \log \frac{Ld+M}{\epsilon} \right)^3, \log \frac{Ld+M}{\epsilon} \cdot \log \frac{1}{\delta'} \right\} \right) \right).$$

Hence, the proof is completed. □

## F    AUXILIARY LEMMAS

### F.1    THE CHAIN RULE OF KL DIVERGENCE

**Lemma F.1** (Lemma 6 in Chen et al. (2023a))**.** *Consider the following two Itô processes,*

$$d\mathbf{x}_t = \boldsymbol{f}_1(\mathbf{x}_t, t)dt + g(t)dB_t, \quad \mathbf{x}_0 = \boldsymbol{a},$$
$$d\mathbf{y}_t = \boldsymbol{f}_2(\mathbf{y}_t, t)dt + g(t)dB_t, \quad \mathbf{y}_0 = \boldsymbol{a},$$

*where $\boldsymbol{f}_1, \boldsymbol{f}_2 \colon \mathbb{R}^d \to \mathbb{R}$ and $g \colon \mathbb{R} \to \mathbb{R}$ are continuous functions and may depend on $\boldsymbol{a}$. We assume the uniqueness and regularity conditions:*

- *The two SDEs have unique solutions.*

- $\mathbf{x}_t, \mathbf{y}_t$ *admit densities* $p_t, q_t \in C^2(\mathbb{R}^d)$ *for* $t > 0$.

*Define the relative Fisher information between* $p_t$ *and* $q_t$ *by*

$$\mathrm{FI}\left(p_t \| q_t\right) := \int p_t(\boldsymbol{x}) \left\| \nabla \log \frac{p_t(\boldsymbol{x})}{q_t(\boldsymbol{x})} \right\|^2 \mathrm{d}\boldsymbol{x}.$$

*Then for any* $t > 0$, *the evolution of* $\mathrm{KL}\left(p_t \| q_t\right)$ *is given by*

$$\frac{\partial}{\partial t} \mathrm{KL}\left(p_t \| q_t\right) = -\frac{g^2(t)}{2} \mathrm{FI}\left(p_t \| q_t\right) + \mathbb{E}\left[\left\langle \boldsymbol{f}_1(\mathbf{x}_t, t) - \boldsymbol{f}_2(\mathbf{x}_t, t), \nabla \log \frac{p(\mathbf{x}_t)}{q(\mathbf{x}_t)} \right\rangle\right].$$

Lemma F.1 is applied to show the KL convergence between the underlying distribution of the SDEs that have the same diffusion term and a bounded difference between their drift terms.

**Lemma F.2** (Lemma 7 in Chen et al. (2023a)). *Under the notation in Section A, for* $k \in \mathbb{N}_{0,K-1}$ *and* $r \in \mathbb{N}_{0,R-1}$, *consider the reverse SDE starting from* $\mathbf{x}_{k,r\eta}^{\leftarrow} = \boldsymbol{a}$

$$\mathrm{d}\hat{\mathbf{x}}_{k,t} = \left[\hat{\mathbf{x}}_{k,t} + 2\nabla \log p_{k,S-t}(\hat{\mathbf{x}}_{k,t})\right]\mathrm{d}t + \sqrt{2}\mathrm{d}B_t, \quad \mathbf{x}_{k,r\eta}^{\leftarrow} = \boldsymbol{a} \tag{54}$$

*and its discrete approximation*

$$\mathrm{d}\mathbf{x}_{k,t}^{\leftarrow} = \left[\mathbf{x}_{k,t}^{\leftarrow} + 2\mathbf{v}_{k,r\eta}^{\leftarrow}\left(\mathbf{x}_{k,r\eta}^{\leftarrow}\right)\right]\mathrm{d}t + \sqrt{2}\mathrm{d}B_t, \quad \mathbf{x}_{k,r\eta}^{\leftarrow} = \boldsymbol{a} \tag{55}$$

*for time* $t \in [k\eta, (k+1)\eta]$. *Let* $\hat{p}_{k,t|r\eta}$ *be the density of* $\hat{\mathbf{x}}_{k,t}$ *given* $\hat{\mathbf{x}}_{k,r\eta}$ *and* $p_{k,t|r\eta}^{\leftarrow}$ *be the density of* $\mathbf{x}_{k,t}^{\leftarrow}$ *given* $\mathbf{x}_{k,r\eta}^{\leftarrow}$. *Then, we have*

- *For any* $\boldsymbol{a} \in \mathbb{R}^d$, *the two processes satisfy the uniqueness and regularity condition stated in Lemma F.1, which means SDE 54 and SDE 55 have unique solutions and* $\hat{p}_{k,t|r\eta}(\cdot|\boldsymbol{a}), p_{k,t|r\eta}^{\leftarrow}(\cdot|\boldsymbol{a}) \in C^2(\mathbb{R}^d)$ *for* $t \in (r\eta, (r+1)\eta]$.

- *For a.e.,* $\boldsymbol{a} \in \mathbb{R}^d$, *we have*

$$\lim_{t \to r\eta_+} \mathrm{KL}\left(\hat{p}_{k,t|r\eta}(\cdot|\boldsymbol{a}) \| \tilde{p}_{k,t|r\eta}(\cdot|\boldsymbol{a})\right) = 0.$$

**Lemma F.3** (Variant of Proposition 8 in Chen et al. (2023a)). *Under the notation in Section A and Algorithm 2, we have*

$$\begin{aligned}
\mathrm{KL}\left(\hat{p}_{0,S} \| p_{0,S}^{\leftarrow}\right) \leq &\mathrm{KL}\left(\hat{p}_{K-1,0} \| p_{K-1,0}^{\leftarrow}\right) \\
&+ \sum_{k=0}^{K-1}\sum_{r=0}^{R-1}\int_0^\eta \mathbb{E}_{(\hat{\mathbf{x}}_{k,t+r\eta}, \hat{\mathbf{x}}_{k,r\eta})}\left[\left\|\nabla \log p_{k,S-(t+r\eta)}(\hat{\mathbf{x}}_{k,t+r\eta}) - \mathbf{v}_{k,r\eta}^{\leftarrow}(\hat{\mathbf{x}}_{k,r\eta})\right\|^2\right]\mathrm{d}t.
\end{aligned}$$

*Proof.* Under the notation in Section A, for $k \in \mathbb{N}_{0,K-1}$ and $r \in \mathbb{N}_{0,R-1}$, let $\hat{p}_{k,t|r\eta}$ be the density of $\hat{\mathbf{x}}_{k,t}$ given $\hat{\mathbf{x}}_{k,r\eta}$ and $p_{k,t|r\eta}^{\leftarrow}$ be the density of $\mathbf{x}_{k,t}^{\leftarrow}$ given $\mathbf{x}_{k,r\eta}^{\leftarrow}$. According to Lemma F.2 and Lemma F.1, for any $\mathbf{x}_{k,r\eta}^{\leftarrow} = \boldsymbol{a}$, we have

$$\frac{\mathrm{d}}{\mathrm{d}t}\mathrm{KL}\left(\hat{p}_{k,t|r\eta}(\cdot|\boldsymbol{a}) \| p_{k,t|r\eta}^{\leftarrow}(\cdot|\boldsymbol{a})\right)$$

$$= -\mathrm{FI}\left(\hat{p}_{k,t|r\eta}(\cdot|\boldsymbol{a}) \| p_{k,t|r\eta}^{\leftarrow}(\cdot|\boldsymbol{a})\right) + 2\mathbb{E}_{\mathbf{x} \sim \hat{p}_{k,t|r\eta}(\cdot|\boldsymbol{a})}\left[\left\langle \nabla \log p_{k,S-t}(\mathbf{x}) - \mathbf{v}_{k,r\eta}^{\leftarrow}(\boldsymbol{a}), \nabla \log \frac{\hat{p}_{k,t|r\eta}(\mathbf{x}|\boldsymbol{a})}{p_{k,t|r\eta}^{\leftarrow}(\mathbf{x}|\boldsymbol{a})}\right\rangle\right]$$

$$\leq \mathbb{E}_{\mathbf{x} \sim \hat{p}_{k,t|r\eta}(\cdot|\boldsymbol{a})}\left[\left\|\nabla \log p_{k,S-t}(\mathbf{x}) - \mathbf{v}_{k,r\eta}^{\leftarrow}(\boldsymbol{a})\right\|^2\right].$$

Due to Lemma F.2, for any $\boldsymbol{a} \in \mathbb{R}^d$, we have

$$\lim_{t \to r\eta_+} \mathrm{KL}\left(\hat{p}_{k,t|r\eta}(\cdot|\boldsymbol{a}) \| p_{k,t|r\eta}^{\leftarrow}(\cdot|\boldsymbol{a})\right) = 0,$$

which implies

$$\mathrm{KL}\left(\hat{p}_{k,t|r\eta}(\cdot|\boldsymbol{a})\|p_{k,t|r\eta}^{\leftarrow}(\cdot|\boldsymbol{a})\right) = \int_{r\eta}^{t}\mathbb{E}_{\mathbf{x}\sim\hat{p}_{\tau|r\eta}(\cdot|\boldsymbol{a})}\left[\left\|\nabla\log p_{k,S-\tau}(\mathbf{x}) - \mathbf{v}_{k,r\eta}^{\leftarrow}(\boldsymbol{a})\right\|^2\right]\mathrm{d}\tau.$$

Integrating both sides of the equation, we have

$$\mathbb{E}_{\hat{\mathbf{x}}_{k,r\eta}\sim\hat{p}_{k,r\eta}}\left[\mathrm{KL}\left(\hat{p}_{k,t|r\eta}(\cdot|\hat{\mathbf{x}}_{k,r\eta})\|p_{k,t|r\eta}^{\leftarrow}(\cdot|\hat{\mathbf{x}}_{k,r\eta})\right)\right] \le \int_{r\eta}^{t}\mathbb{E}\left[\left\|\nabla\log p_{k,S-\tau}(\hat{\mathbf{x}}_{k,\tau}) - \mathbf{v}_{k,r\eta}^{\leftarrow}(\hat{\mathbf{x}}_{k,r\eta})\right\|^2\right]\mathrm{d}\tau.$$

According to the chain rule of KL divergence Chen et al. (2023a), we have

$$\mathrm{KL}\left(\hat{p}_{k,(r+1)\eta}\|p_{k,(r+1)\eta}^{\leftarrow}\right)$$
$$\le \mathrm{KL}\left(\hat{p}_{k,r\eta}\|p_{k,r\eta}^{\leftarrow}\right) + \mathbb{E}_{\hat{\mathbf{x}}_{k,r\eta}\sim\hat{p}_{k,r\eta}}\left[\mathrm{KL}\left(\hat{p}_{k,(r+1)\eta|r\eta}(\cdot|\hat{\mathbf{x}}_{k,r\eta})\|p_{k,(r+1)\eta|r\eta}^{\leftarrow}(\cdot|\hat{\mathbf{x}}_{k,r\eta})\right)\right]$$
$$\le \mathrm{KL}\left(\hat{p}_{k,r\eta}\|p_{k,r\eta}^{\leftarrow}\right) + \int_{0}^{\eta}\mathbb{E}_{(\hat{\mathbf{x}}_{k,t+r\eta},\hat{\mathbf{x}}_{k,r\eta})}\left[\left\|\nabla\log p_{k,S-(t+r\eta)}(\hat{\mathbf{x}}_{k,t+r\eta}) - \mathbf{v}_{k,r\eta}^{\leftarrow}(\hat{\mathbf{x}}_{k,r\eta})\right\|^2\right]\mathrm{d}t.$$

Summing over $r \in \{0, 1, \ldots, R-1\}$, it has

$$\mathrm{KL}\left(\hat{p}_{k,R\eta}\|p_{k,R\eta}^{\leftarrow}\right) \le \mathrm{KL}\left(\hat{p}_{k,0}\|p_{k,0}^{\leftarrow}\right) + \sum_{r=0}^{R-1}\int_{0}^{\eta}\mathbb{E}_{(\hat{\mathbf{x}}_{k,t+r\eta},\hat{\mathbf{x}}_{k,r\eta})}\left[\left\|\nabla\log p_{k,S-(t+r\eta)}(\hat{\mathbf{x}}_{k,t+r\eta}) - \mathbf{v}_{k,r\eta}^{\leftarrow}(\hat{\mathbf{x}}_{k,r\eta})\right\|^2\right]\mathrm{d}t.$$

Similarly, by considering all segments, we have

$$\mathrm{KL}\left(\hat{p}_{0,S}\|p_{0,S}^{\leftarrow}\right) \le \mathrm{KL}\left(\hat{p}_{K-1,0}\|p_{K-1,0}^{\leftarrow}\right)$$
$$+ \sum_{k=0}^{K-1}\sum_{r=0}^{R-1}\int_{0}^{\eta}\mathbb{E}_{(\hat{\mathbf{x}}_{k,t+r\eta},\hat{\mathbf{x}}_{k,r\eta})}\left[\left\|\nabla\log p_{k,S-(t+r\eta)}(\hat{\mathbf{x}}_{k,t+r\eta}) - \mathbf{v}_{k,r\eta}^{\leftarrow}(\hat{\mathbf{x}}_{k,r\eta})\right\|^2\right]\mathrm{d}t.$$

$\square$

**Lemma F.4** (Variant of Lemma 10 in Cheng & Bartlett (2018)). *Suppose* $-\log p_*$ *is* $m$-*strongly convex function, for any distribution with density function* $p$, *we have*

$$\mathrm{KL}\left(p\|p_*\right) \le \frac{1}{2m}\int p(\boldsymbol{x})\left\|\nabla\log\frac{p(\boldsymbol{x})}{p_*(\boldsymbol{x})}\right\|^2\mathrm{d}\boldsymbol{x}.$$

*By choosing* $p(\boldsymbol{x}) = g^2(\boldsymbol{x})p_*(\boldsymbol{x})/\mathbb{E}_{p_*}\left[g^2(\mathbf{x})\right]$ *for the test function* $g\colon\mathbb{R}^d\to\mathbb{R}$ *and* $\mathbb{E}_{p_*}\left[g^2(\mathbf{x})\right] < \infty$, *we have*

$$\mathbb{E}_{p_*}\left[g^2\log g^2\right] - \mathbb{E}_{p_*}\left[g^2\right]\log\mathbb{E}_{p_*}\left[g^2\right] \le \frac{2}{m}\mathbb{E}_{p_*}\left[\|\nabla g\|^2\right],$$

*which implies* $p_*$ *satisfies* $m$-*log-Sobolev inequality.*

**Lemma F.5.** *(Corollary 3.1 in Chafaï (2004)) If* $\nu, \tilde{\nu}$ *satisfy LSI with constants* $\alpha, \tilde{\alpha} > 0$, *respectively, then* $\nu * \tilde{\nu}$ *satisfies LSI with constant* $(\frac{1}{\alpha} + \frac{1}{\tilde{\alpha}})^{-1}$.

**Lemma F.6** (Lemma 16 in Vempala & Wibisono (2019)). *Suppose a probability distribution* $p$ *satisfies LSI with constant* $\mu > 0$. *Let a map* $T\colon\mathbb{R}^d\to\mathbb{R}^d$, *be a differentiable* $L$-*Lipschitz map. Then,* $\tilde{p} = T_{\#}p$ *satisfies LSI with constant* $\mu/L^2$

**Lemma F.7** (Lemma 17 in Vempala & Wibisono (2019)). *Suppose a probability distribution* $p$ *satisfies LSI with a constant* $\mu$. *For any* $t > 0$, *the probability distribution* $\tilde{p}_t = p * \mathcal{N}(\mathbf{0}, t\boldsymbol{I})$ *satisfies LSI with the constant* $(\mu^{-1} + t)^{-1}$.

**Lemma F.8** (Theorem 8 in Vempala & Wibisono (2019)). *Suppose* $p \propto \exp(-f)$ *is* $\mu$ *strongly log concave and* $L$-*smooth. If we conduct ULA with the step size satisfying* $\eta \le 1/L$, *then, for any iteration number, the underlying distribution of the output particle satisfies LSI with a constant larger than* $\mu/2$.

*Proof.* Suppose we run ULA from $\mathbf{x}_0 \sim p_0$ to $\mathbf{x}_k \sim p_k$ where the LSI constant of $p_k$ is denoted as $\mu_k$. When the step size of ULA satisfies $0 < \eta \le 1/L$, due to the strong convexity of $p$, the map $\boldsymbol{x} \mapsto \boldsymbol{x} - \eta\nabla f(\boldsymbol{x})$ is $(1 - \eta\mu)$-Lipschitz. Combining the LSI property of $p_k$ and Lemma F.6, the

distribution of $\mathbf{x}_k - \eta \nabla f(\mathbf{x}_k)$ satisfies LSI with a constant $\mu_k/(1-\eta\mu)^2$. Then, by Lemma F.7, $\mathbf{x}_{k+1} = \mathbf{x}_k - \eta \nabla f(\mathbf{x}_k) + \sqrt{2\eta}\mathcal{N}(0, \boldsymbol{I}) \sim p_{k+1}$ satisfies $\mu_{k+1}$-LSI with

$$\frac{1}{\mu_{k+1}} \leq \frac{(1-\eta\mu)^2}{\mu_k} + 2\eta.$$

For any $k$, if there is $\mu_k \geq \mu/2$, with the setting of $\eta$, i.e., $\eta \leq 1/L \leq 1/\mu$, then

$$\frac{1}{\mu_{k+1}} \leq \frac{(1-\eta\mu)^2}{\mu/2} + 2\eta = \frac{2}{\mu} - 2\eta(1-\eta\mu) \leq \frac{2}{\mu}.$$

It means for any $k' > k$, we have $\mu_{k'} \geq \mu/2$. By requiring the LSI constant of initial distribution, i.e., $p_0$ to satisfy $\mu_0 \geq \mu/2$, we have the underlying distribution of the output particle satisfies LSI with a constant larger than $\mu/2$. Hence, the proof is completed. $\qquad\square$

**Lemma F.9.** *If $\nu$ satisfies a log-Sobolev inequality with log-Sobolev constant $\mu$ then every $1$-Lipschitz function $f$ is integrable with respect to $\nu$ and satisfies the concentration inequality*

$$\nu\{f \geq \mathbb{E}_\nu[f] + t\} \leq \exp\left(-\frac{\mu t^2}{2}\right).$$

*Proof.* According to Lemma F.10, it suffices to prove that for any $1$-Lipschitz function $f$ with expectation $\mathbb{E}_\nu[f] = 0$,

$$\mathbb{E}\left[e^{\lambda f}\right] \leq e^{\lambda^2/(2\mu)}.$$

To prove this, it suffices, by a routine truncation and smoothing argument, to prove it for bounded, smooth, compactly supported functions $f$ such that $\|\nabla f\| \leq 1$. Assume that $f$ is such a function. Then for every $\lambda \geq 0$ the log-Sobolev inequality implies

$$\text{Ent}_\nu\left(e^{\lambda f}\right) \leq \frac{2}{\mu}\mathbb{E}_\nu\left[\left\|\nabla e^{\lambda f/2}\right\|^2\right],$$

which is written as

$$\mathbb{E}_\nu\left[\lambda f e^{\lambda f}\right] - \mathbb{E}_\nu\left[e^{\lambda f}\right]\log\mathbb{E}\left[e^{\lambda f}\right] \leq \frac{\lambda^2}{2\mu}\mathbb{E}_\nu\left[\|\nabla f\|^2 e^{\lambda f}\right].$$

With the notation $\varphi(\lambda) = \mathbb{E}\left[e^{\lambda f}\right]$ and $\psi(\lambda) = \log\varphi(\lambda)$, the above inequality can be reformulated as

$$\lambda\varphi'(\lambda) \leq \varphi(\lambda)\log\varphi(\lambda) + \frac{\lambda^2}{2\mu}\mathbb{E}_\nu\left[\|\nabla f\|^2 e^{\lambda f}\right]$$

$$\leq \varphi(\lambda)\log\varphi(\lambda) + \frac{\lambda^2}{2\mu}\varphi(\lambda),$$

where the last step follows from the fact $\|\nabla f\| \leq 1$. Dividing both sides by $\lambda^2\varphi(\lambda)$ gives

$$\left(\frac{\log(\varphi(\lambda))}{\lambda}\right)' \leq \frac{1}{2\mu}.$$

Denoting that the limiting value $\frac{\log(\varphi(\lambda))}{\lambda}\big|_{\lambda=0} = \lim_{\lambda\to 0^+}\frac{\log(\varphi(\lambda))}{\lambda} = \mathbb{E}_\nu[f] = 0$, we have

$$\frac{\log(\varphi(\lambda))}{\lambda} = \int_0^\lambda \left(\frac{\log(\varphi(t))}{t}\right)' dt \leq \frac{\lambda}{2\mu},$$

which implies that

$$\psi(\lambda) \leq \frac{\lambda^2}{2\mu} \implies \varphi(\lambda) \leq \exp\left(\frac{\lambda^2}{2\mu}\right)$$

Then the proof can be completed by a trivial argument of Lemma F.10. $\qquad\square$

**Lemma F.10.** *Let $\mathbf{x}$ be a real random variable. If there exist constants $C, A < \infty$ such that $\mathbb{E}\left[e^{\lambda\mathbf{x}}\right] \leq Ce^{A\lambda^2}$ for all $\lambda > 0$ then*

$$\mathbb{P}\{\mathbf{x} \geq t\} \leq C\exp\left(-\frac{t^2}{4A}\right)$$

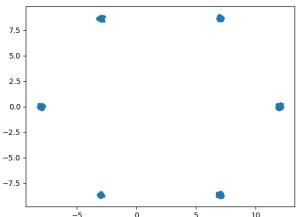

Figure 3: The 1000 particles sampled from the target distribution, which utilized as the criterion by using MMD

*Proof.* According to the non-decreasing property of exponential function $e^{\lambda x}$, we have

$$\mathbb{P}\left\{\mathbf{x} \geq t\right\} = \mathbb{P}\left\{e^{\lambda \mathbf{x}} \geq e^{\lambda t}\right\} \leq \frac{\mathbb{E}\left[e^{\lambda \mathbf{x}}\right]}{e^{\lambda t}} \leq C \exp\left(A\lambda^2 - \lambda t\right),$$

The first inequality follows from Markov inequality, and the second follows from the given conditions. By minimizing the RHS, i.e., choosing $\lambda = t/(2A)$, the proof is completed. $\square$

**Lemma F.11.** *Suppose $q$ is a distribution which satisfies LSI with constant $\mu$, then its variance satisfies*

$$\int q(\boldsymbol{x}) \left\| \boldsymbol{x} - \mathbb{E}_{\tilde{q}}\left[\mathbf{x}\right] \right\|^2 \mathrm{d}\boldsymbol{x} \leq \frac{d}{\mu}.$$

*Proof.* It is known that LSI implies Poincaré inequality with the same constant, i.e., $\mu$, which means if for all smooth function $g \colon \mathbb{R}^d \to \mathbb{R}$,

$$\mathrm{var}_q\left(g(\mathbf{x})\right) \leq \frac{1}{\mu}\mathbb{E}_q\left[\|\nabla g(\mathbf{x})\|^2\right].$$

In this condition, we suppose $\boldsymbol{b} = \mathbb{E}_q[\mathbf{x}]$, and have the following equation

$$\int q(\boldsymbol{x}) \left\| \boldsymbol{x} - \mathbb{E}_q\left[\mathbf{x}\right] \right\|^2 \mathrm{d}\boldsymbol{x} = \int q(\boldsymbol{x}) \left\| \boldsymbol{x} - \boldsymbol{b} \right\|^2 \mathrm{d}\boldsymbol{x}$$

$$= \int \sum_{i=1}^d q(\boldsymbol{x}) \left(\boldsymbol{x}_i - \boldsymbol{b}_i\right)^2 \mathrm{d}\boldsymbol{x} = \sum_{i=1}^d \int q(\boldsymbol{x}) \left(\langle \boldsymbol{x}, \boldsymbol{e}_i \rangle - \langle \boldsymbol{b}, \boldsymbol{e}_i \rangle\right)^2 \mathrm{d}\boldsymbol{x}$$

$$= \sum_{i=1}^d \int q(\boldsymbol{x}) \left(\langle \boldsymbol{x}, \boldsymbol{e}_i \rangle - \mathbb{E}_q\left[\langle \mathbf{x}, \boldsymbol{e}_i \rangle\right]\right)^2 \mathrm{d}\boldsymbol{x} = \sum_{i=1}^d \mathrm{var}_q\left(g_i(\mathbf{x})\right)$$

where $g_i(\boldsymbol{x})$ is defined as $g_i(\boldsymbol{x}) \coloneqq \langle \boldsymbol{x}, \boldsymbol{e}_i \rangle$ and $\boldsymbol{e}_i$ is a one-hot vector ( the $i$-th element of $\boldsymbol{e}_i$ is 1 others are 0). Combining this equation and Poincaré inequality, for each $i$, we have

$$\mathrm{var}_q\left(g_i(\mathbf{x})\right) \leq \frac{1}{\mu}\mathbb{E}_q\left[\|\boldsymbol{e}_i\|^2\right] = \frac{1}{\mu}.$$

Hence, the proof is completed. $\square$

**Lemma F.12.** *(Lemma 12 in Vempala & Wibisono (2019)) Suppose $p \propto \exp(-f)$ satisfies Talagrand's inequality with constant $\mu$ and is $L$-smooth. For any $p'$,*

$$\mathbb{E}_{p'}\left[\|\nabla f(\mathbf{x})\|^2\right] \leq \frac{4L^2}{\mu}\mathrm{KL}\left(p'\|p\right) + 2Ld.$$

## G  EMPIRICAL RESULTS

**Problem settings.** We consider the target distribution defined on $\mathbb{R}^2$ to be a mixture of Gaussian distributions with 6 modals. Meanwhile, we draw $1,000$ particles from the target distribution shown in Fig 3 and used MMD (with RBF kernel) as the metric.

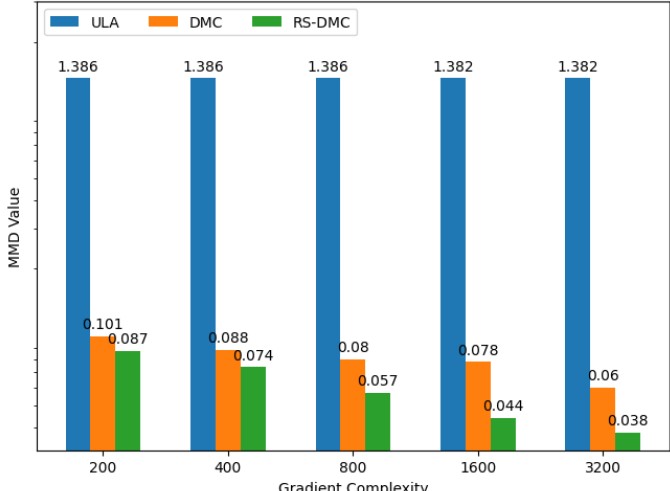

Figure 4: MMD results for ULA, DMC, and RS-DMC for the different gradient complexity. Experimental results show that ULA attains a slow convergence rate with a high sampling error. Both DMC and RS-DMC converge rapidly but RS-DMC can achieve smaller sampling errors than DMC, even with lower gradient complexities. In particular, RS-DCM with 800 gradient complexity can outperform DCM with over 3000 gradient complexity.

We compare the MMDs among three sample algorithms including ULA Neal (1992); Roberts & Tweedie (1996), DMC Huang et al. (2023), and RS-DMC when the number of gradient oracles are $\{200, 400, 600, 800, 1600, 3200\}$. Besides, we tuned hyper-parameters for all three algorithms and left their optimal choice in our code given in the latest supplementary material. Then, we show the final MMDs in Fig 4.

Besides, to compare the behaviors of the three methods, we illustrate the particles when the algorithms return for different gradient complexity in Fig 5. We note that

- ULA will quickly fall into some specific modals with an extremely high sampling error, measured in Maximum Mean Discrepancy (MMD), and is significantly worse than DMC and RS-DMC.
- DMC will quickly cover the different modals. However, converging to their means will be slower than RS-DMC.
- RS-DMC will quickly cover the different modes and converge to their means. Compared to DMC, it requires much fewer gradient complexities to achieve a similar sampling error. For instance, RS-DMC with 800 gradient complexity can outperform DMC with over 3200.

**Explanation about the resemblance of modals' variances.** In the previous experiments, all parameters were tuned to achieve a lower MMD. However, in some cases, MMD may not be able to perfectly characterize the sampling performance, while some statistical information, e.g., the resemblance of variances of modals may be obfuscated. In order to mitigate this issue, we further conduct the experiments with a different set of hyperparameters such that the generated samples can be reasonably closer to the ground truth. In particular, we set the gradient complexity as 200 and generate samples using ULA, DMC, and RS-DMC. The distribution of the output particles is visualized in Fig 6. The comparisons of MMD and variance estimations of different modals are shown in Table 3 and Table 4, respectively. Then it can be clearly observed that

- RS-DMC has better coverage than ULA and DMC, which is measured by MMD (see Fig 6 and Table 3).
- RS-DMC has better concentration than ULA and DMC, which is measured by the variance of modals (see Table 4, where the variance of RS-DMC is closer to that of the ground truth samples for almost all modals).

This again demonstrates the superior performance of RS-DMC compared to baseline algorithms.

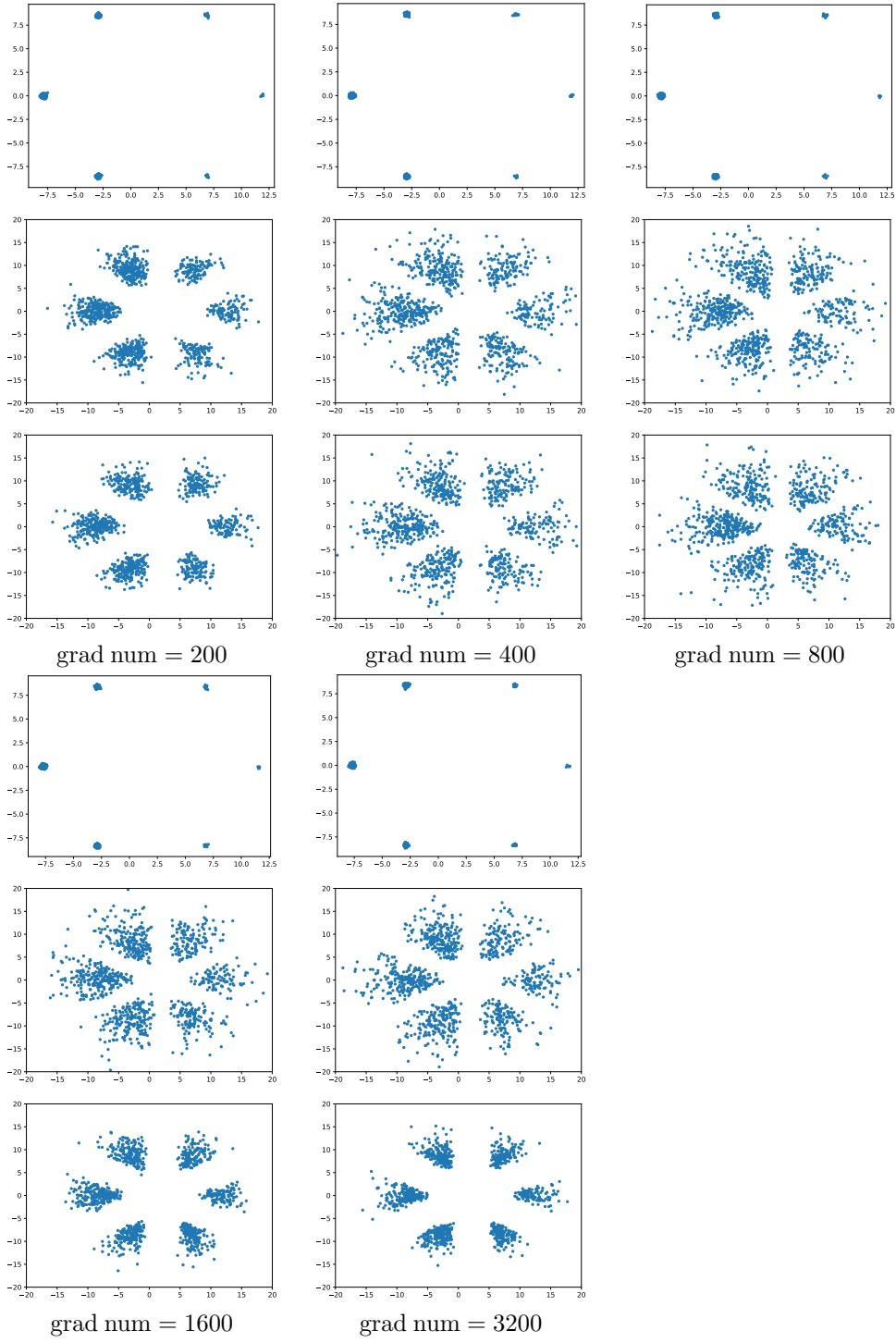

Figure 5: Illustration of the returned particles for ULA, DMC, and RS-DMC. The first row is returned by ULA, the second is DMC and the last is from RS-DMC. Experimental results show that ULA converges fast in the local regions of modes, while it suffers from the problem of covering all modes. DMC can cover most modes with few gradient oracles but converge slowly in local regions. RS-DMC takes advantage of both ULA and DMC, which can cover most modes and admit a relatively faster local convergence.

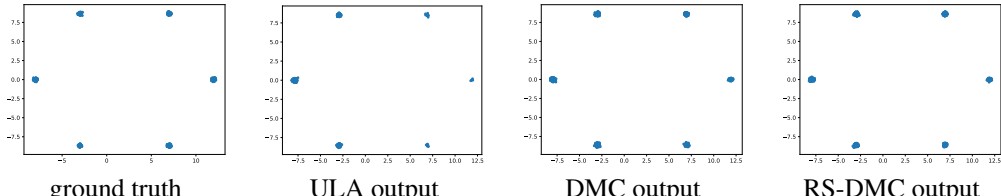

Figure 6: Distribution of the samples generated by ULA, DMC, and RS-DMC, where the gradient complexity is set to be 200.

| | ULA | DMC | RS-DMC |
|---|---|---|---|
| MMD loss | 1.386 | 0.129 | 0.112 |

Table 3: The comparison of MMD in the good resemblance of modals' variances when gradient complexity is chosen as 200.

| Var | ULA | DMC | RS-DMC | ground truth |
|---|---|---|---|---|
| Modal 0 | 0.0337 | 0.0337 | 0.0332 | 0.0217 |
| Modal 1 | 0.0574 | 0.0311 | 0.0314 | 0.0188 |
| Modal 2 | 0.0459 | 0.0299 | 0.0324 | 0.0196 |
| Modal 3 | 0.0394 | 0.0338 | 0.0287 | 0.0184 |
| Modal 4 | 0.0449 | 0.0330 | 0.0289 | 0.0174 |
| Modal 5 | 0.0409 | 0.0321 | 0.0306 | 0.0220 |

Table 4: The comparison of variance of different modals when gradient complexity is chosen as 200. The experimental results show both DMC and RS-DMCcan achieve better variance estimation. In some remote modals, the gap between ULA and DMC/RS-DMCwill be even larger than that between RS-DMCand ground truth.

