# OpenReview forum: "Recursive Score Estimation Accelerates Diffusion-Based Monte Carlo"
_ICLR.cc/2024/Conference — Submitted to ICLR 2024_

### Official Review · Reviewer_rqBy · 2023-10-23

**Soundness:** 3 good
**Presentation:** 3 good
**Contribution:** 4 excellent
**Rating:** 8
**Confidence:** 2

**Summary:**

The authors present a framework for sampling from a general distribution, known up to a normalizing constant, using a denoising process (which is the reversal of the Ornstein-Uhlenbeck or "Gaussian noising" process).

The authors exhibit:
-  a specific discretization of the denoising process (which requires the unknown scores of intermediate distributions)
- a specific estimation procedure (to approximate the unknown scores of the intermediate distributions)

which together provide a discrete algorithm for sampling.

The authors prove the convergence of their algorithm for a general target distribution (no assumption on log-concavity) at a rate that is quasi-polynomial (compared to exponential using classical Langevin sampling).

**Strengths:**

These theoretical results seem very relevant at present. There is a recent trend in the literature of score-based diffusion models [1] to sample from a general  (e.g. multimodal) target distribution following a denoising process as opposed to a classical Langevin process. While it is known that the Langevin process is sub-optimal, in that it can require an exponential number of steps to converge to the target distribution with a given precision, it is not so clear why the denoising process may be preferable. The authors' theory is a welcome step forward.

The authors' results seem comprehensive, namely in terms of:
- proof of convergence of the under general assumptions (the target distribution is known up to a normalizing constant; importantly, it need not be log-concave)
- a provably better convergence rate (or mixing time), that is quasi-polynomial as opposed to exponential, in the number of steps required to achieve a given precision
- a comprehensive analysis of sources of error arising from inter-connected sampling and estimation problems

The authors also make a visible effort to make their theory intelligible and to give intuition on how to set different hyperparameters such as the window length S.


Disclaimer: I have not checked the math carefully nor is sampling using diffusions my primary research area.


[1] Song et al. Score-based Generative Modeling Through Stochastic Differential Equations. ICLR, 2021.

**Weaknesses:**

**Context**. The authors' algorithm consists in approximating the score of the intermediate distributions, in order to run a discrete version of the denoising process. Approximating the score of an intermediate distribution $\nabla \log p_{k, t}$ (window $k$, step $t$ within the window) is achieved by a sequence of estimation and sampling problems (section 3.2.). For context, we recap the authors' method. With the authors' notations, $p$ is used for the intermediate distributions from the sampling process, and $q$ are auxiliary distributions (defined in section 3.1.) that are used to estimate are the scores of the intermediate distributions.

Initialization: start with the known score of the target we start with the known score of the target distribution $\nabla \log p_{0, 0}$.

Loop: for window $i$ in the range of $0$ to $k-1$,
1. Estimate the score at the start of the window,  $\nabla \log p_{i, 0}$
2. Sample from the auxiliary distribution at the start of the next window, $q_{i+1, 0}$

Termination: once we arrive at the desired window $k$, we move to the correct place inside that window,
- Estimate the score at the start of the window $\nabla \log p_{k, 0}$
- Sample from the auxiliary distribution in the correct place inside that window $q_{k, t}$
- Estimate the score at the correct place inside that window $\nabla \log p_{k, t}$

The authors' method depends on making these two steps - estimation and sampling - efficient.

**Q1**. The argument for efficient sampling is clear: the authors choose a "small enough" window length $S$, so that its uniform discretization into steps of length $\eta$, produces sampling distributions $q_{k, t}$ that are log-concave and can therefore be sampled using a classical ULA (Unadjusted Langevin Algorithm) with a polynomial number of steps. Is that right?

**Q2**. However, the argument for efficient estimation is not explicit to me. Is it that we are, in the first equation of section 3.1., essentially just computing the empirical mean of $q$, rescaled by a certain factor. So the error of that estimate is the standard error of the mean (SEM), which is proportional to the variance of $q$. Do we have a handle on the variance of $q$?

**Questions:**

**Q1**. Could the authors reunite their recommendations on setting hyperparameters in a list? For example:

-  the diffusion length T should be "big enough". Practically, T should be at least on the order of $\log d / \epsilon$
-  the window length  S should be "small enough" for the sampled distributions to be log-concave but "big enough" to avoid extra discretization steps. Practically, S should be on the order of $\frac{1}{2} \log \frac{2L + 1}{2L}$.
- the length of a step inside a window $\eta$ should be ...? is there any recommendation for that?

**Q2**. Can the authors discuss which hyperparameters would be easier or harder to set?

For example, T seems to be easier to set for two reasons. First, we can actually compute $\log d / \epsilon$. Second, if T is "too big", the authors' convergence rate would still apply.

However, setting S seems to be trickier. We cannot actually compute $\frac{1}{2} \log \frac{2L + 1}{2L}$ given that we do not known the smoothness constant $L$. So we would set S somewhat heuristically, and S could be potentially "too big" or "too small". If I correctly understand the authors' argument, the more dangerous situation is if S is "too big" as the sampling distributions might not be log-concave and this would introduce an exponential number of steps in sampling.

**Q3**. The efficiency of the authors' method seems to rely on the assumption that the "error propagation is benign, i.e. $l_{k, r}(\epsilon) = \epsilon$". Could the authors discuss the plausibility of this assumption? Is this a common assumption in the literature? Or are there works where benign error propagation appears in another context, supporting the claim that error propagation can indeed be benign in certain cases?

---

> ### Author Response · Authors · 2023-11-17
> **Response to Reviewer rqBy**
>
> Thank you for your positive feedback. We responded to your concerns point by point in the following.
>
> > *W1: The argument for efficient sampling is clear: the authors choose a "small enough"*
>
> Yes, your understanding is correct.  Just as you mentioned in your later question, the choice of “S” is subtle in theory. We set it to be “just small enough” such that sampling distributions $q_{k,t}$’s are log-concave, while in this case, $S$ is still in the constant level. If it is too small (e.g., $O(\epsilon)$),  RS-DMC may not be able to achieve a quasi-polynomial complexity since the recursion number ($\tilde{O}(1/S)$) will be too large and harm the efficiency.
>
> > *W2:  However, the argument for efficient estimation is not explicit to me. Is it that we are, in the first equation of section 3.1., essentially just computing the empirical mean of $q$, rescaled by a certain factor. ...*
>
> We have controlled the variance of q in our proof in an implicit manner. As you mentioned before, the distribution q is strongly log-concave (given in Eq. 4) and we can get a lower bound on the strongly log-concave constant (see Lemma E.2). Therefore, it also satisfies the log-Sobolev inequality due to Lemma F.4, which can imply the variance upper bound (see Lemma F.11). Then in our proof, we directly make use of the Sobolev inequality of $q$ to derive the high-probability bound (or concentration results) for estimating the mean of $q$.
>
> > *Q1: Could the authors reunite their recommendations on setting hyperparameters in a list? ...*
>
> Sorry for the confusion. We have listed all settings of hyper-parameters in Theorem B.1, which is a detailed version of the main theorem (Theorem 4.1) in our paper. Specifically, the step inside a window, i.e.,  $\eta$ is required to be
> $$\eta = C_\eta(M+d)^{-1}\epsilon,$$
> where $C_\eta$ is a constant shown in Table. 2.
>
> > *Q2: Can the authors discuss which hyperparameters would be easier or harder to set? For example, T seems to be easier to set for two reasons. ...*
>
> This is a good point. In particular, the setting $S$ in our algorithm is similar to setting stepsize in optimization, where theoretically people need to use the smoothness constant to determine the feasible choices, while in practice they are just hyperparameters that can be free to tune. We agree that from a theoretical perspective, $S$ needs to be properly chosen in order to achieve good gradient complexity, making it “too big” may lead to a bad gradient complexity in theory. However, in practice, just like trying different stepsize for optimization, we can tune different $S$ in our algorithms. We have conducted numerical experiments on a multi-mode Gaussian-mixture case (see Appendix G of the latest revision), where we find that using a constant $S$ (after a simple grid search) in our algorithm can achieve improved sampling performance than baseline algorithms (DMC and ULA). This implies that the hyperparameters of our algorithms, including $T$ and $S$, are easy to set in practice.
>
> > *Q3: The efficiency of the authors' method seems to rely on the assumption that the error propagation is benign, i.e. $l_{k,r}(\epsilon)=\epsilon$.*
>
> A good question! In our paper, supposing $l_{k,r}(\epsilon)=\epsilon $is only for explaining our intuition easily. It is actually not an assumption, and we have proved it in the Appendix. Actually, we prove $l_{k,r}(\epsilon)=C\epsilon$ by setting $C=\frac{e^{-2S}}{640}$, which will also preserve the quasi-poly gradient complexity. Specifically, we show that a good score estimation approximation can be ensured
>
> $$\mathbb{P}\left[\left\Vert v^\gets_{k,r\eta}(x) - \nabla\log p_{k,S-r\eta}(x)\right\Vert^2\le \epsilon\right] \ge 1-\delta$$
>
> with the following benign error propagation of score estimation, i.e.,
>
> $$\left\Vert \nabla \log p_{k,0}(x^\prime) -v^\prime_{k,0}(x^\prime)\right\Vert^2\le C\epsilon\le  \frac{e^{-2(S-r\eta)}\epsilon}{640}.$$
>
> This statement adapts to Lemma E.7 in our paper.

---

> ### Author Response · Authors · 2023-11-21
>
> Dear Reviewer rqBy:
>
> Thank you again for taking the time to review our submission. As the end of the author-reviewer discussion is approaching, we would greatly appreciate it if you could let us know whether your previous concerns have been addressed. In any case, we are ready to answer any further questions.

---

> ### Comment · Reviewer_rqBy · 2023-11-21
> **Acknowledging author response**
>
> I thank the authors for their response which addresses my concerns. I would ask the authors to reference the above clarifications in the main text, namely:
>
> 1. Please refer in the main text to Table B.1 so that the reader explicitly knows where to find the practical recommendations for hyperparameter values
>
> 2. Please include in the main text your "high level" explanation for efficient estimation. This would give the reader an idea of what arguments to expect and make the paper easier to follow.
>
>  "the distribution q is strongly log-concave (given in Eq. 4) and we can get a lower bound on the strongly log-concave constant (see Lemma E.2). Therefore, it also satisfies the log-Sobolev inequality due to Lemma F.4, which can imply the variance upper bound (see Lemma F.11). Then in our proof, we directly make use of the Sobolev inequality of  to derive the high-probability bound (or concentration results) for estimating the mean of."
>
> 3. Please clarify that "benign error propagation" is actually proven
>
> Thank you for the clarification. I appreciate that the assumption of benign error propagation (section 3.3) is useful for intuition. However, it is not obvious from the main text that this assumption is actually proven in the Appendix. Could you add a sentence (paragraph "Quasi-polynomial Complexity", in section 3.3) that explicitly say so in the main text and point the reader to the specific places where this is proven in the Appendix?

---

> > ### Author Response · Authors · 2023-11-22
> > **Thanks for your good suggestions**
> >
> > Thank you for your good suggestions, which indeed make this paper clearer and easier to understand. We have made the following revisions with blue text in our latest revision.
> >
> > > *Please refer in the main text to Table B.1 so that the reader explicitly knows where to find the practical recommendations for hyperparameter values*
> >
> > In the paragraph after Theorem 4.1, we refer to Theorem B.1 and Table 2 in the main text for the practical recommendations of hyperparameter values.
> >
> > > *Please include in the main text your "high level" explanation for efficient estimation. *
> >
> > We include in the main text explanation for efficient estimation in **Upper bound Term 3** paragraph of the proof sketch sections (Section 4.2).
> >
> > > *Please clarify that "benign error propagation" is actually proven*
> >
> > We add a sentence to refer to the benign error propagation that is proven in Lemma E.7 in Section 3.3.
> >
> > If you have any further concerns, please feel free to let us know. We will address them as possible as we can.

---

> > > ### Comment · Reviewer_rqBy · 2023-11-22
> > > **Thanks to the authors**
> > >
> > > I thank the authors for these modifications

---

### Official Review · Reviewer_FZMK · 2023-10-28

**Soundness:** 2 fair
**Presentation:** 2 fair
**Contribution:** 2 fair
**Rating:** 5
**Confidence:** 3

**Summary:**

Sampling from non-log concave distribution is generally difficult. This paper proposes an efficient sampling method for resolving this difficulty based on the reverse process of diffusion models. The key idea is to decompose the diffusion process into a number of sufficiently short segments, in which the intermediate distributions become log-concave under certain moderate assumptions. A recursive algorithm to estimate score function utilizing this property is proposed. It is shown that the gradient complexity of the algorithm is quasi-polynomial with respect to the gradient error.

**Strengths:**

A new algorithm to estimate the score in diffusion models is developed with a mathematical guarantee.

**Weaknesses:**

Practical usefulness is not clear. "S" could be very small yielding very large "K", which implies the method would be practically difficult to perform even if it is of quasi-polynomial gradient complexity. Some experimental evidence for the practical usefulness is desired.

**Questions:**

Can you show the usefulness of the proposed method by numerical experiments on some concrete examples?

---

> ### Author Response · Authors · 2023-11-17
> **Response to Reviewer FZMK**
>
> Thank you for review. We responded to your concerns point by point in the following.
>
> > *W1: Practical usefulness is not clear. "S" could be very small yielding very large "K", which implies the method would be practically ...*
>
> We respectfully disagree with your comments. It should be emphasized that the length of segments will be $S=1/2\log(1+1/2L)$, where $L$ is the Lipschitz parameter of the score function. Then it can be seen that $S$ is not very small but can be in the constant level (if treating $L$ to be constant). Consequently, the recursion number $K= 2\log [(Ld+M)/\epsilon] \cdot S^{-1}$ will not be extremely large, which is only logarithmic in the target error $\epsilon$ and dimension $d$.
>
> Moreover, in practice, “S” and “K” are hyperparameters that can be tuned. We have conducted numerical experiments on a multi-mode Gaussian mixture case (which we added in Appendix G of the latest version) with tuned $S$ and $K$ (where $K$ is typically in the constant level), from which we can find that the proposed RS-DMC algorithm can achieve much better performance than baselines (DMC and ULA), even with fewer gradient complexities. This supports our theory and justifies the practical usefulness of RS-DMC.
>
> > *Q1: Can you show the usefulness of the proposed method by numerical experiments on some concrete examples?*
>
> To verify the usefulness of RS-DMC, we conduct numerical experiments on 2D multi-mode Gaussian mixture targets, which are non-log-concave with bad isoperimetric properties. We compare ULA, DMC, and RS-DMC. We have added the experiment details and anonymous link to our code in the revision. We attached the error comparisons (measured by Maximum Mean Discrepancy) among them as follows:
>
> | Grad Num| ULA   | DMC   | RS-DMC |
> | ------------------- | ----- | ----- | ------ |
> | 200                 | 1.386 | 0.101 | 0.087  |
> | 400                 | 1.386 | 0.088 | 0.074  |
> | 800                 | 1.386 | 0.080 | 0.057  |
> | 1600                | 1.382 | 0.078 | 0.044  |
> | 3200                | 1.382 | 0.060 | 0.038  |
>
>  From our experimental results, we can observe:
>
> - ULA will quickly fall into some specific modes with an extremely high sampling error, measured in Maximum Mean Discrepancy (MMD), and is significantly worse than DMC and RSDMC.
> - DMC will quickly cover the different modes. However, the process of converging to their means will be slower than RS-DMC.
> - RS-DMC will quickly cover the different modes and converge to their means. Compared to DMC, it requires much fewer gradient complexities to achieve a comparable or better sampling error. For instance, RS-DMC with 800 gradient complexity can outperform DMC with over 3200 gradient complexity.
>
> Then it is clear that RS-DMC significantly outperforms ULA and DMC in terms of gradient complexities, which backs up our theory and demonstrates the superior empirical performance of RS-DMC.  More details and visualizations of the experiment can be found in Appendix G of the latest revision, where we include our source code with an anonymous link. We will further add more numerical experiments to justify the practical advantages of RS-DMC in the revision.

---

> > ### Comment · Reviewer_FZMK · 2023-11-21
> >
> > Thank you for the response.
> >
> > >We respectfully disagree with your comments. It should be emphasized that the length of segments will be , where  is the Lipschitz parameter of the score function. Then it can be seen that  is not very small but can be in the constant level (if treating  to be constant). Consequently, the recursion number  will not be extremely large, which is only logarithmic in the target error  and dimension .
> >
> > > Moreover, in practice, “S” and “K” are hyperparameters that can be tuned. We have conducted numerical experiments on a multi-mode Gaussian mixture case (which we added in Appendix G of the latest version) with tuned  and  (where  is typically in the constant level), from which we can find that the proposed RS-DMC algorithm can achieve much better performance than baselines (DMC and ULA), even with fewer gradient complexities. This supports our theory and justifies the practical usefulness of RS-DMC.
> >
> > I thought "L" could be large, for which "K" would become large.
> > If is difficult to imagine how large "L" in practice, so that showing experimental results for concrete problems is important.
> >
> > The added numerical result is useful for our understanding while it would be much nicer if it were that for more practical problems.

---

> > > ### Author Response · Authors · 2023-11-22
> > >
> > > Thank you very much for your prompt reply and your acknowledgment of our effort in the numerical experiments.
> > >
> > > First, we would like to reiterate that the choice of $K$ in our paper is for establishing the rigorous theoretical guarantee, which has been shown to be significantly better than baselines including DMC and Langevin algorithms, which also require to use $L$ to set their hyperparameters (e.g., stepsize, iteration number, etc.).
> > >
> > > Then, we would like to clarify that, in real practice, the exact quantity of $L$ is not needed. This resembles other numerical algorithms (e.g., Langevin algorithm, gradient descent), whose hyperparameters (e.g., stepsize, iteration numbers) need to be properly chosen according to $L$ in theory,  but are freely tuned in practice without knowing $L$.
> > >
> > > Finally, we would like to point out that when $K=1$, RS-DMC will degenerate to DMC, which can surely converge with a feasible stepsize and iteration number. In our experiments, we just enlarge $K$ from $K=1$ to some other constants and the performance of RS-DMC can be significantly improved. We believe this is sufficient to justify the practical usefulness of RS-DMC.

---

> > > > ### Comment · Reviewer_FZMK · 2023-11-22
> > > >
> > > > Thank you for further explanation. I keep my current score.

---

> ### Author Response · Authors · 2023-11-21
>
> Thank you for your feedback once again. We appreciate your input and hope that our response addresses your concerns and questions. Specifically, we have conducted numerical experiments on RS-DMC and compared it with ULA and DMC to demonstrate the practical usefulness of our method.
>
> As the end of the author-reviewer discussion is approaching, we would greatly appreciate it if you could let us know whether your previous concerns have been addressed. In any case, we are ready to answer any further questions.

---

### Official Review · Reviewer_UkCa · 2023-11-02

**Soundness:** 4 excellent
**Presentation:** 2 fair
**Contribution:** 4 excellent
**Rating:** 8
**Confidence:** 3

**Summary:**

This paper studies the problem of sampling from a distribution $p_* \propto e^{-f_*}$ given access to $f_*$. It proposes a novel recursive score estimation scheme to solve this problem using a quasi-polynomial number of gradient computations in $\epsilon$, the desired error tolerance. This improves significantly on the exponential dependence in prior work. The key observation seems to be that if you have a distribution with lipschitz score function, and you run the OU process for a small time depending on this lipschitzness, then the posterior distribution is log-concave and can be sampled from using ULA given access to the prior score. This observation can be used to estimate the score functions for different smoothing levels, and finally, once these have been estimated, diffusion monte carlo can be used to sample from $p_*$

**Strengths:**

Great paper! I am a fan of the recursive score estimation scheme. I think this is a solid piece of work that will inspire future work in the area.  I'm excited to see whether it inspires new practical algorithms for sampling from diffusion models.

More detailed comments:

- The problem is an interesting one, and the solution proposed is interesting and novel.
- The key observations are crisply stated, and could possibly find other uses.
- The improvement over prior work is substantial.

**Weaknesses:**

- The presentation can use a lot of improvement. The figures are currently difficult to understand. The algorithm blocks are also difficult to interpret.
- Quasi-polynomial is interesting, but I would be curious to know if you think polynomial is possible/what the barriers are. Would really appreciate it if you put something about this at the end of the paper.
- More intuition about why the complexity is quasi-polynomial would be useful.

**Questions:**

- What are the barriers to obtaining polynomial complexity?
- Can you give more intuition for where the quasi-polynomial complexity comes from? Currently, there is a small block that is a bit difficult to interpret.

---

> ### Author Response · Authors · 2023-11-17
> **Response to Reviewer UkCa**
>
> Thank you for your appreciation of this work.  We hope to solve your concerns point by point in the following.
>
> > *W1: The presentation can use a lot of improvement. ...*
>
> We will simplify our figures and the interpretation of our algorithm, and clarify the correspondence between them to make people understand it easier in the next version.
>
> > *W2: Quasi-polynomial is interesting, but I would be curious to know if you think polynomial is possible/what the barriers are. Would really appreciate it if you put something about this at the end of the paper. ...*
>
> > *Q1: What are the barriers to obtaining polynomial complexity?*
>
> A good question! In general, we think the polynomial complexity for sampling from general non-log-concave distribution can hardly be achieved. Our understanding is drawn from the optimization perspective in [raginsky2017non], which uses the sampling algorithm for solving a nonconvex optimization problem. Then, if the polynomial complexity can be achieved, we can probably optimize a general non-convex objective in polynomial time. This potentially contradicts the well-known hardness result, i.e., finding a global minimum for general non-convex objectives is an NP-hard problem.
>
> If we seek better complexity results (not limited to polynomial complexity) for SDE-based sampling algorithms, the barrier could be **balancing the property of the SDE path and the computational cost for implementing this path**. On one hand, it can be observed that the cost of implementing Langevin-based methods is relatively low, i.e., one only needs to calculate the gradient $\nabla f(x)$. **However**, the property of SDE path of Langevin-based methods is not good, it may need exponential time to reach the target distribution. On the other hand, for diffusion-based MCMC, the computational cost for implementation is heavy since it involves heavy and complicated score estimations along the entire SDE path. However,  for exchange, the path property of DMC can be much better, it can reach the target distribution within only logarithmic time. Our proposed algorithm, RS-DMC,  can be regarded as a good trade-off between the property of SDE path and the computational cost for implementing this path and thus achieves much lower gradient complexity compared to Langevin-based algorithms. Then we suspect that if such a trade-off can be improved, i.e., either a better SDE is developed or the implementation is more efficient, one can design a faster sampling algorithm with improved complexities (potentially polynomial).
>
> > *W3: More intuition about why the complexity is quasi-polynomial would be useful.*
>
> > *Can you give more intuition for where the quasi-polynomial complexity comes from? Currently, there is a small block that is a bit difficult to interpret.*
>
> The quasi-polynomial complexity stems from the recursive score estimation. In our theory, each recursion step is decomposed into solving a polynomial number of subproblems,  denoted by $N$, to ensure stable error propagation, which will be further transferred to the next recursion step. Besides, we have also shown that the number of recursive steps $K$ is logarithmic in the target error $\epsilon$ and dimension $d$. Therefore, the total complexity of the score estimation will be $O(N^K)$ as we will need to solve at most $N^K$ subproblems. Recalling that $N$ is polynomial in $\epsilon$ and $d$ and $K$ is logarithmic in $\epsilon$ and $d$, we can conclude that $N^K$ is quasi-polynomial in $\epsilon$ and $d$.
>
> We have already provided a rough calculation in Section 3.3. We will further polish this section to improve the clarity.

---

> ### Author Response · Authors · 2023-11-21
>
> Dear Reviewer UkCa:
>
> Thank you again for taking the time to review our submission. As the end of the author-reviewer discussion is approaching, we would greatly appreciate it if you could let us know whether your previous concerns have been addressed. In any case, we are ready to answer any further questions.

---

> > ### Comment · Reviewer_UkCa · 2023-11-22
> >
> > Thanks for addressing my questions. I will maintain my rating

---

### Official Review · Reviewer_Rkvn · 2023-11-05

**Soundness:** 4 excellent
**Presentation:** 2 fair
**Contribution:** 4 excellent
**Rating:** 5
**Confidence:** 4

**Summary:**

In this paper the authors propose a novel algorithm for learning a denoising diffusion model for sampling from a target distribution $p_*$ of which we only have access to its unnormalized density. The proposed methdology relies on a recursive approximation of the score functions. The main idea behind this score estimation is that: 1. the score at some time step $t_k$ and point $x$ can be expressed as an expectation over the law of $X_{t_k-1} | X_{t_k} = x$ 2. this conditional distribution is log concave if $t_{k-1}$ is close enough to $t_k$ and calls for the use of ULA to sample from it. Then, in order to sample from it, one uses make use of the score associated to $X_{t_{k-1}}$, hence the recursive nature of the algorithm. The authors then go on to show the convergence of the resulting algorithm and its gradient complexity without requiring the standard log Sobolev inequality.

**Strengths:**

The algorithm proposed in the paper is pretty smart and solves all the issues of RDS. Furthermore, it is furnished with nice theoretical results that sidestep the use of inequalities such as Poincaré or log Sobolev inequality. This is a very interesting development since in comparison, existing analyses of Langevin Monte Carlo all require such assumptions. Interestingly, the authors do not require assumptions on the score estimation in contrast with previous methods.

Overall, this is a very interesting work.

**Weaknesses:**

- Of course, the main weakness of this paper is the lack of experiment. The main contribution here is methodological and so one would expect to have numerical experiments backing up the methodological and theoretical results. I find it very strange that the authors did not include numerical experiments, comparing for example their method to ULA or such, on non-log concave target distributions and with runtime comparisons. I am willing to raise my score to 8 if the authors provide such comparisons.

- I found the paper to be quite difficult to read due to some of the notations that seem to be unncessarily confusing. The figures do not help either. I think that it would have been easier to just explain the algorithm by fixing some timesteps $t_1, \dotsc, t_K$ such that $X_{t_k} | X_{t_{k+1}}$ is log concave (using the inequality (4)) and then running the backward diffusion using the same discretization, without further diving the segments $[t_k, t_{k+1}]$, leaving it for the appendix.

**Questions:**

I do not have further questions.

---

> ### Author Response · Authors · 2023-11-17
> **Response to Reviewer Rkvn**
>
> Thank you for your positive feedback. We responded to your concerns point by point in the following.
>
> > *Of course, the main weakness of this paper is the lack of experiment. The main contribution here is methodological and so one would expect to have numerical experiments backing up the methodological and theoretical results. ...*
>
> Thanks for your suggestions. We have added numerical experiments in the latest version (see Appendix G), where we conduct numerical experiments on 2D multi-mode Gaussian mixture targets, which are non-log-concave with bad isoperimetric properties. We compare ULA, DMC, and our RS-DMC. We presented the sampling errors (measured by Maximum Mean Discrepancy) obtained by these three algorithms as follows:
>
> | Grad Num| ULA   | DMC   | RS-DMC |
> | ------------------- | ----- | ----- | ------ |
> | 200                 | 1.386 | 0.101 | 0.087  |
> | 400                 | 1.386 | 0.088 | 0.074  |
> | 800                 | 1.386 | 0.080 | 0.057  |
> | 1600                | 1.382 | 0.078 | 0.044  |
> | 3200                | 1.382 | 0.060 | 0.038  |
>
> Then we can find that:
>
> - ULA will quickly fall into some specific modes with an extremely high sampling error, measured in Maximum Mean Discrepancy (MMD), and is significantly worse than DMC and RS-DMC.
> - DMC will quickly cover the different modes. However, the process of converging to their means will be slower than RS-DMC.
> - RS-DMC will quickly cover the different modes and converge to their means. Compared to DMC, it requires much fewer gradient complexities to achieve a comparable or better sampling error. For instance, RS-DMC with 800 gradient complexity can outperform DMC with over 3200 gradient complexity.
>
> Overall, the additional experimental results, though primitive, have already shown the superior empirical performance of RS-DMC over DMC and ULA. This also supports our theoretical results.
>
> More details and visualizations of the experiment can be found in Appendix G of the latest revision, where we include our source code with an anonymous link. Besides, we will add more numerical experiments to justify the practical advantages of RS-DMC in the revision.
>
> > "W2: I found the paper to be quite difficult to read due to some of the notations that seem to be unncessarily confusing.  ..."
>
> Thank you for your suggestions. Your understanding of the high-level idea of our algorithm design is exactly correct. We will simplify the notations as well as our diagram figures according to your suggestions in the revision.

---

> ### Author Response · Authors · 2023-11-21
>
> Dear Reviewer Rkvn:
>
> Thank you again for taking the time to review our submission. As the end of the author-reviewer discussion is approaching, we would greatly appreciate it if you could let us know whether your previous concerns have been addressed. In any case, we are ready to answer any further questions.

---

> > ### Comment · Reviewer_Rkvn · 2023-11-22
> >
> > Dear authors,
> >
> > thank you for your response. I have coded your algorithm and also played with your code and came to the conclusion that the algorithm is very sensitive to the hyperparameters and quite unstable. Furthermore, it hardly scales beyond dimension 2 from what I've noticed. I have yet to understand what really makes it break down. In its current form, the algorithm is quite limited and  since this is not discussed in the paper, I believe it might mislead the general public into thinking that the proposed algorithm outperforms ULA for example. For this reason I lower my score and lean to rejection.

---

> ### Author Response · Authors · 2023-11-23
>
> Sorry for the confusion, we only show the parameters tuned for sigma = 0.1 in our previous code. For other target distributions, you need to find a proper step size for ULA, and proper inner step sizes for DMC/RS-DMC. With proper hyper-parameters, DMC/RS-DMC will have a  better performance compared with ULA.
>
> **In the latest version of demo.ipynb in our supplementary**, we have reorganized and updated our code for easier hyperparameter tunning. We have also included the results as well as proper hyperparameters for sigma=1 and sigma=2 (where we basically enlarge the inner step size for larger $\sigma$, we hope the choice of their hyper-parameters will provide some tuning intuition for you), where RS-DMC/DMC can perform successful sampling in these cases. We summarize the following rules for parameters tuning of DMC/RS-DMC:
>
> - For saving gradient complexity, we can use inner_sample_num and inner_iter_num as 1 (adding more inner_iter_num can improve the stability).
> - For the step size and iteration number of outer loops, you can keep their product a constant near $10$ (corresponding to $T$ in our paper).
> - The most important tuning is about inner step size. If the target distribution has a bad shape, e.g., sigma=0.1, you should choose a small inner step size, e.g., 0.01. In contrast, if the target distribution has a better shape, e.g., sigma=1, a larger inner step size is preferred, e.g., 0.6. It is similar to the rules for tuning step size in ULA.
> - In practice, we need more segments when the number of outer iterations is larger. This phenomenon can be found in our hyperparameter choice when sigma=0.1.
>
> Finally, we clarify that the major contributions and claims in our paper are regarding the theoretical gradient complexity (there is no sampling algorithm that can handle general non-log-concave distributions with provable quasi-polynomial guarantees), based on which we show that RS-DMC is very promising for sampling hard distributions, which can be further investigated and improved in both theory and practice.
>
> Our current experiments are very primitive and conducted within a very short time, thus we need to consistently refine the experiments to achieve better performance. With the previous hyperparameters’ tuning rules, we ensure DMC/RS-DMC will have a stably better performance compared with ULA in 2-d cases, we think it is a promising starting point. We will scale it to a higher dimension in the future version.
>
> In fact, our current results have already revealed some benefits of RS-DMC for sampling non-log-concave distributions empirically, although we believe that further refinements and hyperparameter configurations are needed to enhance stability and generalizability. However, these further steps may be beyond the main scope of our paper and thus will be left as future research directions.

---

### Official Review · Reviewer_nqUk · 2023-11-09

**Soundness:** 3 good
**Presentation:** 4 excellent
**Contribution:** 3 good
**Rating:** 3
**Confidence:** 4

**Summary:**

The authors propose an algorithm to sample from an unnormalized density. Their algorithm is inspired in Reverse Diffusion Sampling, where the score is approximated by sampling from *hard* subproblems. The proposed method breaks down the task into *simpler* problems, which can be sampled efficiently, resulting in easier, faster score approximations. Which eventually result in good generation.

The main contributions of the paper are:
- Developing a novel algorithm **Recursive Score Diffusion Monte Carlo** that  approximates the score by recursively sampling from easier subproblems
- Establishing a convergence guarantee for the method under mild assumptions
- The proposed algorithm has quasi-polynomial gradient complexity under mild assumptions on the data distribution

**Strengths:**

1. The paper has a clear explanation of the ideas leading up to their method
2. The algorithm provides with novel insights on how to approximate the score
3. The proposed method has a quasi polynomial gradient complexity bound, something that is strongly desirable

**Weaknesses:**

1. The main theorem in the paper results in a high probability bound of the form $KL(P_*||P_{0,S}^\leftarrow) = \tilde O(\epsilon)$ with probability $1-\epsilon$. This means that that the accuracy of the method scales linearly with the probability, so that we are forced to take $\epsilon$ very close to $0$. This would result that in practice many samples are needed to obtain high accuracy
2. The paper lacks numerical examples to demonstrate their techniques. This is important at it would demonstrate if the method is actually an improvement from the DMC paper. One reason for the lack of experiments could be that the total number of samples needed to run this algorithm grows with $n_k * m_k = O(1/\epsilon^5)$ which can be computationally expensive. If this was the case then the algorithm is not implementable in practice despite its remarkable properties

**Questions:**

Despite being a theoretical paper, I think the key of the proposed method is that it tries to find a way to implement the problem for nonconvex problems, something that DMC would struggle with. Because of that I wonder if this method be implementable in practice, considering the computational challenges that come with it? It seems that the recursion although significantly simplifying the sampling tasks, results in very strong computational requirements, so addressing this would be very important for me

---

> ### Author Response · Authors · 2023-11-17
> **Response to Reviewer nqUk Part 1**
>
> Thank you for your detailed review. We responded to your concerns point by point in the following.
>
> >  *W1: The main theorem in the paper results in a high probability bound of the form $\mathrm{KL}(p_\ast \Vert p_{0,S}^\gets)=\tilde{O}(\epsilon)$ with probability $1-\epsilon$. ...*
>
> We would like to explain that the probability does not need to scale linearly with the accuracy and we set the probability $1-\epsilon$ only for the presentation simplicity (as we do not want to involve additional notations). Actually, the convergence of KL  $\mathrm{KL}(p_\ast\Vert p_{0,S}^\gets)=\tilde{O}(\epsilon)$ can be achieved with probability $1-\delta$ for arbitrary small $\delta$, where $\delta$ can be chosen independently on $\epsilon$ shown as follows:
>
> Under almost the same setting as in Theorem 4.1, with probability at least $1-\delta$, it holds that $\mathrm{KL}(p_\ast\Vert p_{0,S}^\gets)=\tilde{O}(\epsilon)$. Besides, the gradient complexity of RS-DMC is
> $$\exp\left[\mathcal{O}\left( \max\left\(\left(\log \frac{Ld+M}{\epsilon}\right)^3, \log \frac{Ld+M}{\epsilon}\cdot \log \frac{1}{\delta}\right) \right)\right].$$
>
> We provide the formal version in Corollary B.2 in the latest revision with blue text.
>
> Moreover, we would like to point out that the high-probability argument can be merged with the KL-divergence sampling error bound, leading to a clean sampling error bound (high probability argument is no longer needed) in total variation distance. To show this, by Pinsker inequality, we have
>
> $\mathrm{TV}(p_*,p_{0,S}^\gets) \le \mathbb{P}[\mathrm{KL}(p_*\Vert p_{0,S}^\gets)=\tilde{O}(\epsilon)]\cdot \sqrt{\frac{1}{2}\mathrm{KL}(p_*\Vert p_{0,S}^\gets)}+\mathbb{P}[\mathrm{KL}(p_*\Vert p_{0,S}^\gets)\not =\tilde{O}(\epsilon)]\cdot \mathrm{TV}(p_*,p_{0,S}^\gets) =\tilde O(\epsilon^{1/2}).$
>
> Then there is no high probability argument in such a sampling error bound, i.e., “it holds with probability $1$’.’ Then, we can arbitrarily choose the number of samples and then the sampling error bound in total variation distance can be exactly formulated as the function of this number.
>
> > *W2: The paper lacks numerical examples to demonstrate their techniques. This is important at it would demonstrate if the method is actually an improvement from the DMC paper. ...*
>
> First, we would like to clarify that the $\tilde{O}(\epsilon^{-5})$ requirement of the total sample number is for the theoretical guarantee. From the theoretical perspective, we have demonstrated that the gradient complexity of our algorithm (see Theorem 4.1 ) has outperformed that of the DMC algorithm.  We have clearly mentioned this after Theorem 4.1.
>
> Second, in practice, both $n_{k,r}$ and $m_{k,r}$ are hyperparameters and we are free to tune them. Then, to compare with DMC regarding their empirical performances, we have performed numerical experiments on 2D multi-mode Gaussian mixture targets (which are non-log-concave with bad isoperimetric properties). We have demonstrated the superior performance of RS-DMC over DMC (as well as ULA) in sampling these hard targets in the experiment. The experimental results are summarized in Appendix G of the latest version and some key comparisons (measured by Maximum Mean Discrepancy) are also provided as follows:
>
> | Grad Num| ULA   | DMC   | RS-DMC |
> | ------------------- | ----- | ----- | ------ |
> | 200                 | 1.386 | 0.101 | 0.087  |
> | 400                 | 1.386 | 0.088 | 0.074  |
> | 800                 | 1.386 | 0.080 | 0.057  |
> | 1600                | 1.382 | 0.078 | 0.044  |
> | 3200                | 1.382 | 0.060 | 0.038  |
>
> Then, it can be seen that in order to match the accuracy obtained by RSDMC with 800 gradient calculations, DMC requires over 3200 gradient calculations. Besides we also visualize the samples generated by ULA, DMC, and RS-DMC. We also observe that the distribution generated by RS-DCM is closer to the target compared to ULA and DMC. More details can be found in Appendix G. We also included our source code with an anonymous link. Besides, we will add more numerical experiments to justify the practical advantages of RS-DMC in future versions.
>
> To sum up, the proposed RS-DMC algorithm is demonstrated to outperform DMC in both theory and experiment. We hope that our clarification and additional experimental results can address your concerns.

---

> > ### Comment · Reviewer_nqUk · 2023-11-21
> >
> > Thanks for addressing my concerns!
> >
> > 1. Regarding the theoretical improvements, these are nice and well appreciated! Thanks for including them!
> > 2. Regarding the experiments I have more concerns here. Before I was concerned about the feasibility of implementing such a method, due to a large number of samples. However, after looking at the code and the resulting plots. I noticed that the generation quality for both DMC and RSDMC is actually very bad, the variances are not at all resembling the target distribution, and despite that MMD gives better results, it doesn't reflect the reality, the samples are very off. For instance a different metric that would show this would be computing the means and variances of every mode. Then we would find that ULA is doing better (judging by the plots). So this is actually a little concerning to me. The reason this is happening is probably that the score is not well approximated, (this is also a limitation of DMC) so we would like to have a clearer explanation as to why, the theoretical work provides a potential answer in that we probably require more samples.
> >
> > The theoretical work done here is very interesting, and provides an idea on how to improve the estimation of the score, however it seems like the method has some drawbacks when it comes to the actual implementation. Despite showing some improvements in the generation accuracy it is using a metric that is advantageous to the method, for this reason I feel like it still requires some work

---

> ### Author Response · Authors · 2023-11-17
> **Response to Reviewer nqUk Part 2**
>
> > *Q1: Because of that I wonder if this method be implementable in practice, considering the computational challenges that come with it?*
>
> RS-DMC is more implementable than DMC in real practice.  Since RS-DMC will be equivalent to DMC when we choose the total recursion number $K =1$. In real practice, the recursion number $K$ is considered a hyperparameter that can be flexibly tuned. You can utilize this hyper-parameter to adjust the iteration number of the inner loops. For example, when you find the iteration number of the inner loop, i.e., $m$, is too large, you can try to reduce it and increase $K$, which may help you save the running time. In summary, RS-DMC will not be worse than DMC and can be much better than DMC when $K>1$ is properly chosen (see our response to your previous questions regarding the numerical experiments).

---

> ### Author Response · Authors · 2023-11-21
>
> Thank you for your insightful feedback once again. We hope that our response addresses your concerns and questions. Specifically, we have decoupled the epsilon dependency in the probability of the algorithm establishment in an additional corollary and added numerical experiments for RS-DMC.
>
> As the end of the author-reviewer discussion is approaching, we would greatly appreciate it if you could let us know whether your previous concerns have been addressed. In any case, we are ready to answer any further questions.

---

> ### Author Response · Authors · 2023-11-22
>
> Thank you for your further feedback. This is a good catch and we agree that MMD may not be perfectly aligned with the “good generation quality”.
>
> However, we would like to clarify that the “bad generation quality” is **not the problem of our algorithm, but the MMD criterion**. In particular, all hyperparameters in our previous experiments are tuned to achieve low MMD, which are misled to generate “bad quality” samples.
>
> We then perform the experiments by using more reasonable hyperparameters and summarize the experimental results in the latest version (Appendix G, Figure 6, Tables 3&4). Then, we can see that RS-DMC can definitely generate “good quality samples” that can well resemble the variance of ground truth. Besides, RS-DMC can also achieve a much lower MMD than ULA (0.112 vs. 1.386) (although the MMD in this case becomes higher than that in the previous experiments).  Moreover, when the gradient complexity is limited to 200,  we compare the variance estimation of all modals between ULA, DMC, and RS-DMC in the following table. It can be seen that RS-DMC better resembles the variance of ground truth for most of modals.
>
> | Var     | ULA    | DMC    | RS-DMC | ground truth |
> | ------- | ------ | ------ | ------ | ------------ |
> | Modal 0 | 0.0337 | 0.0337 | 0.0332 | 0.0217 |
> | Modal 1 | 0.0574 | 0.0311 | 0.0314 | 0.0188 |
> | Modal 2 | 0.0459 | 0.0299 | 0.0324 | 0.0196 |
> | Modal 3 | 0.0394 | 0.0338 | 0.0287 | 0.0184 |
> | Modal 4 | 0.0449 | 0.0330 | 0.0289 | 0.0174 |
> | Modal 5 | 0.0409 | 0.0321 | 0.0306 | 0.0220 |
>
> Finally, we would like to clarify that although MMD may not be a good enough criterion, it still reflects some statistical information beyond variance, such as the coverage of all modes. Our experiments demonstrate the superior performance of RS-DMC over ULA under this criterion. We believe similar results can be found in other criteria and we will conduct more experiments with different criteria in the future.

---

> ### Comment · Reviewer_nqUk · 2023-11-22
>
> Thanks for adding these experiments. I think this is very important and useful to better understand the algorithm. I appreciate the authors effort to do so! There is still some questions that I would like to see answered, like how well does the method approximate the score for instance. I would also like to add that the real ground truth is that the variances should add to .02, therefore this is actually like a 50% discrepancy from the real result, it would be nice to explain if we see these numbers converge or not.Considering the deadline and the significant improvements to the experiments I will increase my rating. I still expect that the ground truth is updated to the actual value in the final version of the paper.

---

> > ### Comment · Reviewer_nqUk · 2023-11-22
> >
> > I apologize for all the going back and forth. There are some things that concern me, and actually push me towards lowering my rating. I was playing with your code, and something as simple as moving sigma = 1  completely breaks the ability of every sampler to work (you can see how increasing sigma slowly breaks everything). I also tested the accuracy of the score evaluation and it seems very off, this last one doesn't surprise due to different reasons, like using very little langevyn samples and iterations. In the current state I feel like there is some very big glaring problem in DMC and that you are also victim off, but no one has investigated and its of big importance to understanding if the method is working. Because of this I will lower my review.

---

> ### Author Response · Authors · 2023-11-23
>
> We are sorry for causing the confusion due to our previous bad organization of the code.  We only show the parameters tuned for sigma = 0.1 in our previous code. When you consider the target distribution with sigma=1, the reason for the breaks of every sampler is the improper choice of hyper-parameters.
>
> You need to find a proper step size for ULA, and proper inner step sizes for DMC/RS-DMC. In the latest version, we have reorganized and updated our code for easier hyperparameter tunning. We have also included the results, **in the latest version of demo.ipynb in our supplementary**, as well as proper hyperparameters for sigma=1 and sigma=2 (where we basically enlarge the inner step size for larger $\sigma$, we hope the choice of their hyper-parameters will provide some tuning intuition for you), where RS-DMC/DMC can perform successful sampling in these cases.
>
> In particular, to save the sample size, we implement DMC/RS-DMC  by replacing a few (only 10) DMC/RS-DMC steps in the final stage (when $p_t$ approaches to the target distribution $p_\ast$) with the ULA steps. In fact, these ULA steps serve as the DMC/RS-DMC steps with a more accurate score estimation (compared to our score estimation with finite samples). In our latest code (see demo.ipynb), we have explicitly shown experimental results including both pure DMC/RS-DMC and DMC/RS-DMC with few ULA steps. This ablation study demonstrates that the DMC/RS-DMC has advantages to fitting the modes and a few ULA steps help us to refine the local structure.
>
> In fact, DMC/RS-DMC, even with inaccurate score estimation, has the advantage of discovering the global structure of the target distribution (e.g., coverage of the modes), compared to ULA, while ULA may be better to handle the local structure of distribution as it makes use of the exact score. Our experiments show that incorporating a few ULA steps into DMC/RS-DMC (using a few samples) can recover both the global and local structures of the target distribution, thus leading to good sampling performance.
>
> Finally, we clarify that the major contributions and claims in our paper are regarding the theoretical gradient complexity, there is no sampling algorithm that can handle general non-log-concave distributions with provable quasi-polynomial guarantees. These theoretical results suggest  RS-DMC can be promising for sampling hard distributions, which can be further investigated and improved in both theory and practice. Besides, our current experiments are very primitive and conducted within a very short time, we have consistently refined them to achieve better performance during the rebuttal. Your suggestions and our experimental observations during the rebuttal phase are indeed very valuable for us to further understand the empirical behavior of DMC/RS-DMC and develop better practical designs. These can be definitely left as promising future directions to explore.

---

### Meta-Review · Area_Chair_7bh1 · 2023-12-05

**Metareview:**

This paper considers the problem of sampling from unnormalized density, and aims at improving the recent work of reverse diffusion Monte Carlo. There, the reverse SDE of a diffusion process is employed to push forward easy-to-sample distribution to the target distribution. A key nontriviality for doing so is to estimate the score function, and the original approach reportedly suffers from high gradient complexity. This paper proposes to improve the score estimation by dividing the entire diffusion process into multiple segments and recursively solve certain sampling sub-problems. It also provides theoretical demonstration of how this technique turns the gradient complexity from exponential to quasi-polynomial. All reviewers and I found this idea interesting, and the strong theoretical component of this paper is also appreciated. However, reviewers nqUk, Rkvn, and FZMK raised similar concerns about the experimental results, and nqUk also looked into the code but was not convinced. Further discussion among the reviewers (moderated by the AC) led to a popular assessment that, this paper proposes a method that can be implemented, but the implementation needs parameter tuning, and for each problem a different, nontrivial tuning is needed, which may undermine the claimed computational efficiency. This overhead is not taken into account in the theoretical demonstration, but not much empirical demonstration is provided. Therefore, I'm afraid I cannot recommend acceptance this time, but I hope the authors could find reviewers' comments helpful and submit a revised version in the future.

**Justification For Why Not Higher Score:**

3 reviewers had common concerns about the experimental results and questioned about the implementability of the proposed method. These concerns unfortunately did not get fully resolved post rebuttal and reviewer-AC discussion.

**Justification For Why Not Lower Score:**

N/A

---

### Decision · Program_Chairs · 2024-01-16

Reject